# Development of global temperature and pH calibrations based on bacterial 3-hydroxy fatty acids in soils

Pierre Véquaud[1], Sylvie Derenne[1], Alexandre Thibault[2], Christelle Anquetil[1], Giuliano Bonanomi[3], Sylvie Collin[1], Sergio Contreras[4], Andrew T. Nottingham[5,6], Pierre Sabatier[7], Norma Salinas[8], Wesley Phillip Scott[9], Josef P. Werne[9], Arnaud Huguet[1]

[1]Sorbonne Université, CNRS, EPHE, PSL, UMR METIS, Paris, 75005, France
[2]Antea Group, Innovation Hub, 803 boulevard Duhamel du Monceau, Olivet, 45160, France
[3]Dipartimento di Agraria, Università di Napoli Federico II, via Università 100, Portici, NA, 80055, Italy
[4]Laboratorio de Ciencias Ambientales (LACA), Departamento de Química Ambiental, Facultad de Ciencias & Centro de Investigación en Biodiversidad y Ambientes Sustentables (CIBAS), Universidad Católica de la Santísima Concepción, Casilla 297, Concepción, Chile
[5]School of Geosciences, University of Edinburgh, Crew Building, Kings Buildings, Edinburgh EH9 3FF United Kingdom
[6]School of Geography, University of Leeds, Leeds, United Kingdom
[7]Univ. Savoie Mont Blanc, CNRS, EDYTEM, Le Bourget du Lac, 73776, France
[8]Instituto de Ciencias de la Naturaleza, Territorio y Energías Renovables, Pontificia Universidad Catolica del Peru, Av. Universitaria 1801, San Miguel, Lima 32, Peru
[9]Department of Geology and Environmental Science, University of Pittsburgh, Pittsburgh, PA 15260, USA

*Correspondence to*: Arnaud Huguet (arnaud.huguet@sorbonne-universite.fr)

**Abstract.** 3-hydroxy fatty acids (3-OH FAs) with 10 to 18 C atoms are membrane lipids mainly produced by Gram-negative bacteria. They have been recently proposed as temperature and pH proxies in terrestrial settings. Nevertheless, the existing correlations between pH/temperature and indices derived from 3-OH FA distribution (RIAN, $RAN_{15}$ and $RAN_{17}$) are based on a small soil dataset (ca. 70 samples) and only applicable regionally. The aim of this study was to investigate the applicability of 3-OH FAs as mean annual air temperature (MAAT) and pH proxies at the global level. This was achieved using an extended soil dataset of 168 topsoils distributed worldwide, covering a wide range of temperatures (5°C to 30°C) and pH (3 to 8). The response of 3-OH FAs to temperature and pH was compared to that of established branched GDGT-based proxies ($MBT'_{5Me}$/CBT). Strong linear relationships between 3-OH FA-derived indices ($RAN_{15}$, $RAN_{17}$ and RIAN) and MAAT/pH could only be obtained locally, for some of the individual transects. This suggests that these indices cannot be used as paleoproxies at the global scale using simple linear regression models, in contrast with the $MBT'_{5Me}$ and CBT. However, strong global correlations between 3-OH FA relative abundances and MAAT/pH were shown by using other algorithms (multiple linear regression, k-NN and random forest models). The applicability of the three aforementioned models for paleotemperature reconstruction was tested and compared with the MAAT record from a Chinese speleothem.

The calibration based on the random forest model appeared to be the most robust. It generally
showed similar trends with previously available records and highlighted known climatic events
poorly visible when using local 3-OH FA calibrations. Altogether, these results demonstrate
the potential of 3-OH FAs as paleoproxies in terrestrial settings.

**Keywords:** 3-hydroxy fatty acids; branched GDGTs; soils; global calibration; temperature and
pH proxy


## 1. Introduction

Investigating past climate variations is essential to understand and predict future environmental changes, especially in the context of global anthropogenic change. Direct records of environmental parameters are available for the last decades, the so-called "instrumental" period. Beyond this period, proxies can be used to obtain indirect information on environmental parameters. A major challenge is to develop reliable proxies which can be applied to continental environments in addition to marine ones. Indeed, available proxies have been mainly developed and used in marine settings, as the composition and mechanism of formation of marine sedimentary cores is less complex than in continental settings, which are highly heterogeneous. Several environmental proxies based on organic (e.g. the alkenone unsaturation index ($U^{k'}_{37}$; Brassell et al., 1986) and inorganic (Mg/Ca ratio and $^{18}O/^{16}O$ ratio of foraminifera; Emiliani, 1955; Erez and Luz, 1983) fossil remains were notably developed for the reconstruction of sea surface temperatures.

Some of the existing proxies are based on membrane lipids synthesized by certain microorganisms (Eglinton and Eglinton, 2008; Schouten et al., 2013). These microorganisms are able to adjust the composition of their membrane lipids in response to the prevailing environmental conditions in order to maintain an appropriate fluidity and to ensure the optimal state of the cellular membrane (Singer and Nicolson, 1972; Sinensky, 1974; Hazel and Williams, 1990; Denich et al., 2003). The structure of glycerol dialkyl glycerol tetraethers (GDGTs), which are membrane lipids biosynthesized by archaea and some bacteria, is especially known to be related to environmental conditions. Archaeal GDGTs are constituted of isoprenoid alkyl chains ether-linked to glycerol, whereas bacterial GDGTs are characterized by branched alkyl chains instead of isoprenoid ones. The latter compounds are ubiquitous in terrestrial (Weijers et al., 2007; Peterse et al., 2012; De Jonge et al., 2014; Naafs et al., 2017) and aquatic environments (Peterse et al., 2009; Tierney and Russell, 2009; Sinninghe Damsté et al., 2009; Loomis et al., 2012; Peterse et al., 2015; Weber et al., 2015). These branched GDGTs (brGDGTs) are produced by still unidentified bacteria, although some of them may belong to the phylum *Acidobacteria* (Sinninghe Damsté et al., 2011, 2014, 2018). The analysis of brGDGTs in a large number of soils distributed worldwide showed that the relative distribution of these compounds is mainly related to mean annual air temperature (MAAT) and soil pH (Weijers et al., 2007; Peterse et al., 2012; De Jonge et al., 2014). Even though brGDGT proxies were largely investigated over the last 10 years (De Jonge et al., 2014; Dearing Crampton-Flood et al., 2020) and were applied to various paleorecords (e.g, Coffinet et al.,

2018; Wang et al., 2020), new molecular proxies, independent of and complementary to
brGDGTs, are needed to improve the reliability of temperature reconstructions in terrestrial
settings.

Recent studies have unveiled the potential of another family of bacterial lipids − 3-
hydroxy fatty acids (3-OH FAs) − for temperature and pH reconstructions in terrestrial (Wang
et al., 2016, 2018; Huguet et al., 2019) and marine (Yang et al., 2020) settings. 3-OH FAs with
10 to 18 carbon atoms are specifically produced by Gram-negative bacteria and are bound to
the lipopolysaccharide (LPS) by ester or amide bonds (Wollenweber et al., 1982; Wollenweber
and Rietschel, 1990). Three types of 3-OH FAs can be distinguished, with either *normal* chains
or branched chains, *iso* or *anteiso*.

The analysis of 3-OH FAs in soils showed that the ratio of $C_{15}$ or $C_{17}$ *anteiso* 3-OH
FA to *normal* $C_{15}$ or $C_{17}$ 3-OH FA ($RAN_{15}$ and $RAN_{17}$ indices, respectively) were negatively
correlated with MAAT along the three mountains investigated so far: Mts. Shennongjia (China;
Wang et al., 2016), Rungwe and Majella ( Tanzania and Italy, respectively; Huguet et al., 2019).
This suggests that Gram-negative bacteria producing these fatty acids respond to colder
temperatures with an increase in *anteiso*-$C_{15}/C_{17}$ vs. *n*-$C_{15}/C_{17}$ 3-OH FAs, in order to maintain
a proper fluidity and optimal state of the bacterial membrane, the so-called homeoviscous
adaptation mechanism (Sinensky, 1974; Hazel and Eugene Williams, 1990). Nevertheless, the
relationships between $RAN_{15}$ and MAAT along the three mountain transects showed the same
slopes but different intercepts (Wang et al., 2016; Huguet et al., 2019), suggesting that regional
or local $RAN_{15}$ relations may be more appropriate to apply for temperature reconstructions in
terrestrial settings. In contrast, a significant calibration between $RAN_{17}$ and MAAT could be
established using combined data from the three mountain regions (Wang et al., 2016; Huguet
et al., 2019).

Another index, defined as the cologarithm of the sum of *anteiso* and *iso* 3-OH FAs
divided by the sum of *normal* homologues (RIAN index), was shown to be strongly negatively
correlated with soil pH along the three aforementioned mountains (Wang et al., 2016; Huguet
et al., 2020), reflecting a general relative increase in normal homologues compared to branched
(*iso* and *anteiso*) ones with increasing pH. This mechanism was suggested to reduce the
permeability and fluidity of the membrane for the cell to cope with lower pH (Russell et al.,
1995; Denich et al., 2003; Beales, 2004).

3-OH FA indices were recently applied for the first time to the reconstruction of the
temperature and hydrological changes over the last 10,000 years in a speleothem from China
(Wang et al., 2018), showing the potential of 3-OH FAs as independent tools for environmental

reconstruction in terrestrial settings. A very recent study based on marine sediments from the North Pacific Ocean suggested that the distribution of 3-OH FAs could also be used to reconstruct sea surface temperature (Yang et al., 2020).

Even though these results are promising, the linear regressions between pH/MAAT and 3-OH FA indices in terrestrial environments are still based on a rather small dataset (ca. 70 soil samples; Wang et al., 2016; Huguet et al., 2019). The aim of this study was to investigate the applicability of 3-OH FAs as MAAT and pH proxies at the global level using an extended soil dataset and refined statistical tools. 3-OH FA distribution from 54 soils was determined in four globally distributed altitudinal transects (Tibet, Italy, Peruvian Andes and Chile) and was combined with data previously published by Wang et al. (2016; Mt Shennongjia, China), Huguet et al. (2019; Mt. Rungwe, Tanzania and Mt. Majella, Italy) and Véquaud et al. (2021; Mts. Lautaret-Bauges, France), leading to a total of 168 samples. In addition to linear regressions, non-parametric, machine learning models were used to improve the global relationships between 3-OH FA distribution and MAAT/pH. These models present the advantage of taking into account non-linear environmental influences, in line with the intrinsic complexity of the environmental settings. Finally, these new models were tested and compared by applying them to a speleothem archive (Wang et al., 2018) representing to date the only available MAAT record derived from 3-OH FA proxies in continental setting. As brGDGTs are the only microbial organic proxies which can be used for temperature and pH reconstructions in terrestrial settings so far, they can serve as a reference proxy to understand the temperature and pH dependency of 3-OH FAs analyzed in the same dataset. 3-OH FAs and brGDGTs have thus been concomitantly analyzed to assess their reliability and complementarity as independent temperature and pH proxies.

## 2. Material and methods

### 2.1. Soil dataset

#### 2.1.1. Study sites

The dataset of the present study is comprised of the globally distributed surface soils previously analyzed for brGDGTs and 3-OH FAs and collected along 4 altitudinal transects: Mts. Shennongjia (China; Yang et al., 2015; Wang et al., 2016), Rungwe (Tanzania; Coffinet et al., 2017; Huguet et al., 2019), Majella (Italy; Huguet et al., 2019) and Lautaret-Bauges

(France; Véquaud et al., 2021). This set was extended with surficial soils (0-10 cm) from 4
additional altitudinal transects described below, located in Italy, Tibet, Peru and Chile (Table
1).

Soil samples were collected from 13 sites along Mount Pollino in the Calabria region
(Italy) between 0 and 2,200 m above sea level (a.s.l.) (Table 1). Mt. Pollino is located in the
calcareous Apennine range and is 2,248 m a.s.l. It is framed to the northwest by the Sierra de
Prete (2,181 m high) and to the south by the Pollino Abyss. The alpine to subalpine area (above
2,100 m a.s.l.) is characterized by the presence of Mediterranean grasslands (*Festuca bosniaca*,
*Carex kitaibeliana*) and the presence of sinkholes (Todaro et al., 2007; Scalercio et al., 2014).
The mountainous vegetation (over 1,200 m a.s.l.) is dominated by *Fagus sylvatica* forests and,
at the treeline, by scattered *Pinus leucodermis* (Bonanomi et al., 2020). The soil is poorly
developed and dominated by calcareous soils. Between 0 to 1,200 m a.s.l (Scalercio et al., 2014
and reference therein), Mt. Pollino is characterized by the presence of *Q. ilex* forests or shrubs.
Climate along this mountain is humid Mediterranean, with high summer temperatures and an
irregular distribution of rainfall throughout the year with pronounced summer drought (39.5%
in winter, 23.7% in spring, 29.2% in autumn, 7.6% in summer; average annual precipitation:
1,570 mm; see Todaro et al., 2007). MAAT is comprised between 7 °C (2,200 m a.s.l) and 18
°C (0 m a.s.l; Scalercio et al., 2014). MAAT along Mt. Pollino was estimated using a linear
regression between two MAAT (16°C at 400 m a.s.l and 10°C at 1,600 m a.s.l.)  from the
meteorological data (Castrovillari station) recorded by Scalercio et al. (2014). The pH of the
soils analyzed in the present study ranges between 4.5 and 6.8 (Table 1).

Soil samples were collected from 17 sites along Mount Shegyla between 3,106 and
4,474 m a.s.l. (southeastern Tibet, China), as previously described by Wang et al. (2015).
Different climatic zonations are observed along this high-altitude site (2,700 to 4,500 m a.s.l):
(i) a mountainous temperate zone between 2,700 and 3,400 m, (ii) a subalpine cold temperate
zone between 3,400 and 4,300 m and (iii) a cold alpine zone above 4,300 m. Plant species, such
as brown oak (*Q. semecarpifolia*) or common fir (*Abies alba*) are abundant within the
mountainous and subalpine levels. In the cold subalpine zone, the Forrest's fir (*Abies georgei*
*var. smithii)* is endemic to western China. In the cold alpine zone, coniferous species (*Sabina*
*saltuaria*) as well as species typical of mountainous regions such as *Rhododendron* are
observed. MAAT was estimated using a linear regression between 7 measured MAAT from the
data recorded by Wang et al. (2015). The average MAAT along the transect is 4.6°C, with a
minimum of 1.1 °C at ca. 4,500 m a.s.l. and a maximum of 8.9 °C at ca. 3,100 m a.s.l. (Table
1). Soil pH ranges between 4.6 and 6.4 (Table 1).

Soils were sampled from 14 sites in the Peruvian Andes along the Kosñipata transect, located in south-eastern Peru, in the upper part of the Madre de Dios/Madeira watershed, east of the Andes Cordillera (Nottingham et al., 2015). This transect (190 m to 3,700 m a.s.l) is well-documented and is the object of numerous ecological studies (Malhi et al., 2010; Nottingham et al., 2015). There is a shift in vegetation zonation with increasing elevation, from tropical lowland forest to montane cloud forest and high-elevation 'Puna' grassland. The tree line lies between 3,200 and 3,600 m a.s.l. For the 14 sites sampled in this study, the lower 13 sites are forest and the highest site is grassland. The 14 sites are part of a network of 1 ha forest plots (Nottingham et al., 2015); for each 1 ha plot, 0-10 cm surface soil was sampled from 5 systematically distributed locations within each 1 ha plot. Mean annual precipitation does not vary significantly with altitude (mean $=2448$ mm.y$^{-1}$, SD $= 503$ mm.y$^{-1}$; Rapp and Silman, 2012; Nottingham et al., 2015). MAAT is comprised between 26.4 °C at 194 m altitude and 6.5°C at 3644 m altitude (Table 1). The pH is characteristic of acidic soils (3.4 - 4.7; Table 1). Further information on these sites and soils is available in Nottingham et al. (2015).

Soil samples were collected from 10 sites between 690 m and 1,385 m a.s.l. from the lake shore (20 to 50 m offshore) of 10 Andean lakes located in Chile (38–39°S) within the temperate forest (Table 1). High-frequency measurements of MAAT over a period of one year are available for the different sampling sites. MAAT is comprised between 5.75°C and 9.2°C. Soil pH ranges between 4.4 and 6.8 (Table 1).

### 2.1.2. pH measurement

Following sampling, soils were immediately transported to the laboratory and stored at -20 °C. Soil samples from the Peruvian Andes, Mt. Pollino and Mt. Shegyla were then freeze-dried, ground and sieved at 2 mm. The pH of the freeze-dried samples was measured in ultrapure water with a 1:2.5 soil water ratio. Typically, 10 ml of ultrapure water were added to 4 g of dry soil. The soil solution was stirred for 30 min, before decantation for 1 hand pH measurement (Carter et al., 2007).

## 2.2. Lipid analyses

BrGDGTs and 3-OH FAs were analyzed in all samples from the Peruvian Andes, Chilean Andes, Mt. Pollino and Mt. Shegyla.

 *2.2.1. 3-OH FA analysis*

Sample preparation for 3-OH FA analysis was identical to that reported by Huguet et

al. (2019) and Véquaud et al. (2021). Soil samples were subjected to acid hydrolysis (3 M HCl)
and extracted with organic solvents. This organic fraction was then rotary-evaporated,
methylated in a 1M HCl-MeOH solution at 80 °C for 1 h and separated into three fractions over
an activated silica column: (i) 30 ml of heptane/EtOAc (98: 2), (ii) 30 ml of EtOAc and (iii) 30
ml of MeOH. 3-OH FAs contained in the second fraction were derivatized at 70°C for 30 min
with a solution of *N,O*- bis(trimethylsilyl)trifluoroacetamide (BSTFA) – Trimethylchlorosilane
(TMCS) 99:1 (Grace Davison Discovery Science, USA) before gas chromatography-mass
spectrometry (GC-MS) analysis.

3-OH FAs were analyzed with an Agilent 6890N GC-5973N using a Restek RXI-5 Sil

MS silica column (60 m × 0.25 mm, i.d. 0.25 μm film thickness), as previously described
(Huguet et al., 2019). 3-OH FAs were quantified by integrating the appropriate peak on the ion
chromatogram and comparing the area with an internal standard (3-hydroxytetradecanoic acid,
2,2,3,4,4-d5; Sigma-Aldrich, France). The internal standard (0.5 mg/ml) was added just before
injection as a proportion of 3 µl of standard to 100 µl of sample, as detailed by Huguet et al.
(2019). The different 3-OH FAs were identified based on their retention time, after extraction
of the characteristic *m/z* 175 fragment (*m/z* 178 for the deuterated internal standard; cf. Huguet
et al., 2019).

The RIAN index was calculated as follows (Wang et al., 2016 ; Eq. 1) in the range

$C_{10}$-$C_{18}$ :

$\text{RIAN} = -\log[(I + A)/ N]$                     (1)

where I, A, N represent the sum of all *iso*, *anteiso* and *normal* 3-OH FAs, respectively.


$RAN_{15}$ and $RAN_{17}$ indices are defined as follows (Wang et al., 2016; Eq. 2 and 3):

$RAN_{15} = [\ anteiso\ C_{15}] / [normal\ C_{15}]$             (2)

$RAN_{17} = [\ anteiso\ C_{17}] / [normal\ C_{17}]$             (3)

Analytical errors associated with the calculation of RIAN, $RAN_{15}$ and $RAN_{17}$ indices

are respectively 0.006, 0.3 and 0.2 based on the analysis of one sample injected nine times
during the analysis and five samples injected in triplicates.

*2.2.2. brGDGT analysis*

Sample preparation for brGDGT analysis was similar to that reported by Coffinet et

al. (2014). Briefly, ca. 5-10 g of soil was extracted using an accelerated solvent extractor (ASE

100, Dionex-ThermoScientific, USA) with a dichloromethane (DCM) / methanol (MeOH) mixture (9: 1) for 3×5 min at 100 °C and a pressure of 100 bars in 34 ml cells. The total lipid extract was rotary evaporated and separated into two fractions of increasing polarity on a column of activated alumina: (i) 30 ml of heptane: DCM (9: 1, v:v) ; (ii) 30 ml of DCM: MeOH (1: 1, v:v). GDGTs are contained in the second fraction, which was rotary evaporated. An aliquot (300 µL) was re-dissolved in heptane and centrifuged using an Eppendorf MiniSpin centrifuge (Eppendorf AG, Hamberg, Germany) at 7000 rpm for 1 min.

GDGTs were then analyzed by high pressure liquid chromatography coupled with mass spectrometry with an atmospheric pressure chemical ionisation source (HPLC-APCI-MS) using a Shimadzu LCMS 2020. GDGT analysis was performed using two Hypersil Gold silica columns in tandem (150 mm × 2.1 mm, 1.9 µm; Thermo Finnigan, USA) thermally controlled at 40 °C, as described by Huguet et al. (2019). This methodology enables the separation of 5- and 6-methyl brGDGTs. Semi-quantification of brGDGTs was performed by comparing the integrated signal of the respective compound with the signal of a $C_{46}$ synthesized internal standard (Huguet et al., 2006) assuming their response factors to be identical.

The MBT'$_{5Me}$ index, reflecting the average number of methyl groups in 5-methyl isomers of GDGTs and considered as related to MAAT, was calculated according to De Jonge et al. (2014; Eq. 4):

$$\text{MBT'}_{5Me} = \frac{[Ia+Ib+Ic]}{[Ia+Ib+Ic]+[IIa+IIb+IIc]+[IIIa]} \tag{4}$$

The CBT' index, reflecting the average number of cyclopentyl rings in GDGTs and considered as related to pH, was calculated as follows (De Jonge et al., 2014; Eq. 5 ):

$$CBT' = \log \left( \frac{[Ic]+[IIa\prime]+[IIb\prime]+[IIc\prime]+[IIIa\prime]+[IIIb\prime]+[IIIc\prime]}{[Ia]+[IIa+IIIa]} \right) \tag{5}$$

The Roman numerals correspond to the different GDGT structures presented in De Jonge et al. (2014). The 6-methyl brGDGTs are denoted by an apostrophe after the Roman numerals for their corresponding 5-methyl isomers. Analytical errors associated with the calculation of MBT'$_{5Me}$ and CBT' indices are 0.015 and 0.02 respectively, based on the analysis of three samples in triplicate among the 44 soil samples.

## 2.3. Statistical analysis

In order to investigate the correlations between environmental variables (pH, MAAT) and the relative abundances of bacterial lipids (brGDGTs and 3-OH FAs) or the indices based on these compounds, pairwise correlation matrices were performed in addition to single or multiple linear regressions. As the dataset is not normally distributed, Spearman correlation was used with a confidence level of 5%.

Principal component analyses (PCA) were performed on the different soil samples to statistically compare the 3-OH FA/brGDGT distributions along the different altitudinal transects. The fractional abundances of the bacterial lipids (3-OH FAs and brGDGTs) were used for these PCAs, with MAAT, pH and location of the sampling site representing supplementary variables (i.e. not influencing the principal components of the analysis).

Independent models should be used for the development of environmental calibrations, as each of them has its own advantages and limits. Linear regression methods are simple to use but many of them suffer from the phenomenon of regression dilution, as previously noted (Naafs et al., 2017; Dearing Crampton-Flood et al., 2020). That is why other models than ordinary least squares or single/multiple regression were also proposed in this study (cf. section 4.2. for discussion of the models): the k-nearest neighbor (k-NN) and random forest models. These models are based on machine-learning algorithms, which are built on a proportion of the total dataset (randomly defined, i.e., training dataset) and then tested on the rest of the dataset, considered as independent (test dataset).

The k-NN model is based on the estimation of the mean distances between the different samples. This is a supervised learning method (e.g. Gangopadhyay et al., 2009). A training database composed of N "input-output" pairs is initially constituted to estimate the output associated with a new input x. The method of the k-neighbors takes into account the k training samples whose input is the closest to the new input x, according to a distance to be defined. This method is non-parametric and is used for classification and regression. In k-NN regression, the result is the value for this object, which is the average of the values of the k nearest neighbors. Its constraints lie in the fact that, by definition, if a range of values is more frequent than the others, then it will be statistically predominant among the k closest neighbors. To overcome this limitation of the k-NN method, data selection was performed randomly on the dataset with a stratification modality according to the MAAT or the pH. This approach allows to limit the impact of extreme values as detailed below.

The random forest algorithm is also a supervised learning method used, among other
things, for regressions (e.g. Ho, 1995; Breiman, 2001; Denisko and Hoffman, 2018;). This
model works by constructing a multitude of decision trees at training time and producing the
mean prediction of the individual trees. Decision tree learning is one of the predictive modeling
approaches used to move from observations to conclusions about the target value of an item.
Decision trees where variables are continuous values are called regression trees.
The training phase required for the random forests, k-NN and multiple linear
regression was performed on 75% of the sample set with an iteration of ten cross-validations
per model. Data selection was performed randomly on the dataset (with no pre-processing of
the individual 3-OH FAs) but with a stratification modality according to the MAAT or the pH
to limit the impact of extreme values on the different models used.  Then, the robustness and
precision of the different models were tested on the remaining 25 % of samples, considered as
an independent dataset. Simple and Multiple linear regressions, PCA, k-NN and random forest
models were performed with R software, version 3.6.1 (R Core Team, 2014) using the packages
- tidymodels (version 0.1.0)- kknn (version 1.3.1), ranger (version 0.11.2). A web application
is available online (https://athibault.shinyapps.io/paleotools) for the reconstruction of 3-OH
FA-derived MAAT using the machine learning models proposed in the present study.


**3. Results**
**3.1. Distribution of bacterial lipids**
*3.1.1. 3-OH FAs*
3-OH FAs were identified in the whole dataset, representing eight elevation transects
and 168 samples (Supplementary table 1; Yang et al., 2015; Wang et al., 2016; Coffinet et al.,
2017; Huguet et al., 2019; Véquaud et al., 2021). Their chain lengths range between 8 and 26
C atoms, indicating that these compounds have various origins (bacteria, plants, and fungi;
Zelles, 1999; Wang et al., 2016 and reference therein). The homologues of 3-OH FAs with 10
to 18 C atoms are considered to be produced exclusively by Gram-negative bacteria
(Wollenweber and Rietschel, 1990; Szponar et al., 2003) and will be the only ones considered
in the following. Compounds with an even carbon number and *normal* chains were the most
abundant 3-OH FAs in all samples (mean 67.9 % of the total 3-OH FAs, Standard Deviation
(SD) 6.8 %), with a predominance of the $n$-$C_{14}$ homologue (21.9 %, SD 3.23 %; Fig. 1). *Iso* (mean
22.9%, SD 5.01%) and *anteiso* (mean 6.33 %, SD 1.79%) isomers were also present. It must be
noted that *anteiso* isomers were only detected for odd carbon-numbered 3-OH FAs (Yang et
al., 2015; Wang et al., 2016; Coffinet et al., 2017; Huguet et al., 2019).

The distribution of 3-OH FAs in the soils of the different altitudinal transects did not

show a large variability (Fig. 1). Thus, there was no major difference in the relative abundances
of most of the 3-OH FAs ($i$-$C_{11}$, $a$-$C_{11}$, $n$-$C_{11}$, $i$-$C_{12}$, $a$-$C_{13}$, $n$-$C_{13}$, $i$-$C_{14}$, $n$-$C_{15}$, $i$-$C_{16}$, $a$-$C_{17}$ and
$n$-$C_{17}$) between the 8 study sites, even though slight differences could be observed for some
compounds as detailed below. For example, the Peruvian samples were characterized by higher
average proportions of $n$-$C_{18}$ 3-OH FA and lower contribution of the $n$-$C_{10}$ and $n$-$C_{12}$
homologues than those from the other transects. Soils from Mt. Shegyla were characterized by
lower average proportions of $n$-$C_{14}$ 3-OH FAs and higher abundances of $i$-$C_{17}$ compounds
compared to the other transects (Fig. 1).

*3.1.2. brGDGTs*

The relative abundances of brGDGTs were compared between the same transects as

for 3-OH FAs, representing a total of 168 samples. The 5- and 6-methyl isomers were separated
in most of the samples (Fig. 2, Supp. Table 2), except in older dataset, i.e. soils from Mt.
Rungwe (Coffinet et al., 2014, 2017). BrGDGT data from Mt. Rungwe will not be further
considered in this study.

The brGDGT distribution was dominated by acyclic compounds (Ia, IIa, IIa', IIIa,

IIIa') which represent on average ca. 83.4% of total brGDGTs (SD = 14.5%; Fig. 2). The
tetramethylated (Ia-c; mean 39.3%, SD of 20.5%) and the pentamethylated (IIa-c; 44.8%, SD
12.8%) brGDGTs were predominant over the hexamethylated ones (IIIa-c; Fig. 2). The 5-
methyl isomers were on average present in a higher proportion (mean 71.9%, SD 23.4%) than
the 6-methyl compounds (Fig. 2).

High variability of the brGDGT distribution was observed among the different

transects. The relative abundance of brGDGT Ia was much higher in the Peruvian soils (mean
83%, SD 12.6%) than in the other transects (mean between 17.3% and 61.7%; Fig. 2). The 5-
methyl isomers were more abundant than the 6-methyl isomers for all sites except for Mt.
Pollino (mean 5-methyl = 44%, SD=11.7%) and Mt. Majella (mean 5-methyl = 33.7 %, SD =
5.5%; Fig. 2).

 **3.2. 3-OH FA and brGDGT-derived indices**

 *3.2.1. 3-OH FA*

 The RIAN index varied between 0.1 and 0.8 among the eight elevation transects (Table

 1). The RIAN index ranged from 0.37 to 0.67 for the Peruvian Andes, 0.23 to 0.56 for Mt.

 Shegyla, 0.15 to 0.34 for Mt. Pollino, 0.21 to 0.53 for the Chilean Andes, 0.26 to 0.80 for Mt.

 Rungwe (Huguet et al., 2019), 0.16 to 0.46 for Mt. Majella (Huguet et al., 2019), 0.20 to 0.69

 for Mt. Shennongjia (Wang et al., 2016) and 0.13 to 0.56 for the French Alps (Véquaud et al.,

 2021).

 The $RAN_{15}$ varied greatly among the different sites (Table 1). It was in the same range

 along Mts. Rungwe (1.04-5.73) and Majella (0.68-6.43; Huguet et al., 2019). In contrast, its

 upper limit was higher for Mts. Shennongjia (0.68-10.18; Wang et al., 2016), Shegyla (4.07-

 12.17), Pollino (2.41-10.26), the Peruvian Andes (2.45-13.77) and the French Alps (1.44-

 12.26). The range of variation in $RAN_{15}$ was narrower for the Chilean Andes (3.82-6.40).

 The $RAN_{17}$ values were similar among the different altitudinal transects (Table 1),

 ranging from 1.72 to 3.90 along Mt. Shegyla, 0.73 to 4.75 along Mt. Majella (Huguet et al.,

 2019), 1.19 to 4.54 along Mt. Pollino, 1.91 to 4.25 for the Chilean Andes and 1.12 to 3.57 along

 Mt. Shennongjia (Wang et al., 2016). The range of $RAN_{17}$ values was narrower for Mt. Rungwe

 (0.33-1.62; Huguet et al., 2019) and the Peruvian Andes (0.61-2.39) and wider for the French

 Alps (0.89-6.42; Véquaud et al., 2021) compared to the other sites.

 *3.2.2. brGDGT*

 The range of variation in the MBT'$_{5Me}$ index was homogeneous along most transects

 (0.32-0.63; Table 1), except the Peruvian Andes, with higher values (0.58-0.98; Table 1).

 Regarding the CBT' index, it showed similar ranges along Chilean Andes (-2.28 to -0.32) and

 Mt. Shegyla (-2.39 to -0.35; Table 1). This index showed different ranges of variations along

 the other altitudinal transects, Mts. Shennongjia (-1.18 to 0.50; Yang et al., 2015), Pollino (-

 0.24 to 0.43) and Peruvian Andes (-1.91 to -1.09). Finally, The CBT' values varied within a

 narrow range along Mt.Majella (0.23-0.59; Huguet et al., 2019) and within a wide range along

 the French Alps (-2.29 to 0.52 ; Véquaud et al., 2021).

### 3.3. Principal component analysis and clustering of 3-OH FA and brGDGT distribution

Principal component analyses were performed to refine the comparison of bacterial lipid distribution (3-OH FAs and brGDGTs) among the different altitudinal transects.

*3.3.1. 3-OH FA*

The first two axes of the 3-OH FA PCA explained 39.1% of the total variance in the dataset (Fig. 3a). Dimension 1 (23.9%) opposed samples from Mt. Pollino in the right quadrant to Peruvian soils and samples from Mt. Shennongjia. Dimension 2 (15.2%) especially separated individuals from Chile and Mt. Rungwe. The Wilks' test showed that the location of the sampling sites was the best variable discriminating the distribution of the individuals in the PCA.

Principal component analysis performed on the temperature ($RAN_{15}$, $RAN_{17}$) and pH (RIAN) indices derived from 3-OH FAs showed that most of the variance was carried by the first two axes of the PCA (Axis 1 = 56.09%; Axis 2 = 35.29%; Supp. Fig. 2). The first axis was highly correlated with the $RAN_{15}$ ($r = 0.87$) and $RAN_{17}$ ($r = 0.93$) as well as with MAAT ($r = -0.67$), while Axis 2 showed strong correlations with the RIAN ($r = 0.96$) and pH ($r = -0.61$). The PCA allowed visualizing relationships at the scale of the whole dataset, between MAAT and $RAN_{15}$ and $RAN_{17}$ ($r = -0.61$; $r = -0.64$ respectively) and between pH and RIAN ($r = -0.53$).

*3.3.2. brGDGT*

The first two axes of the brGDGT PCA explained 57.7% of the total variance in the dataset (Fig. 3b). Dimension 1 (42.6%) strongly discriminated soils from Mt. Majella and, to a lesser extent, Mt. Pollino, in the right quadrant from those from Mt. Shegyla, Peruvian Andes and Chilean Andes in the left quadrant. Mts Majella and Pollino were also discriminated negatively along dimension 2 (15.1%). Samples from Mts. Shennongjia and Lautaret-Galibier were distributed over the entire PCA. As for the 3-OH FAs, Wilks' test showed that the location of the sampling sites was the best variable discriminating the distribution of the brGDGTs in the PCA.

## 4. Discussion

### 4.1. 3-OH FA and brGDGT-derived proxies

Previous studies conducted on soils from individual altitudinal transects revealed (1) local linear relationships between MAAT/pH and 3-OH FA indices and (2) the potential for combined calibrations using simple linear regressions (Wang et al., 2016; Huguet et al., 2019; Véquaud et al., 2021). In the present study, the existence of linear relationships between 3-OH FA-derived indices and environmental variables was further investigated using an extended soil dataset and the corresponding results were compared with those derived from the brGDGTs, used as an established reference proxy.

*4.1.1. Relationships between pH and bacterial lipid-derived proxies*

The relationship between RIAN and pH was investigated along each of the altitudinal transects (Fig. 4a; Supp. Table 3). No significant linear relationship was obtained for the Peruvian Andes, Mts. Rungwe, Pollino and Majella (Huguet et al., 2019) and weak to moderate correlations were observed along Mts. Shegyla and Lautaret-Bauges ($R^2$ = 0.29-0.46; Supp. Table 3). In contrast, strong regressions between RIAN and pH were observed along Mt. Shennongjia ($R^2$ = 0.71) and in Chilean Andes ($R^2$ = 0.66). A weak linear relationship between RIAN and pH ($R^2$=0.34; RMSE = 0.99; $p = 7.39 \times 10^{-17}$) was also obtained when considering the 168 samples for the eight elevation transects altogether. Therefore, our results confirm the general influence of pH on the relative abundance of 3-OH FAs (Huguet et al., 2019) but suggest that strong linear correlations between RIAN and pH can only be obtained (i) at a local level and (ii) only for some of the sites.

As previously suggested (Huguet et al., 2019), the absence or weakness of linear correlations between RIAN and pH may be at least partly due to the small range of variation of pH (<2 units) along some mountains, such as Mts. Rungwe, Majella, and the Peruvian Andes (Fig. 4a; Table 1, Huguet et al., 2019). Transects for the Peruvian Andes and Mt. Majella were also characterized by the absence of relationships between pH and the brGDGT-derived CBT' index, supporting the hypothesis that narrow pH ranges limit the potential of obtaining linear relationships between indices based on bacterial lipids and pH. Nevertheless, the existence of a narrow pH range was not the only limiting factor in obtaining a strong linear regression between RIAN and pH. Indeed, MAAT rather than soil pH was the dominant driver of soil bacterial diversity and community composition for the Peruvian transect (determined using 16S rRNA sequencing (Nottingham et al., 2018) and phospholipid fatty acids (Whitaker et al., 2014)),

consistent with the weak correlation between soil pH and bacterial lipids. The weakness of the RIAN-pH relationship may also be partly due to the heterogeneity of soils encountered along a given altitudinal transect, representing specific microenvironments and to the large diversity of bacterial communities in soils from different elevations (Siles and Margesin, 2016). The distribution of 3-OH FAs varies greatly among Gram-negative bacterial species (Bhat and Carlson, 1992) which may account for the significant variability in RIAN values observed in soils from a given transect. Altogether, these results suggest that linear models are not the most suitable for establishing a global calibration between RIAN and pH in soils.

Concerning GDGTs, moderate to strong relationships between brGDGT-derived CBT' index and pH were observed along 5 of the 7 altitudinal transects investigated (Fig. 4b; Supp. Table 3). All the individual linear relationships between CBT' and pH, where present, had similar slopes and ordinates and share (for most of the samples) the same 95% confidence intervals ($p$-value <0.5). This resulted in a strong linear relationship between CBT' index and pH values for the dataset ($R^2 = 0.68$; RMSE = 0.71; $n = 140$), which is weaker than the global calibration ($R^2 = 0.85$; RMSE = 0.52; $n = 221$) proposed by De Jonge et al. (2014).

The discrepancy in relationships between temperature and brGDGTs and 3-OH FAs might partly be due to differences in the relative abundance of these lipids among bacterial communities. The brGDGTs are produced by a more restricted and less diverse number of bacterial species than 3-OH FAs, which are arguably biosynthesized by a large diversity of Gram-negative bacteria species (e.g. Wakeham et al., 2003, Zelles et al., 1995; Zelles, 1999). So far, only bacteria from the *Acidobacteria* phylum were identified as putative brGDGT producers in soils (Sinninghe Damsté et al., 2018). The hypothetical lower diversity of brGDGT producers, in contrast with 3-OH FAs, might explain the more homogenous response and lower scatter of the relationships between pH and CBT' index. Moreover, the CBT' index is a ratio based on a restricted number of compounds, representing the direct dependence of the degree of cyclisation of bacterial GDGTs on pH. Conversely, the RIAN index is calculated from the relative abundances of all the individual 3-OH FAs between $C_{10}$ and $C_{18}$ (Wang et al., 2016). It cannot be ruled out that some of the compounds used to calculate the RIAN index are preferentially synthesized, as part of the homeoviscous mechanism, in response to environmental variables other than pH. This calls for a better understanding of the ecology of 3-OH FA-producing bacteria and their adaptation mechanisms.

 *4.1.2 Relationships between MAAT and bacterial lipid-derived proxies*

$RAN_{15}$ was previously shown to be correlated with MAAT along Mts. Rungwe,
Majella and Shennongjia (Wang et al., 2016; Huguet et al., 2019). Moderate to strong linear
correlations ($R^2$ =0.49-0.79) between $RAN_{15}$ and MAAT were also observed along most of the
individual transects investigated (Fig. 5a; Supp. Table 3, except along the Chilean and Lautaret-
Bauges transects. The individual correlations do not share the same 95% confidence intervals
and even when some of them present similar slopes, the regression lines display significantly
different intercepts (*p*-value > 0.05) (Fig. 5a).  This supports the hypothesis of a site-dependent
effect of the linear $RAN_{15}$-MAAT relationship previously made by Huguet et al. (2019).
Similarly, to $RAN_{15}$, $RAN_{17}$ was moderately to strongly correlated ($R^2$ =0.53-0.81)
with MAAT along 5 out of 8 individual transects (Fig. 5b; Supp. Table 3). The small range of
variation in MAAT along the Chilean transect (6.0-9.2 °C) (Table 1), associated with that of
the $RAN_{15}$ /$RAN_{17}$, could explain the lack of a linear relationship between the MAAT and these
indices. As for the French Alps (Mts Lautaret-Bauges), the influence of local environmental
parameters (pH and to a lesser extent soil moisture and grain size, related to vegetation and soil
types, or thermal regimes associated with the snow cover) on 3-OH FA distribution was shown
to be predominant over that of MAAT (Véquaud et al., 2021). In contrast with $RAN_{15}$, the linear
regressions between $RAN_{17}$ and MAAT along Mts. Shegyla, Shennongjia, Rungwe and the
Peruvian Andes transects share confidence intervals at 95% and have similar slope and intercept
values (*p*-value <0.05; Fig. 5b; Supp. Table 3), suggesting that $RAN_{17}$ could be a more effective
global proxy for MAAT reconstructions than $RAN_{15}$.
In order to test the hypothesis that $RAN_{17}$, rather than $RAN_{15}$, is a more effective
global proxy for MAAT, the global calibrations between $RAN_{15}$/$RAN_{17}$ and MAAT based on
the entire soil dataset ($n$ = 168) were compared. The two linear regressions had similar moderate
determination coefficients ($R^2$ = 0.37 and 0.41 for $RAN_{15}$ and $RAN_{17}$, respectively) and similar
high RMSE (RMSE = 5.46°C and 5.28°C for $RAN_{15}$ and $RAN_{17}$, respectively; Supp. Table 3).
For all transects (except for the Mt Majella $RAN_{17}$/MAAT relationship), the individual local
regressions between $RAN_{15}$/$RAN_{17}$ and MAAT outperformed the proposed global linear
calibrations in terms of determination coefficients (0.49-0.81) and RMSE (1.98-3.57 °C; Supp.
Table 3), suggesting that local rather than global linear transfer functions based on $RAN_{15}$ or
$RAN_{17}$ may be more appropriate for paleotemperature reconstructions in soils.
The difficulties in establishing global linear $RAN_{15}$/$RAN_{17}$-MAAT calibrations may
partly be due to the fact that microbial diversity, especially for 3-OH FA-producing Gram-
negative bacteria (Margesin et al., 2009; Siles and Margesin, 2016), can vary greatly from one
soil to another, resulting in variation of the $RAN_{15}/RAN_{17}$ indices, as also assumed for the
RIAN. The strong regional dependence of the 3-OH FA distribution may thus explain the weak
correlation between 3-OH FA-derived indices ($RAN_{15}$, $RAN_{17}$ and RIAN) and environmental
variables (MAAT/pH) at a global level. This regional dependency was further supported by the
PCA of the relative abundance of 3-OH FAs across the global dataset, which showed that the
individuals were grouped based on the sampling location (Fig. 3a).

In addition to 3-OH FAs, the relationships between brGDGT distribution and MAAT

were investigated along the seven transects for which the 5- and 6-methyl brGDGT isomers
were separated (Mts Shegyla, Pollino Majella, Lautaret-Bauges, Shennongjia, Peruvian Andes
and Chilean Andes). These individual transects showed moderate to strong relationships
between MAAT and MBT'$_{5Me}$ ($R^2$ 0.35-0.89; Fig. 6 and Supp. Table 3), with similar slopes and
ordinates (except for the Peruvian Andes) and shared 95% confidence intervals for most of the
samples. A distinct relationship between MBT'$_{5Me}$ and MAAT was observed along the Peruvian
Andes and Mt Majella transects (Fig. 6a), as also observed for the RIAN and $RAN_{15}$ indices
(Figs 4a and 5a). The singularity of the Peruvian soils is also visible on the PCA performed on
the brGDGT distribution (Fig. 3b), where the samples from this region are pooled separately
from the rest of the dataset. This specific trend is difficult to explain, even though the Peruvian
Andes are subjected to warmer climatic conditions (Table 1) than the other temperate transects,
which may in turn affect the nature of the microbial communities encountered in the soils and
the bacteria lipid distribution (Siles and Margesin, 2016; Hofmann et al., 2016; De Jonge et al.,

2019).

A moderate linear relationship between MAAT and MBT'$_{5Me}$ (MAAT = 24.5 × MBT'$_{5Me}$

-4.78; $R^2$ = 0.57, RMSE = 3.39 °C, $n$ = 140; Supp. Table 3) was observed after combining the
data for the seven aforementioned altitudinal transects. This global relationship follows a
similar trend as the calibration proposed by De Jonge et al. 2014 (MAAT = 31.45 × MBT'$_{5Me}$ -
8.57) and is more robust and accurate than those obtained between the $RAN_{15}/RAN_{17}$ and
MAAT (Supp. Table 3). This confirms that the MBT'$_{5Me}$ index can be applied at a global scale
using a simple linear regression model as previously shown (De Jonge et al., 2014; Naafs et al.,
2017), in contrast with the $RAN_{15}$ and $RAN_{17}$ proxies, for which only strong local calibrations
with MAAT were found.

As a similar conclusion was obtained for the RIAN-pH proxy, it appears necessary to use

more complex models to develop global calibrations between 3-OH FA-derived proxies and
MAAT/pH. This novel method allows taking into account the complexity and specificity of
each environmental site.

## 4.2. Development of new models for the reconstruction of MAAT and pH from 3-OH FA

Several complementary methods were recently used to derive calibrations with environmental parameters from organic proxies. Most calibrations between lipid distribution and environmental variables were based on simple linear regression models, most often the ordinary least square regression (e.g. for brGDGTs: De Jonge et al., 2014; Wang et al., 2016), as it is simple and easy to implement and understand. Other linear models, such as Deming regression (Naafs et al., 2017) or Bayesian regression (Tierney and Tingley, 2014; Dearing Crampton-Flood et al., 2020) were also used. Nevertheless, these single linear regression methods rely on a given index (e.g. MBT'$_{5Me}$ or CBT' for brGDGTs) which is correlated with environmental parameters. This represents a limitation, as the relative distribution of bacterial lipids can be concomitantly influenced by several environmental parameters (e.g. Véquaud et al., 2021) and can also depend on the diversity of the bacteria producing these compounds (Parker et al., 1982; Bhat and Carlson, 1992; Zelles, 1999). In contrast, using bacterial lipid relative abundances rather than a single index in the relationships with environmental variables appears less restrictive, and more representative of the environmental complexity. Other models can be used in this way, such as those based on multiple regressions (e.g. Peterse et al., 2012; De Jonge et al., 2014; Russell et al., 2018), describing the relationships between one or several explained variables (e.g. bacterial lipid abundances) and one or several explanatory variables (e.g. MAAT, pH). Multiple regressions can reveal the presence of linear relationships among several known variables but cannot take into account non-linear influences, which may occur in complex environmental settings. This limitation, common to all linear models, can be overcome using non-parametric methods such as some of the machine-learning algorithms (e.g. nearest neighbours or random forest; Dunkley Jones et al., 2020). The reliability of the latter models lies in the fact that they are non-linear, which helps capturing the intrinsic complexity of the environmental setting, and that they avoid the regression dilution phenomenon observed in most linear models. Moreover, their robustness is improved by the fact that they are built on a randomly defined proportion of the total dataset and then tested on the rest of the dataset, considered as independent. Last, these machine-learning algorithms are flexible and are continuously evolving when adding new samples.

As shown in section 4.1., robust global calibrations between 3-OH FA-derived indices (RIAN, RAN$_{15}$ and RAN$_{17}$) and MAAT/pH could not be established using a simple linear regression model, contrary to what was observed with brGDGT-derived indices. Therefore,

three different independent and complementary models were tested to potentially establish stronger statistical relationships between 3-OH FA distributions and pH/MAAT at the global level : (i) a parametric model – multiple linear regression; (ii) two non-parametric models – random forest (e.g. Ho, 1995; Denisko and Hoffman, 2018) and k-NN algorithms (e.g. Gangopadhyay et al., 2009). As discussed above, the multiple linear regression model allows the determination of linear relationships between MAAT/pH and the individual relative abundances of 3-OH FAs, instead of indices derived from the latter. As for the two non-parametric models, they present among other things the advantage of taking into account non-linear environmental influences.

The three models, based on a supervised machine learning approach, were applied to the total soil dataset ($n$=168). All the 3-OH FA homologues of Gram-negative bacterial origin (i.e. with chain lengths between $C_{10}$ and $C_{18;}$ Wilkinson et al., 1988) were included in the models whatever their abundance to keep the maximum variability and take into account the specificity and complexity of each altitudinal transect. Indeed, the nature of the individual 3-OH FAs whose fractional abundance is mainly influenced by MAAT/pH may be site-dependent, as previously observed (Véquaud et al., 2021). The performances of these three models were compared with those of the linear calibrations between 3-OH FA-derived indices (RAN$_{15}$, RAN$_{17}$, RIAN) and MAAT/pH (Table 2).

### 4.2.1. Temperature calibrations

The multiple linear regression model yielded a strong relationship between 3-OH FA relative abundances and MAAT (Fig. 7a; Eq.6):

$MAAT (°C) = -59.02 \times [nC_{10}] + 102.1 \times [iC_{11}] + 2628.49 \times [aC_{11}] - 165.58 \times [nC_{11}] - 79.799 \times [nC_{12}] + 89.93 \times [iC_{13}] + 205.06 \times [aC_{13}] - 136.25 \times [nC_{13}] - 309.71 \times [iC_{14}] - 43.16 \times [nC_{14}] - 9.27 \times [iC_{15}] - 308.53 \times [aC_{15}] + 66.06 \times [nC_{15}] - 60.57 \times [iC_{16}] + 15.53 \times [nC_{16}] + 13.52 \times [iC_{17}] - 228.76 \times [aC_{17}] - 91.12 \times [nC_{17}] + 43.71$
$(n = 168; R^2 = 0.79; RMSE = 3.0 °C)$ (6)

This model, which takes into account the Gram-negative bacterial 3-OH FAs ($C_{10}$-$C_{18}$; Wilkinson et al., 1988), presents a higher strength than the global linear relationships between 3-OH FA derived indices and MAAT ($R^2$=0.37 and 0.41; RMSE =5.5°C and 5.3°C for RAN$_{15}$ and RAN$_{17}$, respectively; Table 2). The multiple linear regression also improves the accuracy and robustness of MAAT prediction in comparison with single linear relationships, with lower RMSE (3.0 °C), variance of the residuals (9.2 °C; Fig. 7d) and mean absolute error (MAE; 2.3

°C) than with the $RAN_{15}$ and $RAN_{17}$ calibrations (RMSE of 5.5 and 5.3 °C; variance of 29.8
and 27.9 °C; MAE of 4.0 and 3.9 °C for $RAN_{15}$ and $RAN_{17}$, respectively; Table 2).
Similarly to the multiple linear regression model (Fig. 7a), the random forest (Fig. 7b)
and k-NN (Fig. 7c) calibrations are characterized by strong determination coefficients ($R^2$ 0.83
and 0.77, respectively). The variance in residuals, MAE and RMSE of the random forest
calibration are slightly lower than those of the multiple linear regression and k-NN models
(Table 2). An advantage of the random forest algorithm lies in the fact that the weight of the
different variables used to define the model can be quantified using the permutation importance
method (Breiman, 2001). The $a$-$C_{15}$, $i$-$C_{14}$, $a$-$C_{17}$, $n$-$C_{12}$, $n$-$C_{15}$, and to a lesser extent $n$-$C_{17}$, $n$-
$C_{16}$ and $i$-$C_{13}$ 3-OH FAs were observed to be the homologues predominantly used by the model
to estimate MAAT values (Fig. 9a). They include all the 3-OH FAs involved in the calculation
of the $RAN_{15}$ and $RAN_{17}$ indices, especially the $a$-$C_{15}$ homologue. This may explain why linear
relationships between the $RAN_{15}$/$RAN_{17}$, and MAAT could be established along some, but not
all, of the altitudinal transects investigated until now (Wang et al., 2016; Huguet al., 2019;
Véquaud et al., 2021; this study). Nevertheless, other individual 3-OH FAs than those appearing
in the calculation of the $RAN_{15}$ and $RAN_{17}$ have also a major weight in the random forest model
and seem to be influenced by temperature changes, explaining the moderate determination
coefficients of the global $RAN_{15}$/$RAN_{17}$-MAAT linear relationships observed in this study.
On the whole, the strength and accuracy of the multiple linear regression, k-NN and
random forest models are much higher than those based on the $RAN_{15}$ and $RAN_{17}$ indices
(Table 2). This is likely related to the fact that the three aforementioned models integrate the
whole suite of 3-OH FAs homologues ($C_{10}$ to $C_{18}$) and thus better capture the complexity of the
response of soil Gram-negative bacteria and their lipid distribution to temperature changes than
the $RAN_{15}$ and $RAN_{17}$ indices. They also present the advantage of increasing the range of
temperature which may be predicted by more than 4 °C in comparison with the $RAN_{15}$ and
$RAN_{17}$ calibrations (Table 2). Indeed, even though the lower limit of MAAT estimates for the
three models tested in the present study is slightly higher than those based on the $RAN_{15}$ and
$RAN_{17}$ indices, the upper limit of the MAAT which can be estimated using the multiple linear
regression, random forest and k-NN models is substantially higher (ca. 25 °C) than that based
on the $RAN_{15}$ or $RAN_{17}$ indices (ca. 17 °C; Table 2).
The three proposed models show the potential of 3-OH FAs as MAAT proxies at the
global level, which was not visible using $RAN_{15}$ and $RAN_{17}$ indices. The non-parametric
models (random forest and k-NN) may benefit from the fact that they take into account the
complex, non-linear relationships between environmental parameters and bacterial lipid
abundance. This is highlighted when comparing the independent variations of the individual 3-
OH FA relative abundances with estimated MAAT for the three proposed models, with non-
linear trends for the k-NN and random forest models, in contrast with the multiple linear
regression (Supp. Fig. 2).

*4.2.2. pH calibrations*
A robust linear relationship between the RIAN and pH could not be obtained from the
whole soil dataset (Fig. 4a; Table 2). In contrast, the multiple regression model provided a
strong correlation between the 3-OH FA fractional abundances and pH (Fig. 8a; Eq. 7):
*pH = -1.45 × [nC$_{10}$] − 31.70 × [iC$_{11}$] − 162.09 × [aC$_{11}$] − 53.22 × [nC$_{11}$] − 6.21× [nC$_{12}$] +*
*56.24× [iC$_{13}$] − 2.02 × [aC$_{13}$] + 15.10 × [nC$_{13}$] + 23.99 × [iC$_{14}$] − 4.54× [nC$_{14}$] − 13.79 ×*
*[iC$_{15}$] − 15.74 × [aC$_{15}$] + 1.93 × [nC$_{15}$] − 46.29 × [iC$_{16}$] − 3.20 × [nC$_{16}$] − 1.80 × [iC$_{17}$] −*
*8.90 × [aC$_{17}$] + 11.46 ×[nC$_{17}$] − 3.63×[nC$_{18}$] + 7.84   (n = 168; R$^2$ = 0.64; RMSE = 0.8)   (7)*
The random forest (Fig. 8b) and k-NN pH models (Fig. 8c) appeared to be slightly more
robust and accurate than the multiple linear regression (Fig. 8a), as the former two models
presented slightly higher determination coefficients (R$^2$ = 0.68 and 0.70 for k-NN and random
forest, respectively) and slightly lower RMSE (0.7), variance in residuals (0.5) and MAE (0.5)
than the multiple linear regression (Table 2).
As for the MAAT random forest model, the weight of the individual 3-OH FAs in the pH
random forest calibration was determined (Fig. 9b). Three homologues − *i*-C$_{13}$, *n*-C$_{15}$, *i*-C$_{16}$ −
had a larger weight in the global pH model than the others (Fig. 9b). This is consistent with a
detailed study of 3-OH FA distribution in soils from the French Alps (Véquaud et al., 2021),
where the *i*-C$_{13}$ and *i*-C$_{16}$ 3-OH FAs were observed to be predominantly influenced by pH.
Nevertheless, in addition to the three aforementioned homologues, most of the C$_{10}$ to C$_{18}$ 3-OH
FAs have a non-negligible influence in the random forest pH model, except the *a*-C$_{15}$ and *i*-C$_{14}$
compounds (Fig. 9b). This is in line with the definition of the 3-OH FA-based pH index (RIAN)
defined by Wang et al. (2016) which includes the whole suite of 3-OH FAs. These results
suggest that soil Gram-negative bacteria may respond to pH variations by modifying the whole
distribution of associated 3-OH FAs (C$_{10}$-C$_{18}$). This would need to be further confirmed by e.g.
investigating the influence of pH variations on pure strains of Gram-negative bacteria isolated
from soils.
In any case, in contrast with the RIAN index, the multiple linear regression, k-NN and
random forest models provided strong global calibrations with pH (Fig. 8), as robust as the
global CBT'-pH relationship (Fig. 4b). The three proposed models also increase the range of

pH which can be estimated (~ 4 pH units) in comparison with the RIAN global calibration (~ 3 pH units), further strengthening the potential of these models for soil pH reconstruction. As MAAT models, the independent variations of the individual 3-OH FA relative abundances with estimated pH highlight non-linear trends for the k-NN and random forest models, in contrast with the multiple linear regression (Supp. Fig. 3), which might favor the use of the two non-parametric models in order to take into account such non-linear influences. The machine-learning MAAT and pH models proposed in this paper are flexible and could be further improved by increasing the number of soil samples analyzed and the representativeness of the different MAAT and pH values within the dataset.

**4.3. Paleoclimate application of the new 3-OH FA/MAAT models**

The multiple regression, random forest and k-NN models developed for MAAT reconstruction using 3-OH FAs were similar in terms of robustness and precision (Figs. 7a, b, c; Table 2). The performance and validity of these global terrestrial calibrations for paleotemperature reconstructions were thus tested and compared with the MAAT record from a Chinese speleothem (HS4 stalagmite) covering the last 9,000 years BP (Wang et al., 2018). This terrestrial archive was the object of previous paleostudies, thus providing a context for the interpretation of the MAAT data and, to the best of our knowledge, represents the only published application of 3-OH FAs as a paleotemperature proxy in terrestrial settings (Wang et al., 2018). The local comparison of 3-OH FA distributions in the overlying soils and stalagmites and the analyses of bacterial diversity and transport pathways suggested that the 3-OH FAs in the HS4 speleothem were mainly soil-derived (Wang et al., 2018), supporting the application of soil calibrations for MAAT reconstruction from this archive, although not being a paleosoil itself. The first paleoapplication of 3-OH FAs (Wang et al., 2018) on this speleothem relied on a local calibration between the $RAN_{15}$ index and MAAT proposed by Wang et al. (2016) using soils from Mt. Shennogjia. The MAAT estimates derived from our global soil calibrations were compared with those obtained from this local soil calibration (Wang et al., 2016).

*4.3.1 Comparison of the multiple linear regression, k-NN and random forest global MAAT calibrations*

The multiple regression model (Eq. 6; Fig. 7a) yielded MAAT estimates ranging between -35 and 22.8 °C over the last 9,000 years (Supp. Fig. 4). The temperature minimum (-

35°C) observed at 560 yrs BP can be considered as an outlier, with a significantly lower MAAT
estimate than those provided by the other samples. After having ignored this apparent outlier,
the MAAT range over the last 9,000 years was comprised between 3.2°C and 22.8°C, with
temperature shifts of up to 15 °C within very short periods of time. The observed range of
MAAT and large variations in temperature over such short periods appear far too excessive, as
the expected amplitude of MAAT during the Holocene is expected to be up to ca. 2-3 °C (Liu
et al., 2014). This highly questions the reliability of the multiple linear regression model for
MAAT reconstruction from this archive.
MAAT estimates derived from the k-NN calibration ranged between 6.5 and 19.7 °C
over the last 9,000 years (Supp. Fig. 4). Abrupt shifts in MAAT of more than 10 °C were
observed between 2,000 and 4,000 yrs BP. Such variations, higher than the RMSE of the
calibration, appear excessive for the Holocene period, as previously discussed for the multiple
regression model. The bias in MAAT estimates may be due to the intrinsic definition of the k-
NN model, which is better suited for uniformly distributed datasets. This is not the case here,
as the individual transects heterogeneously cover a wide range of temperatures. The application
of a global calibration at the local scale – that of the HS4 stalagmite – using the k-NN method
and based on the similarities among samples, thus does not appear appropriate. Such a
calibration might be improved by extending the dataset with samples more equally distributed
across a wider range of global climatic gradients.
Finally, the random forest model yielded MAAT estimates between 10.6 and 19.3°C,
i.e. a smaller estimation range than the k-NN algorithm and multiple regression model (Supp.
Fig. 4). The amplitude of the shifts observed between 2,000 and 4,000 yrs BP was ca. 4°C,
which is climatically more consistent than the variations obtained with the k-NN method and
multiple regression model, even though these large variations in MAAT over such short periods
of time still appear too excessive. Furthermore, the application of the global random forest
calibration roughly provided similar temperature trends as those derived from the local RAN$_{15}$
calibration by Wang et al. (2018; Fig. 10), despite some largest oscillations for the global model.
These results suggest that the random forest calibration is more reliable than the multiple
regression and k-NN ones. This can be explained by the intrinsic definition of the random forest
algorithm, which averages the results of several independent models (so-called decision trees),
thus reducing the variance and thus the forecast error on the final model. This is also in line
with the slightly higher accuracy of the random forest calibration compared with the other two
models (Table 2), as previously discussed. In contrast, the multiple regression calibration was
the less performant of the three models on the investigated archive. This may be related to its
parametric nature and the fact that it does not take into account the natural non-linear variations
on 3-OH FA fractional abundances highlighted by the random forest and k-NN models (Supp.
Figs. 2 and 3).

In conclusion, the three models proposed in this study, especially the random forest,

have potential for MAAT reconstruction, even though the application to a well-known
paleoclimate archive showed their limitations. This highlights the importance of testing new
calibrations on well-characterized archives to investigate their reliability.

*4.3.2. Comparison of the global random forest and local RAN$_{15}$ calibrations for MAAT*

*reconstruction*

The random forest model was observed to be the most reliable of the three proposed

global MAAT calibrations (Fig. 7). To go further, we compared the temperature record derived
from our global random forest calibration with that derived from the local MAAT/RAN$_{15}$
transfer function proposed by Wang et al. (2016; Fig. 10). The application of the local RAN$_{15}$
calibration to the HS4 stalagmite yielded an average MAAT of ca. 18.4 °C over the most recent
part of the record (last 800 yrs; Fig. 10), consistent with the MAAT of 18 °C recorded *in situ*
by a temperature logger (Hu et al., 2008; Wang et al., 2018). In contrast, absolute MAAT
estimates derived from the random forest model were on average 14.2 °C over the last 800 yrs
and were generally lower than those obtained from the local RAN$_{15}$ calibration over the whole
record. Altogether, these results suggest that the random forest model tends to underestimate
absolute MAAT, in contrast with the RAN$_{15}$ calibration proposed by Wang et al. (2016). This
discrepancy may be due the fact that the calibration proposed in the present study is based on a
global dataset, with samples subject to a large variety of environmental and climatic conditions,
whereas the RAN$_{15}$-MAAT transfer function by Wang et al. (2016) was constructed using soil
samples from a regional altitudinal transect, located at only 120 km distance from the stalagmite
site (Wang et al., 2018).

Even though the local calibration by Wang et al. (2016) provides more accurate

absolute MAAT values than the present global random forest model, as it could be expected,
both calibrations roughly generate similar qualitative MAAT trends over time. A regular slight
decrease in temperature of ca. 1 °C was observed between 9,000 and ca. 1,000 yrs BP based on
the local RAN$_{15}$ calibration (Fig. 10a; Wang et al., 2018). This general decreasing trend was
also visible when using the random forest model, but with larger oscillations and mainly
between 9,000 and 4,000 yrs BP, in agreement with the general trend recorded by the $\partial^{18}O$
record (mixture of temperature and hydrological signals, Wang et al., 2018) of the HS4
stalagmite (Fig. 10b,c; Hu et al., 2008). In addition, both the global random forest, local RAN$_{15}$
calibrations and the $\partial^{18}O$ record allowed the identification of several climatic events in the
Northern hemisphere, in agreement with the reconstructed total solar irradiance (TSI,
Steinhilber et al., 2009, Fig. 10d). Thus, both models highlighted, with slightly different
amplitudes, the Medieval Warm Period (800-1000 years BP) and Little Ice Age (LIA; 200-500
years BP) periods (Mann et al., 2008; Ljungqvist, 2010; Wang et al., 2018). The LIA event is
particularly well represented by the global random forest calibration, in line with the decrease
in the TSI (Fig. 10b,d) associated with a relative increase in the $\partial^{18}O$ of HS4 carbonates
(dry/cool event, Wang et al., 2018). Before the MWP, the global random forest calibration
shows slight oscillations, which can be assumed to be representative of TSI variations between
500 and 1,300 yrs BP. Similarly, an important cooling event, well correlated with a significant
decrease in the TSI (Fig. 10a, b, d), was recorded by the two calibrations at 1300 yrs BP.

The global random forest calibration also highlighted two cooling events, poorly

represented by the local RAN$_{15}$ calibration: one at ca. 4,200 yrs BP ago and, to a lesser extent,
another one between 2,800 and 3,000 yrs BP (Bond et al., 2001; Mayewski et al., 2004).  The
event at 4,200 yrs BP is consistent with the $\partial^{18}O$  and solar irradiance records and is referenced
in the literature as the "4.2 kiloyear event" (deMenocal, 2001). This intense drought event was
suggested to have had a major impact on different civilizations (collapses, migrations;
(Gibbons, 1993; Staubwasser et al., 2003; Li et al., 2018; Bini et al., 2019). Thus, in some parts
of China, the production of rice fields sharply decreased during this period, leading to a decrease
in population (Gao et al., 2007).

Both calibrations additionally shows a cooling period between 4,000 yrs and 3,200 yrs

BP, more pronounced based on the global random forest model, followed by another cooling
between 3,200 years BP and 3,000 yrs BP. This cooling period is consistent with the trends
derived from $\partial^{18}O$ and solar irradiance records. It culminates with a cold episode at 3000 yrs
BP, also known as Late Bronze Age Collapse (Kaniewski et al., 2013). Indeed, this cold
episode, combined with droughts, may have led to a decrease in agricultural production in
China, contributing to the degradation of trade routes and ultimately to the collapse of Bronze
Age civilizations (Weiss, 1982; Knapp and Manning, 2016). Last, the global random forest
calibration also highlights two additional cold events, between 5,600 and 5,900 yrs BP, as well
as around 7,100 yrs BP, corresponding to solar irradiance minima (Bond et al., 2001; Mayewski
et al., 2004) and which are not as clearly visible with the local RAN$_{15}$ calibration by Wang et
al. (2016).
The first application of the random forest calibration to a natural archive shows the
potential of 3-OH FAs as paleotemperature proxies at a global scale, as known and documented
climatic events were recorded, with a similar RMSE (2.8 °C; Table 2) as that of the local
calibration by Wang et al. (2.6 °C; 2016). In summary, we demonstrate that 3-OH FAs are
promising and effective temperature proxies for terrestrial settings, complementary to, and
independent of, the brGDGTs (De Jonge et al., 2014; Naafs et al., 2017; Dearing Crampton-
Flood et al., 2020), and also highlight the usefulness of non-parametric models using machine
learning, especially the random forest algorithm, to establish global MAAT calibrations. We
expect that analyses of 3-OH FAs in a larger number of globally distributed soils will further
improve the accuracy and robustness of the global random forest calibration for
paleotemperature reconstruction. Additional paleoapplications are also required to further test
and validate the applicability of the global MAAT and pH calibrations based on 3-OH FAs
presented in this study.

**5. Conclusions**
3-OH FAs have been recently proposed as environmental proxies in terrestrial settings,
based on local studies. This study investigated for the first time the applicability of these
compounds as MAAT and pH proxies at the global scale using an extended soil dataset across
a series of globally distributed elevation transects ($n = 168$). Strong linear relationships between
3-OH FA-derived indices (RAN$_{15}$, RAN$_{17}$ and RIAN) and MAAT/pH could only be obtained
locally, for some individual transects, suggesting that these indices cannot be used as
paleoproxies at the global scale through this kind of model. Other algorithms (multiple linear
regression, k-NN and random forest models) were tested and, in contrast with simple linear
regressions, provided strong global correlations between MAAT/pH and 3-OH FA relative
abundances. The applicability of these three models for paleotemperature reconstruction was
tested and compared with the MAAT record from the unique available record: a Chinese
speleothem. The calibration based on the random forest model appeared to be the most robust
and showed similar trends to previous reconstructions and known Holocene climate variations.
Furthermore, the global random forest model highlighted documented climatic events poorly
represented by the local RAN$_{15}$ calibration. This new global model is promising for
paleotemperature reconstructions in terrestrial settings and could be further improved by
analyzing 3-OH FAs in a larger number of globally distributed soils. This study demonstrates
the major potential of 3-OH FAs as MAAT/pH proxies in terrestrial environments through the
different models presented and their application for paleoreconstruction.

**Data availability.** All data are available in the Supplementary tables.

**Author contributions.** P.V. performed the lipid and statistical analyses and wrote a first draft
of the paper., A.H. and S.D. supervised the work of P.V. and corrected the first draft, P.V. and
A.T. developed the different models, G.B., A.N., W.P.S., N.S., J.P.W. and S.C. provided
samples and/or associated data, and all the co-authors reviewed and commented on the paper.
**Competing interests.** The authors declare that they have no conflict of interest.

**Acknowledgments.** We thank Sorbonne Université for a PhD scholarship to P.V. and the Labex
MATISSE (Sorbonne Université) for financial support. The EC2CO program (CNRS/INSU –
BIOHEFECT/MICROBIEN) is thanked for funding of the SHAPE project. A.H. and S.C. are
grateful for funding of the ECOS SUD/ ECOS ANID #C19U01 project. We are grateful to
Jérôme Poulenard for discussions on soil characteristics, and for comments on the manuscript.
We thank Dr. Juntao Wang and Prof. Jinzheng He for having provided soils from Mt. Shegyla.
We thank the Peruvian program led by NS, including CONCYTEC/FONDECYT through
contract 116-2016. We thank the associate editor Dr. van der Meer and the reviewers for their
comments which helped in improving the manuscript.

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

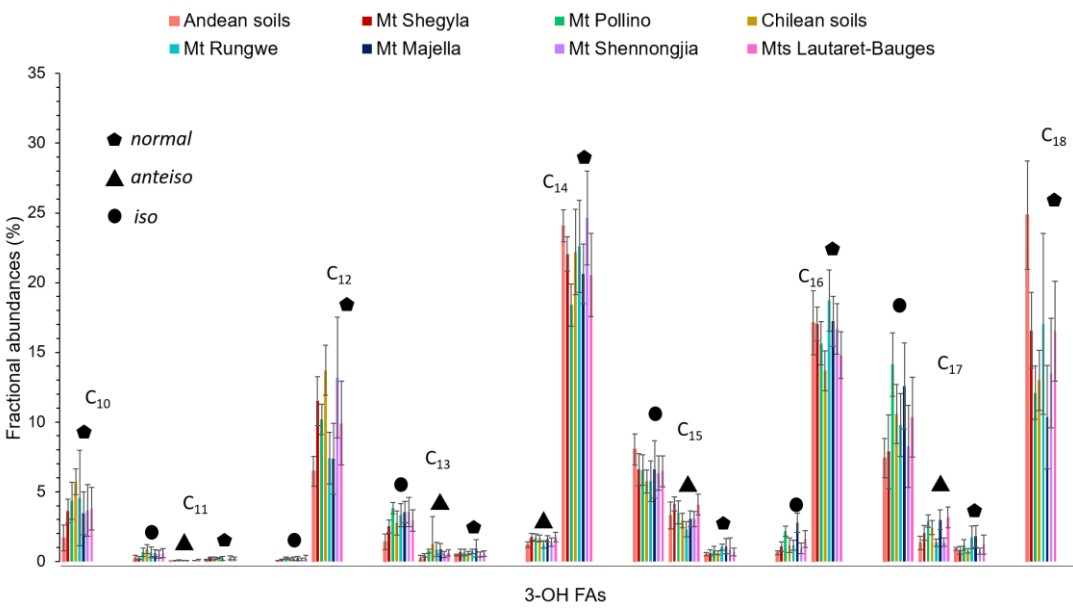

**Figure 1.** Average distribution of 3-OH FAs along the 8 altitudinal transects investigated in this study. Data from Mts. Majella and Rungwe were taken from Huguet et al. (2019). Data from Mt. Shennongjia were taken from Wang et al. (2016). Data from Mts. Lautaret-Galibier were taken from Véquaud et al. (2021).

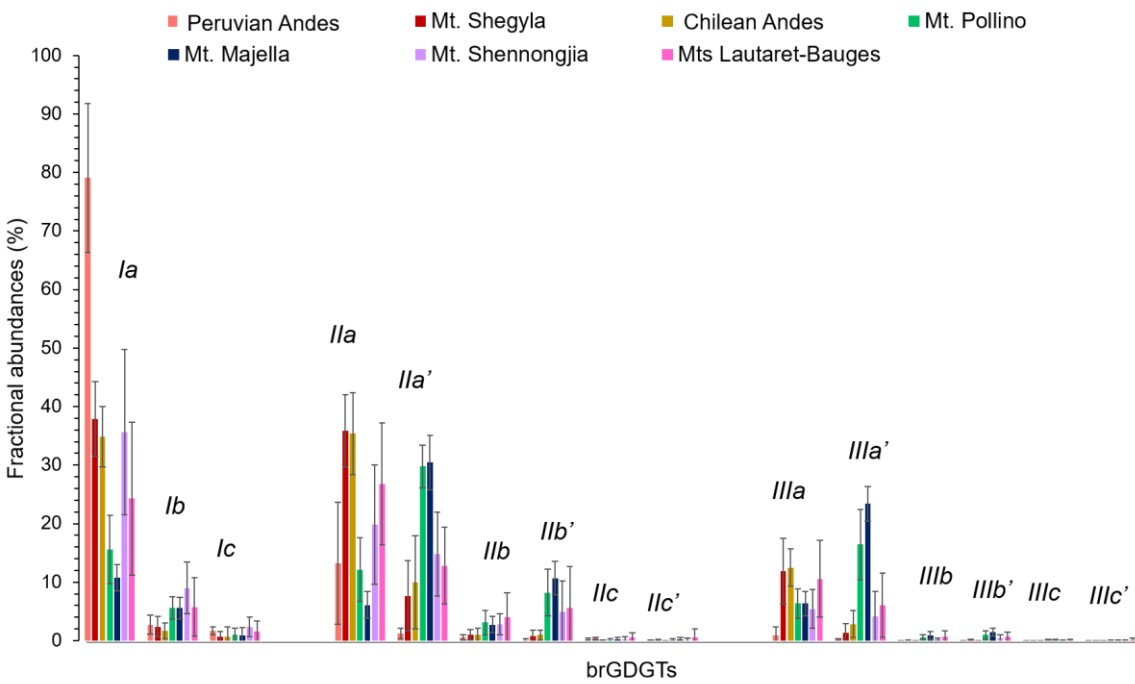

**Figure 2.** Average distribution of 5- and 6-methyl brGDGTs, along Mts. Shegyla, Pollino Majella, Lautaret-Bauges, Peruvian Andes and Chilean Andes. Data from Mt. Majella were taken from Huguet et al. (2019). Data from Mt. Shennongjia were taken from Yang et al. (2015). Data from Mts. Lautaret-Galibier were taken from Véquaud et al. (2021).

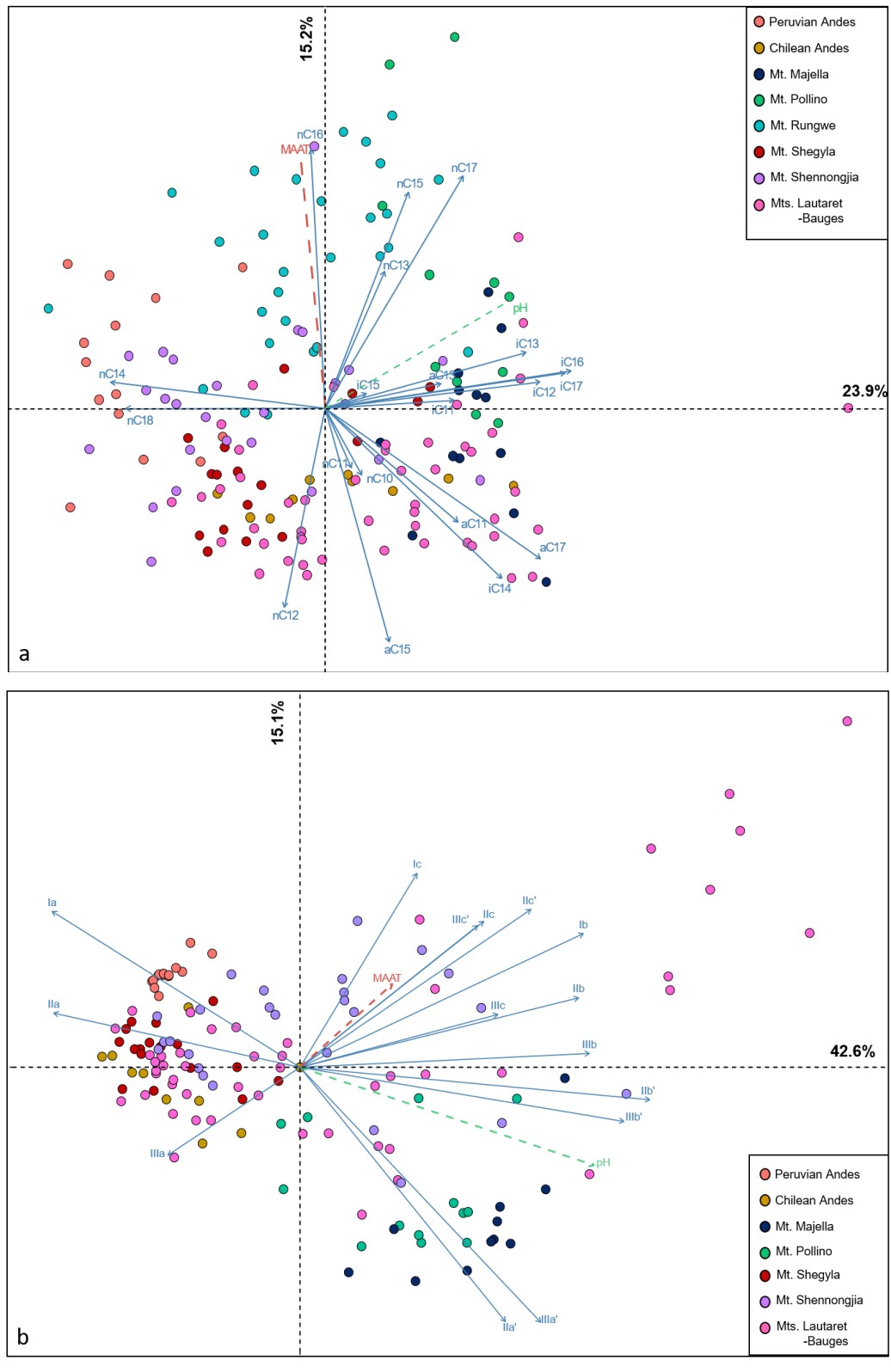

**Figure 3.** PCA biplot of (a) 3-OH FA fractional abundances in soil samples from the 8 altitudinal transects and (b) brGDGT fractional abundances in soil samples from 7 of the 8 altitudinal transects. BrGDGT data from Mt. Rungwe, for which 5- and 6-methyl isomers were not separated, were not included in the PCA.

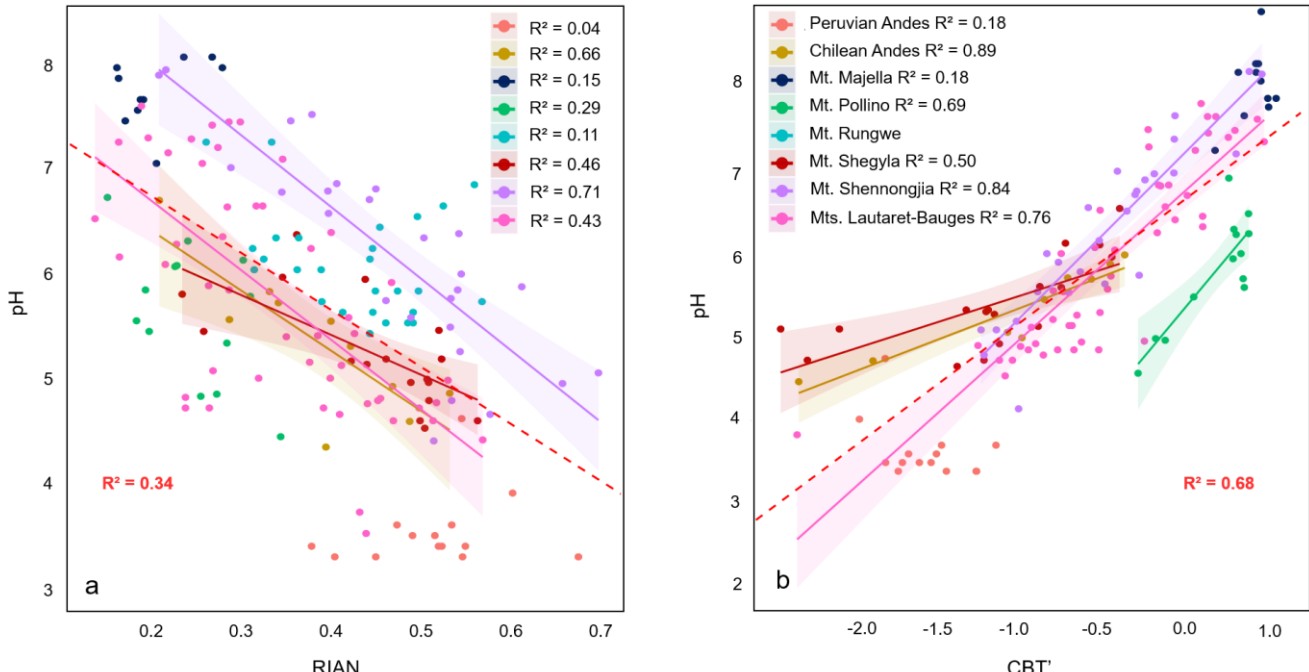

**Figure 4.** Linear regressions between (a) pH and RIAN and (b) pH and CBT' along the 8 altitudinal transects investigated. Dotted lines represent the 95% confidence interval for each regression and colored areas represent the 95% confidence interval for each regression. Data for Mts. Majella and Rungwe were taken from Huguet et al. (2019). Data from Mt. Shennongjia were taken from Yang et al. (2015) and Wang et al. (2016). Data from Mts. Lautaret-Galibier were taken from Véquaud et al. (2021). Only significant regressions (p < 0.05) are shown.

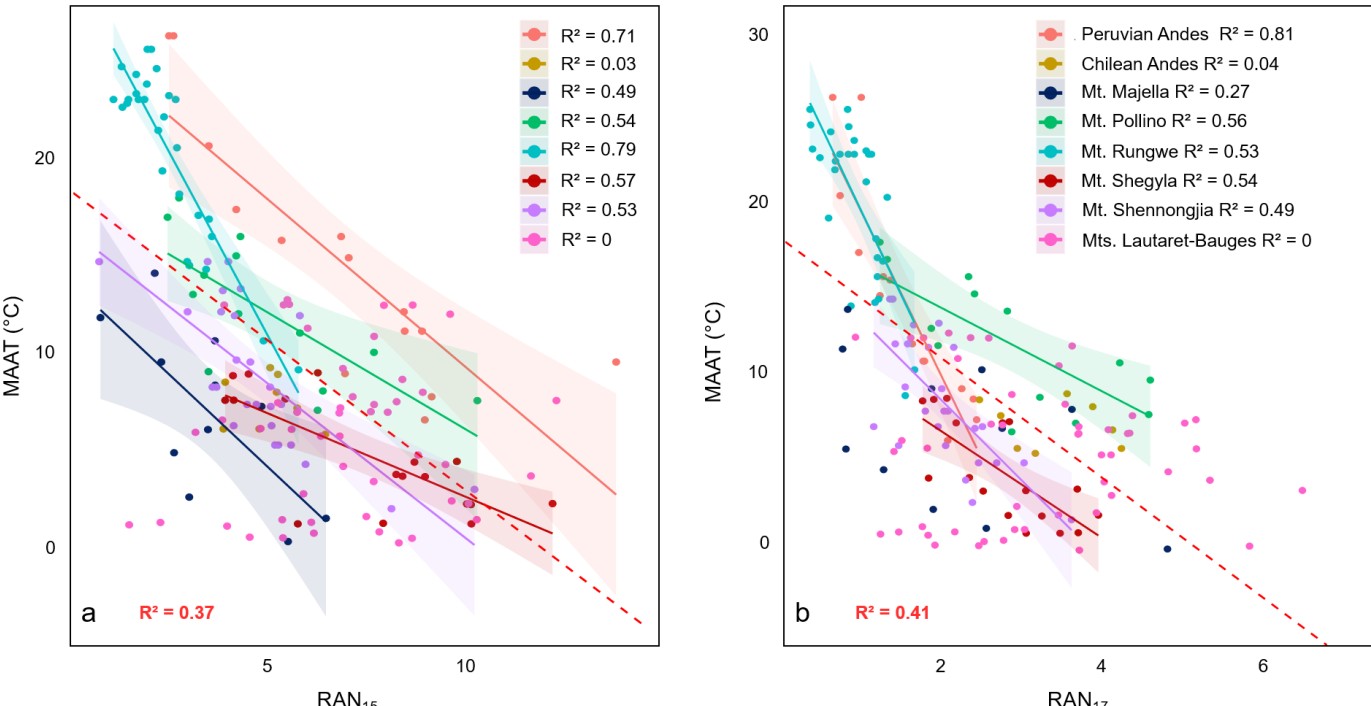

**Figure 5.** Linear regressions between (a) MAAT and RAN$_{15}$ and (b) MAAT and RAN$_{17}$ along the 8 altitudinal transects investigated. Dotted lines represent the 95% confidence interval for each regression and colored areas represent the 95% confidence interval for each regression. Data from Mts. Majella and Rungwe were taken from Huguet et al. (2019). Data from Mt. Shennongjia were taken from Wang et al. (2016). Data from Mts. Lautaret-Galibier were taken from Véquaud et al. (2021). Only significant regressions (p < 0.05) are shown.

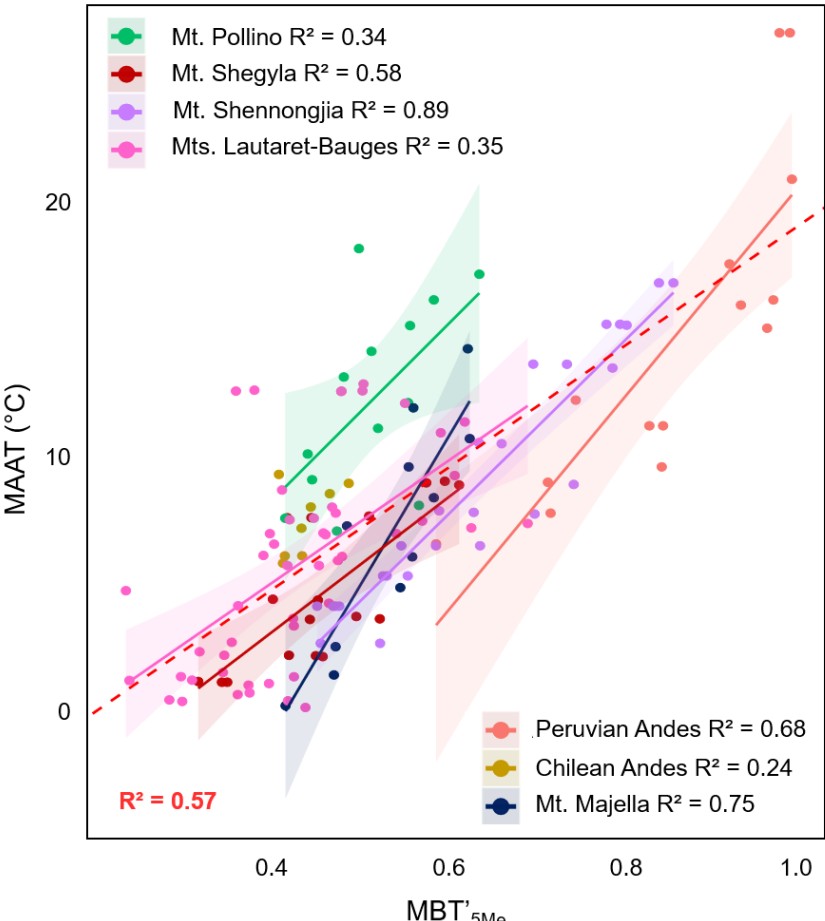

**Figure 6.** Linear regressions between (a) MAAT and MBT'$_{5Me}$ along 7 of the 8 altitudinal transects investigated. Data from Mt. Rungwe (Coffinet et al., 2014), for which 5- and 6-methyl brGDGTs were not separated, were not included in this graph. Dotted lines represent the 95% confidence interval for each regression and colored areas represent the 95% confidence interval for each regression. Data from Mt. Majella were taken from Huguet et al. (2019). Data from Mts. Lautaret-Galibier were taken from Véquaud et al. (2021). Data from Mt. Shennongjia were taken from Yang et al. (2015). The global soil calibration by De Jonge et al. (2014) was applied to all these transects. Only significant regressions (p < 0.05) are shown.

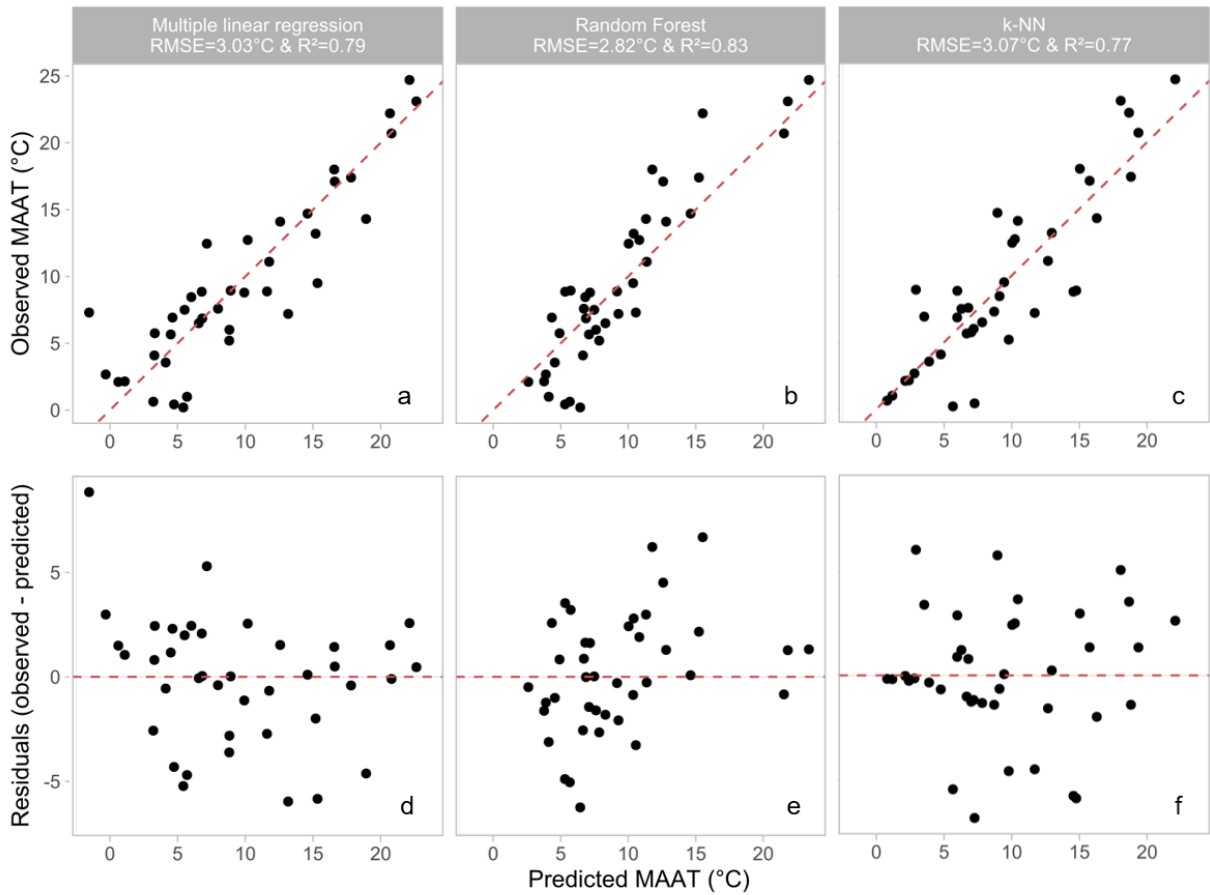

**Figure 7.** Results of the three different models tested to reconstruct the MAAT from 3-OH FA distribution: observed MAAT (°C) vs Predicted MAAT (°C) for (a) the multiple linear regression model, (b) the random forest model and (c) the k-NN method. MAAT residuals plotted against the predicted MAAT for (d) the multiple linear regression model, (e) the random forest model and (f) the k-NN method.

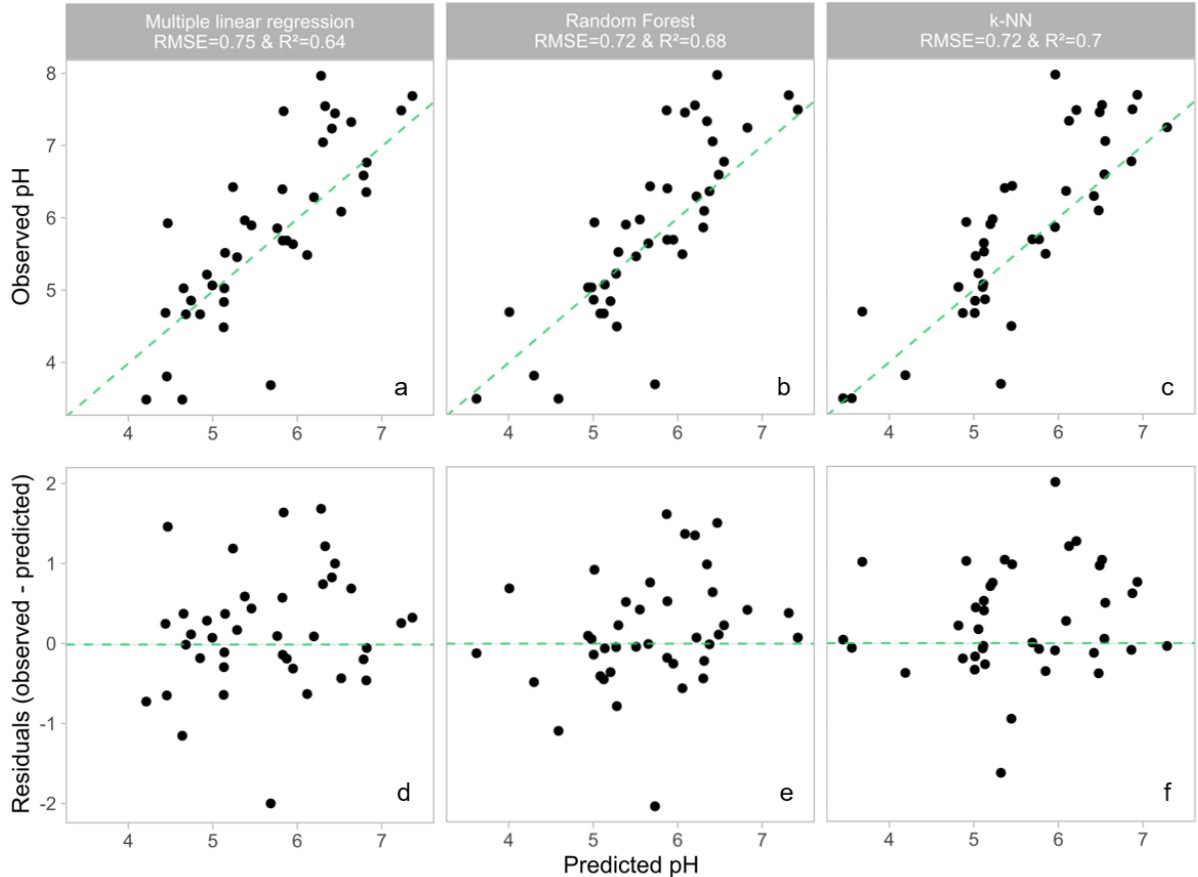

**Figure 8.** Results of the three different models tested to reconstruct the pH from 3-OH FA distribution: observed pH vs predicted pH for (a) the multiple linear regression model, (b) the random forest model, (c) the k-NN method. pH residuals plotted against the predicted pH for (d) the multiple linear regression model, (e) the random forest model and (f) the k-NN method.

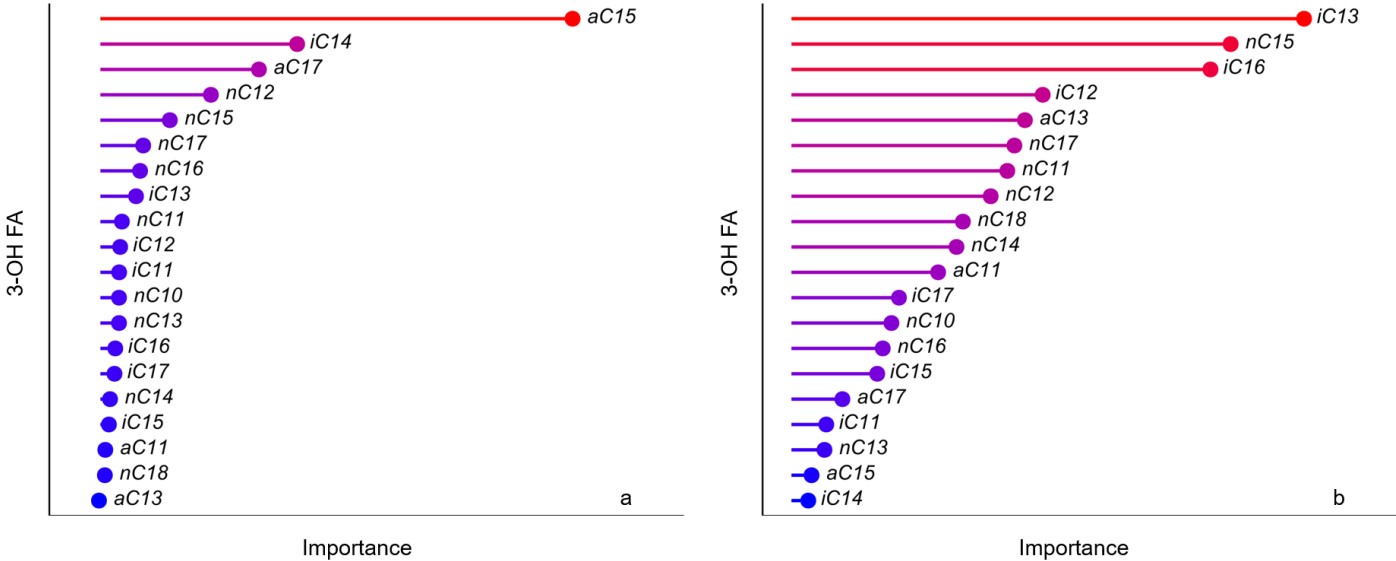

**Figure 9.** Importance (arbitrary unit) of the 3-OH FAs used to estimate (a) MAAT and (b) pH in the random forest models proposed in this study according to the permutation importance method (Breiman, 2001).

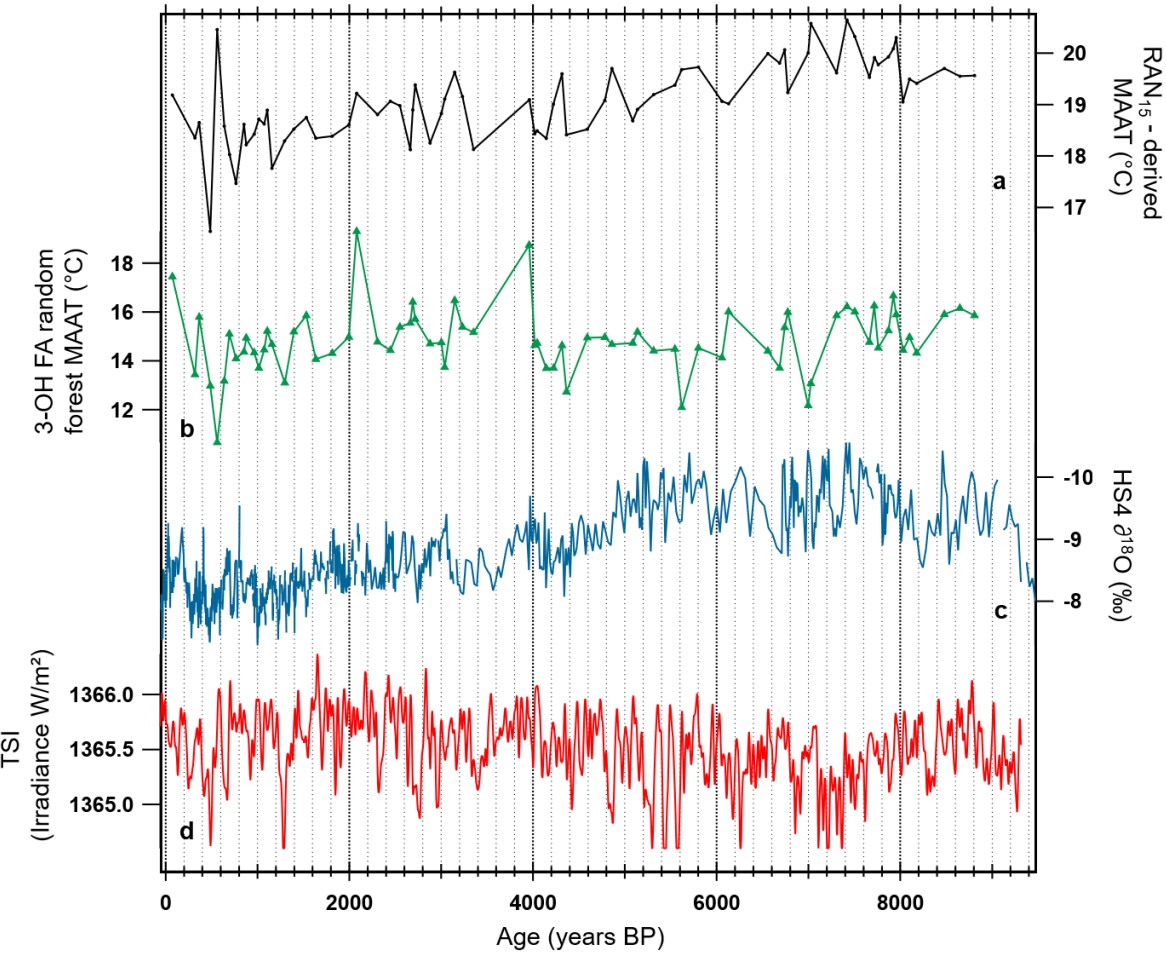

**Figure 10.** Comparison of the 3-OH FA model-MAAT record with other time-series and proxy records for the HS4 speleothem (Wang et al., 2018). (a) RAN₁₅-MAAT record reconstructed using a local Chinese calibration (Wang et al., 2016; Wang et al., 2018). (b) 3-OH FA random forest model-MAAT. (c) The CaCO₃ oxygen isotope record (Hu et al., 2008b). (d) Total solar irradiance (TSI; W/m²) during the Holocene (past 9300 years) based on a composite described in Steinhilber et al. (2009).

| ID | Location | Altitude (m) | MAAT(°C) | pH | RAN$_{15}$ | RAN$_{17}$ | RIAN | MBT'$_{5Me}$ | CBT' |
|---|---|---|---|---|---|---|---|---|---|
| 1 | Peruvian Andes | 194 | 26.4 | 3.7 | 2.45 | 0.96 | 0.47 | 0.96 | -1.09 |
| 2 | Peruvian Andes | 210 | 26.4 | 4 | 2.56 | 0.61 | 0.60 | 0.97 | -1.92 |
| 3 | Peruvian Andes | 1063 | 20.7 | 4.7 | 3.46 | 0.70 | 0.54 | 0.98 | -1.76 |
| 4 | Peruvian Andes | 1500 | 17.4 | 3.5 | 4.15 | 0.93 | 0.51 | 0.91 | -1.55 |
| 5 | Peruvian Andes | 1750 | 15.8 | 3.6 | 5.30 | 1.32 | 0.51 | 0.92 | -1.62 |
| 6 | Peruvian Andes | 1850 | 16 | 3.5 | 6.81 | 1.23 | 0.54 | 0.96 | -1.76 |
| 7 | Peruvian Andes | 2020 | 14.9 | 3.4 | 7.00 | 1.19 | 0.54 | 0.95 | -1.68 |
| 8 | Peruvian Andes | 2520 | 12.1 | 3.7 | 8.40 | 1.59 | 0.53 | 0.74 | -1.42 |
| 9 | Peruvian Andes | 2720 | 11.1 | 3.6 | 8.42 | 1.73 | 0.48 | 0.83 | -1.45 |
| 10 | Peruvian Andes | 3020 | 9.5 | 3.4 | 13.78 | 2.21 | 0.44 | 0.83 | -1.21 |
| 11 | Peruvian Andes | 3200 | 8.9 | 3.5 | 6.91 | 2.35 | 0.37 | 0.71 | -1.48 |
| 12 | Peruvian Andes | 3025 | 11.1 | 3.5 | 8.86 | 1.74 | 0.52 | 0.82 | -1.66 |
| 13 | Peruvian Andes | 3400 | 7.7 | 3.4 | 9.10 | 2.39 | 0.40 | 0.71 | -1.39 |
| 14 | Peruvian Andes | 3644 | 6.5 | 3.4 | 8.93 | 2.03 | 0.67 | 0.58 | -1.21 |
| 15 | Mt. Shegyla, Tibet | 3106 | 8.9 | 5.53 | 6.22 | 2.02 | 0.51 | 0.59 | -0.83 |
| 16 | Mt. Shegyla, Tibet | 3117 | 8.9 | 6.43 | 4.47 | 1.86 | 0.36 | 0.57 | -0.35 |
| 17 | Mt. Shegyla, Tibet | 3132 | 8.8 | 6.01 | 4.07 | 1.72 | 0.43 | 0.61 | -0.47 |
| 18 | Mt. Shegyla, Tibet | 3344 | 7.6 | 6.03 | 5.40 | 2.80 | 0.34 | 0.51 | -0.67 |
| 19 | Mt. Shegyla, Tibet | 3355 | 7.5 | 5.87 | 4.09 | 2.71 | 0.23 | 0.44 | -0.39 |
| 20 | Mt. Shegyla, Tibet | 3356 | 7.5 | 5.52 | 3.87 | 2.14 | 0.25 | 0.42 | -0.70 |
| 21 | Mt. Shegyla, Tibet | 4030 | 3.7 | 5.21 | 8.21 | 3.64 | 0.43 | 0.49 | -1.10 |
| 22 | Mt. Shegyla, Tibet | 4046 | 3.6 | 4.68 | 8.37 | 3.00 | 0.49 | 0.52 | -1.17 |
| 23 | Mt. Shegyla, Tibet | 4050 | 3.6 | 4.61 | 8.94 | 2.47 | 0.50 | 0.44 | -1.33 |
| 24 | Mt. Shegyla, Tibet | 3912 | 4.3 | 5.04 | 9.74 | 2.30 | 0.48 | 0.40 | -2.39 |
| 25 | Mt. Shegyla, Tibet | 3918 | 4.3 | 4.68 | 8.67 | 1.80 | 0.56 | 0.45 | -2.23 |
| 26 | Mt. Shegyla, Tibet | 4298 | 2.1 | 5.04 | 10.00 | 2.78 | 0.50 | 0.45 | -2.04 |
| 27 | Mt. Shegyla, Tibet | 4295 | 2.2 | 4.87 | 12.17 | 3.90 | 0.50 | 0.42 | -1.07 |
| 28 | Mt. Shegyla, Tibet | 4304 | 2.1 | 5.26 | 10.10 | 3.20 | 0.46 | 0.46 | -1.14 |
| 29 | Mt. Shegyla, Tibet | 4479 | 1.1 | 5.26 | 10.11 | 3.42 | 0.52 | 0.35 | -1.27 |
| 30 | Mt. Shegyla, Tibet | 4479 | 1.1 | 5.07 | 5.71 | 3.00 | 0.50 | 0.35 | -0.84 |
| 31 | Mt. Shegyla, Tibet | 4474 | 1.1 | 5.24 | 7.88 | 3.65 | 0.42 | 0.32 | -1.15 |
| 32 | Mt. Pollino, Italy | 0 | 18 | 6.78 | 2.71 | 1.19 | 0.15 | 0.50 | 0.31 |
| 33 | Mt. Pollino, Italy | 200 | 17 | 6.19 | 2.41 | 1.28 | 0.30 | 0.63 | 0.34 |
| 34 | Mt. Pollino, Italy | 400 | 16 | 6.13 | 4.26 | 2.29 | 0.22 | 0.58 | 0.35 |
| 35 | Mt. Pollino, Italy | 600 | 15 | 6.14 | 4.15 | 2.36 | 0.22 | 0.55 | 0.43 |
| 36 | Mt. Pollino, Italy | 800 | 14 | 4.53 | 3.34 | 2.77 | 0.34 | 0.51 | -0.24 |
| 37 | Mt. Pollino, Italy | 1000 | 13 | 5.41 | 3.06 | 1.83 | 0.28 | 0.48 | 0.10 |
| 38 | Mt. Pollino, Italy | 1200 | 12 | 6.37 | 4.21 | 1.91 | 0.24 | 0.55 | 0.43 |
| 39 | Mt. Pollino, Italy | 1400 | 11 | 5.62 | 5.77 | 4.16 | 0.18 | 0.52 | 0.40 |
| 40 | Mt. Pollino, Italy | 1600 | 10 | 4.93 | 7.64 | 4.54 | 0.27 | 0.44 | -0.13 |
| 41 | Mt. Pollino, Italy | 1800 | 9 | 4.91 | 3.45 | 3.17 | 0.25 | 0.45 | -0.07 |
| 42 | Mt. Pollino, Italy | 2000 | 8 | 5.52 | 6.35 | 4.52 | 0.19 | 0.56 | 0.40 |
| 43 | Mt. Pollino, Italy | 2100 | 7.5 | 5.91 | 10.26 | 3.62 | 0.19 | 0.42 | 0.38 |
| 44 | Mt. Pollino, Italy | 2200 | 7 | 5.85 | 6.21 | 2.82 | 0.31 | 0.47 | 0.34 |
| 45 | Chilean Andes | 690 | 9.2 | 5.38 | 5.01 | 3.51 | 0.42 | 0.41 | -0.80 |
| 46 | Chilean Andes | 870 | 8.9 | 5.62 | 5.21 | 2.43 | 0.39 | 0.49 | -0.52 |
| 47 | Chilean Andes | 891 | 7.9 | 4.94 | 5.18 | 2.69 | 0.53 | 0.44 | -0.94 |
| 48 | Chilean Andes | 915 | NA | 6.75 | 4.67 | 4.25 | 0.21 | NA | NA |
| 49 | Chilean Andes | 980 | 8.5 | 5.63 | 3.87 | 3.83 | 0.28 | 0.46 | -0.66 |
| 50 | Chilean Andes | 985 | 5.8 | 4.67 | 6.41 | 3.12 | 0.48 | 0.41 | -1.83 |
| 51 | Chilean Andes | 1125 | 6.0 | 5.00 | 3.83 | 4.18 | 0.46 | 0.42 | -1.02 |
| 52 | Chilean Andes | 1151 | 6.0 | 5.89 | 4.74 | 2.89 | 0.33 | 0.43 | -0.32 |
| 53 | Chilean Andes | 1196 | 7.1 | 5.79 | 5.70 | 4.07 | 0.34 | 0.43 | -0.40 |
| 54 | Chilean Andes | 1385 | NA | 4.43 | 4.85 | 1.91 | 0.39 | 0.41 | -2.28 |

**Table 1.** List of the soil samples collected along Mts. Shegyla, Pollino, Peruvian Andes and Chilean Andes, with corresponding altitude (m), MAAT (°C), pH and 3-OH FA/brGDGT-derived indices.

| | Model | n (training) | n (test) | R² | RMSE | Variance in residuals | Mean absolute error | Lower estimation limit | Upper estimation limit |
|---|---|---|---|---|---|---|---|---|---|
| MAAT (°C) | $RAN_{15}$ | - | 168 | 0.37 | 5.5 | 29.8 | 4.0 | -3.1 | 17.2 |
| | $RAN_{17}$ | - | 168 | 0.41 | 5.3 | 27.9 | 3.9 | -4.3 | 17.0 |
| | k-NN | 128 | 40 | 0.77 | 3.1 | 9.4 | 2.3 | 0.5 | 25.0 |
| | Multiple linear regression | 128 | 40 | 0.79 | 3.0 | 9.2 | 2.3 | -1.2 | 25.8 |
| | Random forest | 128 | 40 | 0.83 | 2.8 | 8.0 | 2.2 | 0.8 | 24.9 |
| pH | RIAN | - | 168 | 0.34 | 1.0 | 1.0 | 0.8 | 4.1 | 7.9 |
| | k-NN | 128 | 40 | 0.70 | 0.7 | 0.5 | 0.5 | 3.4 | 8.7 |
| | Multiple linear regression | 128 | 40 | 0.64 | 0.8 | 0.6 | 0.6 | 4.0 | 8.3 |
| | Random forest | 128 | 40 | 0.68 | 0.7 | 0.5 | 0.5 | 3.5 | 7.8 |

**Table 2.** Characteristics of the different models proposed in this study to estimate MAAT and pH: R², RMSE, variance of the residuals, mean absolute error (MAE) and the upper and lower limits of estimation. The "training" samples were used to develop the different machine learning models, which were then tested on a "test" sample set.