# Peer review of "Development of global temperature and pH calibrations based on bacterial 3-hydroxy fatty acids in soils"

_Biogeosciences, 2020_

## Referee Comment (RC1) · Anonymous Referee #1 · 18 Dec 2020

Véquaud and co-authors have compiled a large set of soils along elevation gradients in an attempt to determine global relationships of 3-OH FAs with MAAT and soil pH. They find that there is a global relationship with MAAT, but that relationships are better on a local scale. They then use several statistical methods to quantify the relationships and to develop a temperature proxy based on 3-OH FAs. I appreciate the effort of the authors to compile this relatively large dataset and their attempt to assess the temperature and pH sensitivity of these compounds on a global scale. I do however have a few concerns and suggestions to further improve this work before I can recommend it for publication in Biogeosciences.

[Figure]

Major comments:

- BrGDGT data: Next to 3-OH FA data, the authors also present brGDGT data from the same samples. However, I fail to see the added value of the brGDGT data to this manuscript, and I thus suggest leaving this data out. Especially given the facts that a) the focus of the paper is on assessing the temperature and pH dependency of 3-OH FAs, which has nothing to do with brGDGTs, and b) most of the brGDGT datasets presented here are already published elsewhere. In addition, the brGDGT data for two of the elevation transects (Rungwe and Shennongjia) are based on 'old' chromatography methods that do not separate the 5- and 6-methyl brGDGT isomers. Note that these datasets should not be directly compared with 'new' brGDGT data for the other transects, as MAATs and pH are derived based on different equations. The 'old' MBT and CBT indices both contain compounds that appear to have a 6-methyl isomer (e.g. IIa and IIIa in the MBT index, and IIa and IIb in the CBT index). This makes it impossible to determine whether a relative change in the peak area of one of these compounds is driven by the 5- or by the 6-methyl isomer. Especially in the case of MBT this is problematic, as the occurrence of 6-methyl brGDGTs is linked to changes in pH and not MAAT (De Jonge et al., 2014). The authors may be unaware, but 'new' brGDGT data for Mt. Shennongjia has been published by Yang et al., 2015, who, in fact, state that 6-methyl brGDGTs significantly affect the performance of the MBT' in this transect, proving my point that the use of 'new' data is important (if at all, in case of this manuscript).

- Ecology of 3-OH FA producers: There are several points in the manuscript where the authors state that a better understanding of the adaptation mechanisms of 3-OH FA producing organisms to changing environmental conditions may improve the proxies (e.g. L484), or that the use of (a) subset(s) of 3-OH FAs could results in better relations with MAAT or pH (as suggested for the RIAN-pH relationship). I feel that there is a missed opportunity here and improvements should be made in a revised version. If it is known that 3-OH FAs are produced by Gram negative bacteria, what can we tell about

their occurrence, response to environmental change, etc? Or is this group simply too big to say anything sensible?

- Proxy model development From the brGDGT and other biomarker proxy studies we have learned that many of the transfer functions based on linear regressions suffer from regression dilution (e.g. Tierney and Tingley, 2014; Naafs et al., 2017; Dearing-Crampton-Flood et al., 2020). I miss an assessment of this aspect for the 3-OH FA proxies in this manuscript, which are all based on methods that involve linear regressions. I would also like to point out that there is a general move towards using Bayesian statistics to derive proxies. This approach will allow you to treat your biomarker-based index as dependent variable in the calibration model whilst accounting for possible errors in the measurement of both MAAT and your biomarker index (see the explanation in the introduction of Dearing Crampton-Flood et al., 2020). I encourage the authors to look into this method and consider its use for their purpose.

- Application of the 3-OH FA proxies The authors have chosen a stalagmite as test archive for their proxy. It may be the only 3-OH FA-based paleorecord in the literature so far, but I really do not think that a speleothem is a suitable archive to test proxies that are based on the occurrence and behavior of 3-OH FAs in soils. Speleothems and soils are completely different environments, and it is no way guaranteed, let alone tested that the sources of 3-OH FAs in both environments are the same, let alone their relationship with temperature and/or pH. I would either leave this example out, or find another (soil!) archive to test their proxies.

- Data compilation I notice that the authors have included one dataset that is still under review in Organic Geochemistry (Véquaud et al). In case this manuscript is accepted first (actually, the data is already published online as part of this discussion paper), is there any novelty left for the manuscript in OG? Aren't you shooting yourself in the foot here?

Minor comments: L70: the use of Eglinton and Eglinton as a reference to proxies

based on membrane lipids is technically not incorrect, but it seems more appropriate to use e.g. Schouten et al., 2002, Weijers et al., 2007 ('they were first'), or the review paper of Schouten et al., 2013. L79: For aquatic environments, references to Peterse et al., 2009 (marine brGDGTs) and Tierney and Russell, 2009 and Sinninghe Damsté et al., 2009 (lacustrine brGDGTs) are better suited ('they were first'). L98: explain what RAN15 and RAN17 stand for prior to using the abbreviation. L109: replace calibration by relation L112: explain RIAN prior to using the abbreviation. L130: replace 'collected along globally distributed' by 'determined in several elevation transects' L134: This sentence is not really grammatically correct. Rewrite to something like 'even though brGDGT-based MAAT and pH reconstructions still have a relatively large uncertainty, they can serve as a reference to test the temperature dependency of 3OH FAs analyzed in the same dataset.' L137: use either 'the 3OH FA distribution' or '3OH FA distributions'. L177: what is a 'fir'? L230: can you add the compound that was used as an internal standard? L274: Is there a reason why the latest definition of the CBT' index by De Jonge et al., 2014 was not used (their Eq. 10)? This index takes the separation of 5- and 6-methyl isomers into account and also has a lower RMSE compared to the one of Peterse et al., 2012 that is based on data generated with the 'old' chromatography method. If not, I would recommend using the CBT' index. L397: replace inertia by variance (inertia means the tendency to remain unchanged, which is not what you mean here, I presume) L400: how reliable is 'location' as a factor influencing 3OH FAs? I don't really think that microorganisms would take their coordinates into account when they synthesize 3OH FAs. Instead, I suspect that location represents an indirect parameter that integrates the local environmental conditions at each site. Thus, 3OH FAs are more likely linked to temperature, precipitation, vegetation, pH, whatever on the first PC, so I suggest re-running the PCA without taking location into account. L457-459: this sentence seems to be missing words? L459: is the lack of a relation between the RIAN and pH caused by the use of a linear model, or by the relatively small range of soil pH of this dataset? How can you tell? And would another model be better suitable to capture this relation? If so, which kind of model? L467: Please

refer to and compare with the newer calibration studies. Several have been published since 2012, but at least use De Jonge et al., 2014 that includes the 5- and 6-methyl isomers. L490: replace similarly by also L492 (here and elsewhere in the ms): replace 'prediction interval' by 'confidence interval'. L542ff: How does the MBT'5me relation with MAAT from this study compare to the one in the global surface soil dataset? Does it have the same slope and intercept?

General: Replace Damsté by Sinninghe Damsté

Figures 1 and 2: I'm not sure how useful it is to show the relative distributions of 3OH-FAs and brGDGTs, especially since every site represents an elevation gradient along which a large variation in temperature (and thus biomarker composition) is expected. Instead, display the proxy values with elevation (or completely delete them, as the proxies are already plotted against pH and MAAT in Figs 4-6). Figures 4 and 5 and 6. Can you add the regressions, or at least the regression coefficient (R2) of each significant regression to the panels? Figure 9: The y-axis of panel a does not have a title.

References: Dearing Crampton-Flood et al., 2020 GCA 268, 142-159 De Jonge et al., 2014 GCA 141, 97-112 Naafs et al., 2017 Org geochem 106, 48-56. Peterse et al., 2009 Org Geochem 40, 692-699. Peterse et al., 2012 GCA 96, 215-229. Schouten et al., 2002 EPSL 204, 265-274. Schouten et al., 2013 Org Geochem 54, 19-61. Sinninghe Damsté et al., 2009 GCA 73, 4232-4249. Tierney and Tingley, 2014 GCA 127, 83-106 Tierney and Russell, 2009 Org Geochem 40, 1032-1036 Weijers et al., 2007 GCA 71, 703-713 Yang et al., 2015 Org Geochem 82, 42-53.

---

## Referee Comment (RC2) · Anonymous Referee #2 · 3 Feb 2021

I have finished the review of the manuscript "development of global temperature and p calibrations based on bacterial 3-hydroxy fatty acids in soils", submitted to biogeosciences by Pierre Vequaud and co-authors.

In general, the authors present a good contribution to the developing field of 3-OH fatty acids, by compiling several published and novel altitudinal gradients. The performance of the local and compiled linear regressions are compared with that of brGDGT lipids, for those sites where 5- and 6-methyl compounds were separated (and where the MBT'5ME could thus be calculated). This is an interesting dataset, and a necessary next step in the development of 3-OH fatty acids as temperature proxies.

However, the performance of the three additional models (multiple linear regression, random forest and k-nearest neighbor), that move away from the previously established RAIN, RAN15 and RAN17 ratios, are not evaluated thoroughly. There is no explanation of what the models represent, and why they allow a better correlation between the 3-OH fatty acids and MAAT or pH, compared to the simple linear regression. There is for instance no analysis of residuals, and no indication of maximum reconstructed temperatures. I would recommend the authors to look at a recent calibration paper (fi Dearing-Crampton Flood, 2020), to see what the state-of-the-art in lipid proxy calibration is. As it stands, the authors will have a difficult time convincing readers that the more complex models are the preferred choice when doing 3-OH FA-based climate reconstructions.

When developing a MAAT-calibration, the authors should also mention the following weakness: there is currently only 3 samples with MAAT > 15 °C in the dataset (maximum temperature: +- 20 °C). Do these samples influence the MAAT calibration disproportionally?

Minor comments: L 91. Include 'bacterial' before 'lipids'. L 129. 'more developed statistical approach', perhaps rephrase as 'further development of the statistical approaches'? L 148. For easy comparison: include the altitudinal, pH and MAAT ranges of the previously published transects. L 200-201. Soil sensors have been used, but the calibration is done with mean annual air temperature (MAAT) instead. As the soil sensor data is not used or discussed further, I'd remove their mention here, and use the MAAT for this site as well. L 322 (and further in the manuscript). As far as I can see, the supp. tables referred to are not present. Please include these tables in a revised version.

L 541. Perhaps the authors can refer here to De Jonge et al., 2019, who argue that temperature can modulate the soil bacterial composition, and the dependency between MBT'5ME and soil temperature? L551. Here the authors should explain better how 'more complex models' allow to take the complexity of each site into account. At this

point, in the manuscript, we have observed that the linear dependency between the 3-OH fatty acids is better on a local scale, than on a global scale. What mechanism do the authors propose for this, and how can a more complex model correct for it? L 567. Have the authors done a selection of the fatty acids that are necessary for the model? For instance, forward selection or reverse selection? Does the model not suffer from overfitting? Is there any fatty acids that are generally present in low abundance (and can thus be absent in geological archives) that are important for the regression? If yes, is it prudent to include these low abundance compounds in the model as well? L 587. The random forest model and k-NN model need to be explained much better before the results are presented. What are they based on and how do they compare variability in the lipid distribution with the MAAT? L 593. Same comments as at L 567. L 651. Can the authors comment on the probable source of the 3-OH fatty acids in the speleothem? Can we assume that all lipids are derived from the soil, or is there a (variable) proportion produced in the cave environment as well?

Refs: Dearing Crampton-Flood E., Tierney J. E., Peterse F., Kirkels F. M. S. A. and Sinninghe Damsté J. S. (2020) BayMBT: A Bayesian calibration model for branched glycerol dialkyl glycerol tetraethers in soils and peats. Geochimica et Cosmochimica Acta 268, 142–159. De Jonge C., Radujković D., Sigurdsson B. D., Weedon J. T., Janssens I. and Peterse F. (2019) Lipid biomarker temperature proxy responds to abrupt shift in the bacterial community composition in geothermally heated soils. Organic Geochemistry, S0146638019301275.

---

## Author Comment (AC1) · 18 Feb 2021

We thank the anonymous reviewer for these comments. A detailed list of changes and arguments answering to the different comments is provided below.

Reviewers' comments:

Reviewer #1: Review

Summary - Véquaud and co-authors have compiled a large set of soils along elevation gradients in an attempt to determine global relationships of 3-OH FAs with MAAT and soil pH. They find that there is a global relationship with MAAT, but that relationships

are better on a local scale. They then use several statistical methods to quantify the relationships and to develop a temperature proxy based on 3-OH FAs. I appreciate the effort of the authors to compile this relatively large dataset and their attempt to assess the temperature and pH sensitivity of these compounds on a global scale. I do however have a few concerns and suggestions to further improve this work before I can recommend it for publication in Biogeosciences.

Author comment : We would like to thank the reviewer for his comments and for acknowledging the work made to evaluate the temperature and pH sensitivity of 3-OH FAs on a global scale. It should be highlighted that half of the 3-OH FA and brGDGT data presented in the manuscript are new (those corresponding to 4 altitudinal transects, i.e. Mt. Pollino, Italy; Mt. Shegyla, Tibet; Peruvian and Chilean Andes). Our new lipid data were combined with those previously published (Mt. Shennongjia, China, Yang et al., 2015 and Wang et al., 2016; Mt. Rungwe, Tanzania, Coffinet et al., 2017 and Huguet et al., 2019; Mt. Majella, Italy, Huguet et al., 2019; French Alps, Véquaud et al., 2021). Gathering these "old" and new data allowed establishing a large dataset which was used to evaluate the applicability of 3-OH FAs as MAAT and pH proxies in soils and to establish a comparison with the more established brGDGT proxies. Previous papers presenting brGDGT and 3-OH FA calibrations similarly rely on a combination of previously and newly acquired data (e.g. De Jonge et al., 2014; Naafs et al., 2017a; Huguet et al., 2019; Dearing Crampton-Flood et al., 2020).We also appreciate that the reviewer did not question the way the data were acquired, processed and interpreted.

Major comments: - BrGDGT data: Next to 3-OH FA data, the authors also present brGDGT data from the same samples. However, I fail to see the added value of the brGDGT data to this manuscript, and I thus suggest leaving this data out.

Author comment : We kindly disagree with the reviewer here. Indeed, to date, brGDGTs are the only organic proxies available for temperature and pH reconstruction in terrestrial settings. Therefore, as mentioned in the manuscript, the development of new molecular proxies, independent of and complementary to brGDGTs, is essential to

improve the reliability of paleotemperature (and pH) reconstructions in such settings". 3-OH FAs could provide such a proxy. As a result, 3-OH FAs and brGDGTs have to be concomitantly analyzed to assess their reliability and complementarity as independent temperature and pH proxies. Such a comparison cannot be made if brGDGT data are not presented and cannot be used as "a reference proxy". The interest of brGDGT data in this manuscript will be further clarified in the revised version. It should also be noted that all the previous papers dealing with the influence of environmental parameters on 3-OH FA distribution in soils (Wang et al., 2016; Huguet et al., 2019; Véquaud et al., 2021) made a comparison between brGDGT-derived and 3-OH FA-derived proxies. Our approach is thus consistent with these previous papers.

a) the focus of the paper is on assessing the temperature and pH dependency of 3-OH FAs, which has nothing to do with brGDGTs.

Author comment : We agree with the reviewer that the main aim of this study is to evaluate the influence of pH and MAAT on 3-OH FA distribution in globally distributed topsoils. Nevertheless, it would be inappropriate not to compare these results with those derived from brGDGTs, the only other organic temperature/pH proxies terrestrial environments, as mentioned above. The comparison between brGDGT and 3-OH FA data is necessary, as it highlights the similarities and differences between these two independent families of molecules in terms of response to temperature and pH. This also shows the link between this study and previous ones dealing with the development of organic temperature/pH proxies in terrestrial settings. The different proxies and related studies should not be disconnected from each other.

b) most of the brGDGT datasets presented here are already published elsewhere.

Author comment : We kindly disagree with the reviewer, as only half of the brGDGT dataset is already published. The other half of the dataset, corresponding to 4 altitudinal transects (Mt. Pollino, Italy; Mt. Shegyla, Tibet; Peruvian and Chilean Andes), is presented here for the first time.

- In addition, the brGDGT data for two of the elevation transects (Rungwe and Shennongjia) are based on 'old' chromatography methods that do not separate the 5- and 6-methyl brGDGT isomers. Note that these datasets should not be directly compared with 'new' brGDGT data for the other transects, as MAATs and pH are derived based on different equations. The 'old' MBT and CBT indices both contain compounds that appear to have a 6-methyl isomer (e.g. IIa and IIIa in the MBT index, and IIa and IIb in the CBT index). This makes it impossible to determine whether a relative change in the peak area of one of these compounds is driven by the 5- or by the 6-methyl isomer. Especially in the case of MBT this is problematic, as the occurrence of 6-methyl brGDGTs is linked to changes in pH and not MAAT (De Jonge et al., 2014). The authors may be unaware, but 'new' brGDGT data for Mt. Shennongjia has been published by Yang et al., 2015, who, in fact, state that 6-methyl brGDGTs significantly affect the performance of the MBT' in this transect, proving my point that the use of 'new' data is important (if at all, in case of this manuscript).

Author comment : We totally agree with the reviewer on the importance of separating 5- and 6-methyl brGDGT isomers and never said the opposite in our paper. The new data presented in this study were all obtained using two silica columns, allowing the separation of these isomers. Similarly, previously published brGDGT data from the French Alps (Véquaud et al., 2021) and Mt. Majella (Huguet et al., 2019) were also acquired with this new methodology. Regarding Mt. Shennongjia, we agree with the reviewer that 5- and 6-methyl brGDGTs were indeed separated in the Organic Geochemistry paper published by Yang et al. (2015). However, the relative abundances of the individual 5- and 6-methyl brGDGTs were not directly provided in this paper, which prevented us to use them. Following up the review, the authors kindly sent us these data upon request. These abundances and the values of MBT'5Me index can now be used in our paper. This point will be clarified in the revised manuscript. On the whole, data from 7 out of 8 transects investigated in this study were obtained using the "new" chromatography method allowing the separation of 5- and 6-methyl brGDGTs. All these data can be directly compared and exploited. This will be clearly mentioned in the revised manuscript. Regarding the data from Mt. Rungwe, they were obtained using the "old" chromatography method relying on a cyano column. As 5- and 6-methyl brGDGTs were not separated, this dataset cannot be directly compared with the 7 others. Nevertheless, the data for the 8 transects with the old brGDGT separation method will be presented as supplementary data in the revised version of the manuscript. These results show the same trends with and without isomer separation.

- Ecology of 3-OH FA producers: There are several points in the manuscript where the authors state that a better understanding of the adaptation mechanisms of 3-OH FA producing organisms to changing environmental conditions may improve the proxies (e.g. L484), or that the use of (a) subset(s) of 3-OH FAs could results in better relations with MAAT or pH (as suggested for the RIAN-pH relationship). I feel that there is a missed opportunity here and improvements should be made in a revised version. If it is known that 3-OH FAs are produced by Gram negative bacteria, what can we tell about their occurrence, response to environmental change, etc? Or is this group simply too big to say anything sensible?

Author comment : 3-OH FAs are produced by Gram-negative bacteria which, as suspected by the reviewer, are a highly diverse, non-monophyletic and ubiquitous group of microorganisms represented by numerous genera (Lecointre and Guyader, 2006). This explains the large genetic and biochemical differences between the various Gram-negative bacteria. The numerous genera of Gram-negative bacteria are characterized by highly diverse lipid profiles, with different relative abundances of the C10-C18 3-OH FAs, and by the fact that all the homologues are not biosynthesized by all the strains (Wilkinson et al., 1988). Thus, the 3-OH FA lipid distribution in soils is highly dependent on the diversity of Gram-negative bacterial species (Parker et al., 1982; Bhat and Carlson, 1992; Zelles, 1999), which may vary with altitude (Margesin et al., 2009; Siles and Margesin, 2016). Considering these different facts, it is difficult to draw out a general hypothesis on the response of Gram-negative bacteria and their associated lipids to environmental changes. A multiplicity of environmental parameters, including

MAAT and pH, may have an influence on 3-OH FA lipid distribution. The latter can reflect changes in the diversity of bacteria communities (e.g. Siles and Margesin, 2016) and/or adaptation of a constant community.

- Proxy model development From the brGDGT and other biomarker proxy studies we have learned that many of the transfer functions based on linear regressions suffer from regression dilution (e.g. Tierney and Tingley, 2014; Naafs et al., 2017; Dearing-Crampton-Flood et al., 2020). I miss an assessment of this aspect for the 3-OH FA proxies in this manuscript, which are all based on methods that involve linear regressions.

Author comment : We agree with the reviewer that a weak point of all methods based on regression methods is that they suffer from the phenomenon of regression dilution. That is why proposing alternative models that are not based on linear regression methods, such as those proposed in this study (random forest and k-NN algorithms), represent a major advantage. This aspect will be clearly mentioned in the revised manuscript and better assessed.

-I would also like to point out that there is a general move towards using Bayesian statistics to derive proxies. This approach will allow you to treat your biomarker-based index as dependent variable in the calibration model whilst accounting for possible errors in the measurement of both MAAT and your biomarker index (see the explanation in the introduction of Dearing Crampton-Flood et al., 2020). I encourage the authors to look into this method and consider its use for their purpose.

Author comment :Several methods were recently used to derive calibrations from organic proxies, such as multivariate regression (Russell et al., 2018), Deming regression (Naafs et al., 2017) , Bayesian linear regressions models (Tierney and Tingley, 2014; Dearing Crampton-Flood et al., 2020) or machine-learning models (Dunkley Jones et al., 2020). None of these methods excludes or prohibits the use of the others. Independent models should be used for the development of environmental calibrations, as each

of them has its own advantages and drawbacks. The different models have to be seen as complementary. As mentioned by the reviewer, Bayesian statistics were recently used to develop new calibration models, e.g. with brGDGTs (Dearing Crampton-Flood et al., 2020). The robustness of the Bayesian models relies on the fact that they consider a given index (for example, the MBT'5Me for brGDGTs) as the variable dependent on environmental parameters, in contrast with models based on linear regressions. This is realistic, as lipids are produced in response to the variations of environmental parameters. This model also avoids regression dilution phenomena, as mentioned by the reviewer. Nevertheless, the Bayesian method is always restricted to the use of an index, which allows the reconstruction of a given environmental parameter (for example temperature). This represents a limitation, as the relative distribution of bacterial lipids can be concomitantly influenced by several environmental parameters (e.g. Véquaud et al., 2021). In contrast, using bacterial relative abundances rather than a single index in models appears less restrictive, and more representative of the environmental complexity. The different models (multiple regression, k-NN method, Random Forest) presented in this paper allow overcoming the limitations related to the use of a single index, as they take into account the whole suite of bacterial lipids (here C10-C18 3-OH FAs) to estimate MAAT and pH values. In addition, the Bayesian model is parametric and it only takes into account linear relationships. Conversely, the k-NN and Random Forest models used in this study are non-parametric and can take into account non-linear environmental influences, in line with the intrinsic complexity of the environmental settings. This helps in improving the reliability of the models. Moreover, the models based on machine learning algorithms are built on a proportion of the total dataset (randomly defined) and then tested on the rest of the dataset, considered as independent. Such an approach improves the robustness of the models. As the Bayesian model, the k-NN and random forests are not based on linear regressions, thus avoiding the phenomenon of regression dilution. In conclusion, we chose the k-NN and random forest models in the present study, as they present major advantages for the development of robust calibrations between bacterial lipid distribution and

environmental parameters, as explained above. Such a choice will be better justified and discussed in the revised manuscript. Once again, this does not exclude the use of other independent and complementary models, such as those based on Bayesian statistics, but this is beyond the scope of the present study.

- Application of the 3-OH FA proxies The authors have chosen a stalagmite as test archive for their proxy. but I really do not think that a speleothem is a suitable archive to test proxies that are based on the occurrence and behavior of 3-OH FAs in soils. Speleothems and soils are completely different environments, and it is no way guaranteed, let alone tested that the sources of 3-OH FAs in both environments are the same, let alone their relationship with temperature and/or pH. I would either leave this example out, or find another (soil!) archive to test their proxies.

Author comment : Testing the performance and validity of the proposed global models on an existing archive is essential, as the calibration models are developed from modern samples to then provide robust and reliable paleoreconstructions. Most of the recent studies proposing new global or local MAAT calibrations, based on either brGDGTs or 3-OH FAs, adopted this approach and applied their models to a well-constrained archive to assess its robustness and accuracy (e.g. Naafs et al., 2017; Dearing Crampton-Flood et al., 2020; Wang et al., 2020; Yang et al., 2020). As a result, we developed MAAT calibrations from 3-OH FA distribution in soils using machine learning models (k-NN and random forest algorithms) and then tested and compared these calibrations using a speleothem archive. We agree with the reviewer that a soil archive would have better fitted to test our new calibrations and that soils and speleothems are different terrestrial settings. Nevertheless, the validation of the terrestrial models we propose requires using a well-known archive, for which paleoclimatic and paleoenvironmental data are available. This allows constraining the climatic/environmental conditions at the time of the reconstruction based on a multi-proxy approach. So far, the only 3-OH FA based paleorecord available in the literature is the Chinese speleothem previously investigated by Wang et al. (2018), which was the object of previous paleostudies, thus providing a context for the interpretation of the MAAT data. Moreover, Wang et al. (2018) suggested that the 3-OH FAs in the investigated speleothem are largely derived from the overlying soils based on the "broad similarity of 3-OH FA distributions in the overlying soils and stalagmites" as well as "site-specfic analyses of bacterial diversity and transport pathways". Thus, using 3-OH FA-based soil calibrations to reconstruct MAAT/pH from this speleothem is not so illogical. By the way, a local Chinese MAAT calibration obtained after the analysis of 3-OH FAs in 26 soil samples (Wang et al., 2016) was previously applied to this record. In the present study, we compare the MAAT estimates provided by our global soil calibrations with those previously obtained using the local calibration by Wang et al. (2016). In conclusion, it seems reasonable to us to test our new models on a terrestrial archive (even though this is not a soil) to show their potential and limits, as the concrete goal of such calibrations is to be used in the framework of paleoenvironmental studies. Such a choice will be better justified in the revised version of the manuscript to avoid any confusion in the mind of the reader.

- Data compilation I notice that the authors have included one dataset that is still under review in Organic Geochemistry (Véquaud et al.,). In case this manuscript is accepted first (actually, the data is already published online as part of this discussion paper), is there any novelty left for the manuscript in OG? Aren't you shooting yourself in the foot here?

Author comment : We thank the reviewer for this concern. However, there is no problem since the OG manuscript is in press (Véquaud et al., 2021) and available online (https://doi.org/10.1016/j.orggeochem.2021.104194). The OG paper aimed to investigate in detail the influence of environmental parameters on 3-OH FA distribution in soils from the French Alps. The present paper notably uses the data from the OG paper to create a large dataset based on soils from all over the world.

Minor comments:

L70: the use of Eglinton and Eglinton as a reference to proxies based on membrane lipids is technically not incorrect, but it seems more appropriate to use e.g. Schouten et al., 2002, Weijers et al., 2007 ('they were first'), or the re-view paper of Schouten et al., 2013.

Author comment : This will be corrected in the revised version of the manuscript.

L79: For aquatic environments, references to Peterse et al., 2009 (marine brGDGTs) and Tierney and Russell, 2009 and Sinninghe Damsté et al., 2009 (lacustrine brGDGTs) are better suited ('they were first').

Author comment : These references will be added in the revised version of this paper.

L98: explain what RAN15 and RAN17 stand for prior to using the abbreviation.

Author comment : This will be corrected.

L109: re-place calibration by relation L112: explain RIAN prior to using the abbreviation.

Author comment : This will be amended.

L130: replace 'collected along globally distributed' by 'determined in several elevation tran-sects'

Author comment : This sentence will be corrected following this comment.

L134: This sentence is not really grammatically correct. Rewrite to somethinglike 'even though brGDGT-based MAAT and pH reconstructions still have a relatively large uncertainty, they can serve as a reference to test the temperature dependency of 3OH FAs analyzed in the same dataset.'

Author comment : This sentence will be corrected as suggested.

L137: use either 'the 3OH FA distribution'or '3OH FA distributions'.

Author comment : This will be corrected.

L177: what is a 'fir'?

Author comment : Abies forrestii is a species of conifer in the family of Pinaceae, commonly referred as Forrest's fir. This sentence will be clarified in the revised version.

L230: can you add the compound that was used as an internal standard?

Author comment : The internal standard was a deuterated and methylated 3-OH fatty acid, the 3-hydroxytetradecanoic acid, 2,2,3,4,4-d5 (Sigma-Aldrich, France). This will be specified in the revised version of the paper.

L274: Is there a reason why the latest definition of the CBT' index by De Jonge et al., 2014 was not used (their Eq. 10)? This index takes the separation of 5- and 6-methyl isomers into account and also has a lower RMSEcompared to the one of Peterse et al., 2012 that is based on data generated with the'old' chromatography method. If not, I would recommend using the CBT' index.

Author comment : As suggested by the reviewer, the CBT' will be used in this manuscript instead of the CBT index. Nevertheless, the interpretations and conclusions from the CBT' remain the same as with the CBT, as the values of the two indices are similar.

L397: replace inertia by variance (inertia means the tendency to remain unchanged, which isnot what you mean here, I presume)

Author comment : This will be corrected.

L400: how reliable is 'location' as a factor influencing 3OH FAs? I don't really think that microorganisms would take their coordinates intoaccount when they synthesize 3OH FAs. Instead, I suspect that location represents anindirect parameter that integrates the local environmental conditions at each site. Thus, 3OH FAs are more likely linked to temperature, precipitation, vegetation, pH, whateveron the first PC, so I suggest re-running the PCA without taking location into account.

Author comment : We think there is a misunderstanding here. In fact, the PCAs were performed using the fractional abundances of the bacterial lipids (3-OH FAs and brGDGTs) in the total dataset, without taking the location into account. Nevertheless, it is possible to statistically qualify which variables best explain the distribution of the samples in the PCA. It clearly appears that the samples are located in the PCA based on to their location, which reflects the contrasting environmental conditions influencing the 3-OH FA and brGDGT source organisms at each site. This point will be clarified in the revised version of this manuscript.

L457-459: this sentence seems to be missing words?

Author comment : This sentence will be corrected as follows: "The distribution of 3-OH FAs varies greatly among Gram-negative bacterial species (Bhat and Carlson, 1992), which may account for significant variability in RIAN values among soils within a given transect".

L459: is the lack of a relationbetween the RIAN and pH caused by the use of a linear model, or by the relativelysmall range of soil pH of this dataset? How can you tell? And would another modelbe better suitable to capture this relation? If so, which kind of model?

Author comment : In our global dataset, pH values range between 3 and 8. This cannot be considered as a small range of variation. Our hypothesis is that more complex models (non-parametric and non-linear models, e.g. Random forest, k-NN method) take into account the non-linear influences of environmental parameters on lipid distribution and can be more suitable to specifically explain the relationship between 3-OH FA distribution and pH values. This point will be clarified in the revised version of the manuscript.

L467: Please refer to and compare with the newer calibration studies. Several have been publishedsince 2012, but at least use De Jonge et al., 2014 that includes the 5- and 6-methylisomers.

Author comment : As previously suggested, the CBT' index will be used instead of the CBT and will allow a direct comparison with the newer calibration studies (e.g. De Jonge et al., 2014). This reference and comparison will be added in the revised version of the manuscript.

L490: replace similarly by also L492 (here and elsewhere in the ms): replace'prediction interval' by 'confidence interval'.

Author comment :This will be corrected.

L542ff: How does the MBT'5me relationwith MAAT from this study compare to the one in the global surface soil dataset? Doesit have the same slope and intercept?

Author comment : The relationship between the MBT'5Me and the MAAT obtained in this study (MAAT = 24.5 × MBT'5Me -4.78) follows a similar trend as the global calibration proposed by De Jonge et al. 2014 (MAAT = 31.45 × MBT'5Me - 8.57). In the next version of this manuscript, a direct statistical comparison between these two calibrations will be proposed.

General: Replace Damsté by Sinninghe Damsté Author comment : This will be made.

Figures 1 and 2: I'm not sure how useful it is to show the relative distributions of 3OH-FAs and brGDGTs, especially since every site represents an elevation gradient along which a large variation in temperature (and thus biomarker composition) is expected.Instead, display the proxy values with elevation (or completely delete them, as the proxies are already plotted against pH and MAAT in Figs 4-6).

Author comment : These figures showing the relative distributions of 3OH-FAs and brGDGTs are useful to understand the natural variability of these compounds along the different transects. This comprehension is necessary to understand the PCAs calculated with the fractional abundances of 3-OH FAs and brGDGTs.

Figures 4 and 5 and 6. Can you add the regressions, or at least the regression coefficient (R2) of eachsignificant regression to the panels?

[Figure]

Figure 9: The y-axis of panel a does not have atitle

Author comment : These figures will be modified and corrected according to these comments. 

References Bhat, U.R., Carlson, R.W., 1992. A new method for the analysis of amide-linked hydroxy fatty acids in lipid-As from gram-negative bacteria. Glycobiology 2, 535–539.

Coffinet, S., Huguet, A., Pedentchouk, N., Bergonzini, L., Omuombo, C., Williamson, D., Anquetil, C., Jones, M., Majule, A., Wagner, T. and Derenne, S., 2017: Evaluation of branched GDGTs and leaf wax n-alkane $\delta$2H as (paleo) environmental proxies in East Africa, Geochimica et Cosmochimica Acta, 198, 182–193, doi:10.1016/j.gca.2016.11.020.

De Jonge, C., Hopmans, E.C., Zell, C.I., Kim, J.-H., Schouten, S., Sinninghe Damsté, J.S., 2014. Occurrence and abundance of 6-methyl branched glycerol dialkyl glycerol tetraethers in soils: Implications for palaeoclimate reconstruction. Geochimica et Cosmochimica Acta 141, 97–112.

Dearing Crampton-Flood, E., Tierney, J.E., Peterse, F., Kirkels, F.M.S.A., Sinninghe Damsté, J.S., 2020. BayMBT: A Bayesian calibration model for branched glycerol dialkyl glycerol tetraethers in soils and peats. Geochimica et Cosmochimica Acta 268, 142–159.

Dunkley Jones, T., Eley, Y.L., Thomson, W., Greene, S.E., Mandel, I., Edgar, K., Bendle, J.A., 2020. OPTiMAL: a new machine learning approach for GDGT-based palaeothermometry. Climate of the Past 16, 2599–2617.

Huguet, A., Coffinet, S., Roussel, A., Gayraud, F., Anquetil, C., Bergonzini, L., Bonanomi, G., Williamson, D., Majule, A., Derenne, S., 2019. Evaluation of 3-hydroxy fatty acids as a pH and temperature proxy in soils from temperate and tropical altitudinal gradients. Organic Geochemistry 129, 1–13.

Lecointre G. and Guyader H. L. 2006. The Tree of Life: A Phylogenetic Classification., Harvard 925 University Press

Margesin, R., Jud, M., Tscherko, D., Schinner, F., 2009. Microbial communities and activities in alpine and subalpine soils: Communities and activities in alpine and subalpine soils. FEMS Microbiology Ecology 67, 208–218.

Naafs, B.D.A., Gallego-Sala, A.V., Inglis, G.N., Pancost, R.D., 2017a. Refining the global branched glycerol dialkyl glycerol tetraether (brGDGT) soil temperature calibration. Organic Geochemistry 106, 48–56.

Naafs, B.D.A., Inglis, G.N., Zheng, Y., Amesbury, M.J., Biester, H., Bindler, R., Blewett, J., Burrows, M. A., del Castillo Torres, D., Chambers, F.M., Cohen, A.D., Evershed, R.P., Feakins, S.J., Gałka, M., Gallego-Sala, A., Gandois, L., Gray, D.M., Hatcher, P.G., Honorio Coronado, E.N., Hughes, P.D.M., Huguet, A., Könönen, M., Laggoun-Défarge, F., Lähteenoja, O., Lamentowicz, M., Marchant, R., McClymont, E., Pontevedra-Pombal, X., Ponton, C., Pourmand, A., Rizzuti, A.M., Rochefort, L., Schellekens, J., De Vleeschouwer, F., Pancost, R.D., 2017b. Introducing global peat-specific temperature and pH calibrations based on brGDGT bacterial lipids. Geochimica et Cosmochimica Acta 208, 285–301.

Parker, J.H., Smith, G.A., Fredrickson, H.L., Vestal, J.R., White, D.C., 1982. Sensitive assay, based on 959 hydroxy fatty acids from lipopolysaccharide lipid A, from Gram negative bacteria in sediments. 960 Applied and Environmental Microbiology 44.

Russell, J.M., Hopmans, E.C., Loomis, S.E., Liang, J., Sinninghe Damsté, J.S., 2018. Distributions of 5- and 6-methyl branched glycerol dialkyl glycerol tetraethers (brGDGTs) in East African lake sediment: Effects of temperature, pH, and new lacustrine paleotemperature calibrations. Organic Geochemistry 117, 56–69. Siles, J.A., Margesin, R., 2016. Abundance and Diversity of Bacterial, Archaeal, and Fungal Communities Along an Altitudinal Gradient in Alpine Forest Soils: What Are the Driving Factors? Microbial Ecology 72, 207–220.

Tierney, J.E., Tingley, M.P., 2014. A Bayesian, spatially-varying calibration model for the TEX86 proxy. Geochimica et Cosmochimica Acta 127, 83–106.

Véquaud, P., Derenne, S., Anquetil, C., Collin, S., Poulenard, J., Sabatier, P., Huguet, A., 2021. Influence of environmental parameters on the distribution of bacterial lipids in soils from the French Alps: Implications for paleo-reconstructions. Organic Geochemistry 104194.

Wang, C., Bendle, J., Yang, Y., Yang, H., Sun, H., Huang, J. and Xie, S., 2016: Impacts of pH and temperature on soil bacterial 3-hydroxy fatty acids: Development of novel terrestrial proxies, Organic Geochemistry, 94, 21–31, doi:10.1016/j.orggeochem.2016.01.010.

Wang, C., Bendle, J.A., Zhang, H., Yang, Y., Liu, D., Huang, J., Cui, J., Xie, S., 2018. Holocene temperature and hydrological changes reconstructed by bacterial 3-hydroxy fatty acids in a stalagmite from central China. Quaternary Science Reviews 192, 97–105.

Wang, H., An, Z., Lu, H., Zhao, Z., Liu, W., 2020. Calibrating bacterial tetraether distributions towards in situ soil temperature and application to a loess-paleosol sequence. Quaternary Science Reviews 231, 106172.

Wilkinson, S.G. 1988 Gram-negative bacteria. In: Ratledge C., Wilkinson S.G. (Eds), Microbial Lipids, 1063 vol. 1. Academic Press, New York, pp. 199-488.

Yang, H., Lü, X., Ding, W., Lei, Y., Dang, X. and Xie, S., 2015: The 6-methyl branched tetraethers significantly affect the performance of the methylation index (MBT′) in soils from an altitudinal transect at Mount Shennongjia, Organic Geochemistry, 82, 42–53, doi:10.1016/j.orggeochem.2015.02.003.

Yang, Y., Wang, C., Bendle, J.A., Yu, X., Gao, C., Lü, X., Ruan, X., Wang, R., Xie, S., 2020. A new sea surface temperature proxy based on bacterial 3-hydroxy fatty acids. Organic Geochemistry 141, 103975.

Zelles, L., 1999. Fatty acid patterns of phospholipids and lipopolysaccharides in the characterisation of microbial communities in soil: a review. Biology and Fertility of Soils 29, 111–129.

---

## Author Comment (AC2) · 18 Feb 2021

We thank the anonymous reviewer for these comments. A detailed list of changes and arguments answering to the different comments is provided below.

Reviewers' comments:

Reviewer #2: Review

-I have finished the review of the manuscript "development of global temperature and p calibrations based on bacterial 3-hydroxy fatty acids in soils", submitted to biogeosciences by Pierre Vequaud and co-authors. In general, the authors present a good

contribution to the developing field of 3-OH fatty acids, by compiling several published and novel altitudinal gradients. The performance of the local and compiled linear regressions are compared with that of brGDGT lipids, for those sites where 5- and 6-methyl compounds were separated (and where the MBT'5ME could thus be calculated). This is an interesting dataset, and a necessary next step in the development of 3-OH fatty acids as temperature proxies.

Author comment: We would like to thank the reviewer for his positive comments and acknowledging the work made to evaluate 3-OH FAs as temperature and pH proxies on a global scale.

- However, the performance of the three additional models (multiple linear regression, random forest and k-nearest neighbor), that move away from the previously established RIAN, RAN15 and RAN17 ratios, are not evaluated thoroughly. There is no explanation of what the models represent, and why they allow a better correlation between the3-OH fatty acids and MAAT or pH, compared to the simple linear regression. There is for instance no analysis of residuals, and no indication of maximum reconstructed temperatures. I would recommend the authors to look at a recent calibration paper (fi Dearing-Crampton Flood, 2020), to see what the state-of-the-art in lipid proxy calibration is. As it stands, the authors will have a difficult time convincing reader that the more complex models are the preferred choice when doing 3-OH FA-based climate reconstructions.

Author comment: As suggested by the reviewer, the detailed characteristics of the models proposed in this work will be added. Thus, a study of the residuals of these models will be proposed, and in addition to the $R^2$ and RMSE values, a table of the variance values of the residuals, model bias and upper/lower limits of pH and MAAT estimates will be presented (taking as an example Table 2 in the paper by Dearing-Crampton Flood, 2020). The advantage of the three models of the present manuscript over simple linear regression models is to take into account the whole suite of bacterial lipids (in our case C10-C18 3-OH FAs) to estimate MAAT and pH values, which

allows overcoming the limitations related to the use of a single index. In addition, the Random Forest and k-NN models are non-parametric and non-linear models, allowing to take into account the potential non-linear influences of MAAT/pH on the fractional abundances of 3-OH FAs. The functioning of the different models will be clarified in the revised manuscript, with more didactic explanations. Quantitative evaluations of the influence of each 3-OH FA on the MAAT/pH calibrations will also be proposed, thus providing mechanistic details on the models. These explanations should highlight the advantages of these new models and especially their ease of use through the web application provided in this study.

-When developing a MAAT-calibration, the authors should also mention the following weakness: there is currently only 3 samples with MAAT > 15 âĚĞD ĚĞC in the dataset (maximum temperature: +- 20 âĚĞDCĚĞ ). Do these samples influence the MAAT calibration disproportionally?

Author comment: There are > 30 samples collected in locations with MAAT > 15 °C, especially from Mt. Rungwe (Tanzania) and Peruvian Andes. Nevertheless, as explained in L. 305 "the training phase required for the random forests, k-NN and multiple linear regression was performed on 75% of the sample set with an iteration of ten cross-validations per model. Data selection was performed randomly on the dataset but with a stratification modality according to the MAAT or the pH to limit the impact of extreme values". Thus, the developed models include 25 % of the dataset, considered as independent from the training dataset. The test dataset, randomly chosen, indeed contains only 3 samples with MAAT > 15 °C. Nevertheless, the cross validations and stratification modalities allow to reduce the influence of extreme values. To go further, the models proposed in this study were tested again without the extreme temperature values. This did not change the proposed MAAT estimates. To take into account the reviewers' comment, a discussion will be added in the revised version of this manuscript on the potential influence of "extreme" MAAT values on our models.

Minor comments:

L 91. Include 'bacterial' before 'lipids'. L 129. 'more developed statistical approach', perhaps rephrase as 'further development of the statistical approaches'?

Author comment: This sentence will be modified as suggested.

L 148. For easy comparison: include the altitudinal, pH and MAAT rangesnof the previously published transects. L 200-201. Soil sensors have been used, but the calibration is done with mean annual air temperature (MAAT) instead. As the soil sensor data is not used or discussed further, I'd remove their mention here, and use the MAAT for this site as well. L 322 (and further in the manuscript). As far as I can see, the supp. tables referred to are not present. Please include these tables in a revised version.

Author comment: pH and MAAT ranges will be added for all the previously published transects. In the supplementary tables we will provide the new data obtained from Mt Pollino, Shegyla, Peruvian and Chilean Andes. For the other previously published transects, the reader will be directed towards the references of the corresponding papers. This point will be clarified in the revised version of this manuscript. We will remove the mention of the soil sensors. We will check that the supp. Tables are included in the revised version.

L 541. Perhaps the authors can refer here to De Jonge et al., 2019, who argue that temperature can modulate the soil bacterial composition, and the dependency between MBT'5ME and soil temperature?

Author comment: This reference will be added to the manuscript.

L551. Here the authors should explain better how 'more complex models' allow to take the complexity of each site into account. At this point, in the manuscript, we have observed that the linear dependency between the 3-OH fatty acids is better on a local scale, than on a global scale. What mechanism do the authors propose for this, and how can a more complex model correct for it?

Author comment: It should be highlighted that the relative abundances of the 3-OH

FAs were linearly correlated with MAAT/pH at a local scale, but only along some of the transects. The absence of linear relationships at a local scale for part of the sites, and more generally at a global scale, may at least "be partly due to the heterogeneity of soils encountered along a given altitudinal transect, representing specific microenvironments and to the large diversity of bacterial communities in soils from different elevations (Siles and Margesin, 2016)", as specified in the manuscript (L. 435). The models proposed in this study for MAAT/pH reconstruction are based on the whole suite of 3-OH FAs (C10-C18) instead of indices. They allow better capturing the complexity of the microoenvironments found along each transect and the variability of the lipid distribution, in contrast with linear models. In addition, two of the three models presented in this paper (k-NN and random forest) are both non-parametric and non-linear and present the advantage of taking into account the potential non-linear relationships between the relative abundances of 3-OH FAs and environmental parameters (here MAAT / pH).

L567. Have the authors done a selection of the fatty acids that are necessary for the model? For instance, forward selection or reverse selection? Does the model not suffer from overfitting? Is there any fatty acids that are generally present in low abundance (and can thus be absent in geological archives) that are important for the regression? If yes, is it prudent to include these low abundance compounds in the model as well?

Author comment: In order to prevent any overfitting of the models, several steps were followed and will be better explained in the next version of the manuscript: (1) Cross-validation of the training dataset to find the optimal combination of 3-OH FA relative abundances and maximize the model performance. (2) No pre-processing (i.e.no pre-selection) of the individual 3-OH FAs. All the 3-OH FA homologues, whatever the abundance, were used in the different models to keep the maximum variability and take into account the specificity and complexity of each altitudinal transect. (3) Testing of the model on a randomly selected dataset (i.e. test dataset) that was not used to build the model. In order to clarify the importance of the different 3-OH FAs in the

models, figures presenting the variance of the different homologues in each model will be proposed in the next version of the manuscript.

L 587. The random forest model and k-NN model need to be explained much better before the results are presented. What are they based on and how do they compare variability in the lipid distribution with the MAAT? L 593. Same comments as at L 567.

Author comment: As explained above, each model will be better explained, and the results of these different new models will be directly and statistically compared to those based on previously defined indices (RAN15, RAN17 and RIAN).

L 651. Can the authors comment on the probable source of the 3-OH fatty acids in the speleothem? Can we assume that all lipids are derived from the soil, or is there a (variable) proportion produced in the cave environment as well?

Author comment: Wang et al. (2018), who analyzed 3-OH FAs in this speleothem, suggested that these compounds are largely derived from the overlying soils based on geochemical and microbiological analyses. Indeed, as specified by these authors, "the broad similarity of 3-OH-FA distributions in the overlying soils and stalagmites, supported by the site-specific analyses of bacterial diversity and transport pathways, supports a major contribution of 3-OH-FAs from Gram-negative bacteria dwelling in the overlying soils to the stalagmite samples". As also stated by Wang et al. (2018), this does not totally exclude that a proportion of the lipids may be derived from the cave ecosystem. This point will be mentioned in the revised manuscript.

Refs: Dearing Crampton-Flood E., Tierney J. E., Peterse F., Kirkels F. M. S. A. and Sinninghe Damsté J. S. (2020) BayMBT: A Bayesian calibration model for branched glycerol dialkyl glycerol tetraethers in soils and peats. Geochimica et Cosmochimica Acta 268, 142–159. De Jonge C., Radujkovi′c D., Sigurdsson B. D., Weedon J. T., Janssens I. and Peterse F. (2019) Lipid biomarker temperature proxy responds to abrupt shift in the bacterial community composition in geothermally heated soils. Organic Geochemistry, S0146638019301275.

References

Dearing Crampton-Flood, E., Tierney, J.E., Peterse, F., Kirkels, F.M.S.A., Sinninghe Damsté, J.S., 2020. BayMBT: A Bayesian calibration model for branched glycerol dialkyl glycerol tetraethers in soils and peats. Geochimica et Cosmochimica Acta 268, 142–159.

Siles, J.A., Margesin, R., 2016. Abundance and Diversity of Bacterial, Archaeal, and Fungal Communities Along an Altitudinal Gradient in Alpine Forest Soils: What Are the Driving Factors? Microbial Ecology 72, 207–220.

Wang, C., Bendle, J.A., Zhang, H., Yang, Y., Liu, D., Huang, J., Cui, J., Xie, S., 2018. Holocene temperature and hydrological changes reconstructed by bacterial 3-hydroxy fatty acids in a stalagmite from central China. Quaternary Science Reviews 192, 97–105.

---

## Author Response (AR1)

We thank the anonymous reviewers for these comments. A detailed list of changes and arguments answering to the different comments is provided below. The line numbers are those of the final version of the manuscript (without track change).

**Reviewers' comments:**

**Reviewer #1: Review**

**Summary**

Véquaud and co-authors have compiled a large set of soils along elevation gradients in an attempt to determine global relationships of 3-OH FAs with MAAT and soil pH. They find that there is a global relationship with MAAT, but that relationships are better on a local scale. They then use several statistical methods to quantify the relationships and to develop a temperature proxy based on 3-OH FAs. I appreciate the effort of the authors to compile this relatively large dataset and their attempt to assess the temperature and pH sensitivity of these compounds on a global scale. I do however have a few concerns and suggestions to further improve this work before I can recommend it for publication in Biogeosciences.

*We would like to thank the reviewer for his comments and for acknowledging the work made to evaluate the temperature and pH sensitivity of 3-OH FAs on a global scale. It should be highlighted that half of the 3-OH FA and brGDGT data presented in the manuscript are new (those corresponding to 4 altitudinal transects, i.e. Mt. Pollino, Italy; Mt. Shegyla, Tibet; Peruvian and Chilean Andes). Our new lipid data were combined with those previously published (Mt. Shennongjia, China, Yang et al., 2015 and Wang et al., 2016; Mt. Rungwe, Tanzania, Coffinet et al., 2017 and Huguet et al., 2019; Mt. Majella, Italy, Huguet et al., 2019; French Alps, Véquaud et al., 2021). Gathering these "old" and new data allowed establishing a large dataset which was used to evaluate the applicability of 3-OH FAs as MAAT and pH proxies in soils and to establish a comparison with the more established brGDGT proxies. Previous papers presenting brGDGT and 3-OH FA calibrations similarly rely on a combination of previously and newly acquired data (e.g. De Jonge et al., 2014; Naafs et al., 2017a; Huguet et al., 2019; Dearing Crampton-Flood et al., 2020).We also appreciate that the reviewer did not question the way the data were acquired, processed and interpreted.*

**Major comments:**

-BrGDGT data: Next to 3-OH FA data, the authors also present brGDGT data from the same samples. However, I fail to see the added value of the brGDGT data to this manuscript, and I thus suggest leaving this data out.

*We kindly disagree with the reviewer here. Indeed, to date, brGDGTs are the only organic proxies available for temperature and pH reconstruction in terrestrial settings. Therefore, as mentioned in the introduction of the manuscript, "the development of new molecular proxies, independent of and complementary to brGDGTs, is essential to improve the reliability of paleotemperature (and pH) reconstructions in such settings". 3-OH FAs could provide such a proxy. As a result, 3-OH FAs and brGDGTs have to be concomitantly analyzed to assess their reliability and complementarity as independent temperature and pH proxies. Such a comparison cannot be made if brGDGT data are not presented and cannot be used as "a reference proxy". The interest of brGDGT data in this manuscript was further clarified in the revised version of the manuscript (end of the introduction, L. 139):*

[Figure]

*"As brGDGTs are the only microbial organic proxies which can be used for temperature and pH reconstructions in terrestrial settings, they can serve as a reference proxy to understand the temperature and pH dependency of 3-OH FAs analyzed in the same dataset, taking into account the large uncertainties persisting in the global temperature/pH brGDGT calibrations (De Jonge et al., 2014; Dearing Crampton-Flood et al., 2020). 3-OH FAs and brGDGTs have thus been concomitantly analyzed to assess their reliability and complementarity as independent temperature and pH proxies".*

*It should also be noted that all the previous papers dealing with the influence of environmental parameters on 3-OH FA distribution in soils (Wang et al., 2016; Huguet et al., 2019; Véquaud et al., 2021) made a comparison between brGDGT-derived and 3-OH FA-derived proxies. Our approach is thus consistent with these papers.*

a) the focus of the paper is on assessing the temperature and pH dependency of 3-OH FAs, which has nothing to do with brGDGTs.

*We agree with the reviewer that the main aim of this study is to evaluate the influence of pH and MAAT on 3-OH FA distribution in globally distributed topsoils. Nevertheless, it would be inappropriate not to compare these results with those derived from brGDGTs, the only other organic temperature/pH proxies terrestrial environments, as mentioned above.*
*The comparison between brGDGT and 3-OH FA data is necessary, as it highlights the similarities and differences between these two independent families of molecules in terms of response to temperature and pH. This also shows the link between this study and previous ones dealing with the development of organic temperature/pH proxies in terrestrial settings. The different proxies and related studies should not be disconnected from each other.*

b) most of the brGDGT datasets presented here are already published elsewhere.
*We kindly disagree with the reviewer, as only half of the brGDGT dataset is already published. The other half of the dataset, corresponding to 4 altitudinal transects (Mt. Pollino, Italy; Mt. Shegyla, Tibet; Peruvian and Chilean Andes), is presented here for the first time.*

-In addition, the brGDGT data for two of the elevation transects (Rungwe and Shennongjia) are based on 'old' chromatography methods that do not separate the 5- and 6-methyl brGDGT isomers. Note that these datasets should not be directly compared with 'new' brGDGT data for the other transects, as MAATs and pH are derived based on different equations. The 'old' MBT and CBT indices both contain compounds that appear to have a 6-methyl isomer (e.g. IIa and IIIa in the MBT index, and IIa and IIb in the CBT index). This makes it impossible to determine whether a relative change in the peak area of one of these compounds is driven by the 5- or by the 6-methyl isomer. Especially in the case of MBT this is problematic, as the occurrence of 6-methyl brGDGTs is linked to changes in pH and not MAAT (De Jonge et al., 2014). The authors may be unaware, but 'new' brGDGT data for Mt. Shennongjia has been published by Yang et al., 2015, who, in fact, state that 6-methyl brGDGTs significantly affect the performance of the MBT' in this transect, proving my point that the use of 'new' data is important (if at all, in case of this manuscript).

*We totally agree with the reviewer on the importance of separating 5- and 6-methyl brGDGT isomers and never said the opposite in our paper. The new data presented in this study were all obtained using two silica columns, allowing the separation of these isomers. Similarly,*

[Figure]

previously published brGDGT data from the French Alps (Véquaud et al., 2021) and Mt. Majella (Huguet et al., 2019) were also acquired with this new methodology.

Regarding Mt. Shennongjia, we agree with the reviewer that 5- and 6-methyl brGDGTs were indeed separated in the Organic Geochemistry paper published by Yang et al. (2015). However, the relative abundances of the individual 5- and 6-methyl brGDGTs were not directly provided in this paper, which prevented us to use them. Following up the review, the authors kindly sent us these data upon request. These abundances and the values of MBT'$_{5Me}$ index can now be used in our paper. This point was clarified in the revised manuscript:

"The relative abundances of brGDGTs were compared between the same transects as for 3-OH FAs, representing a total of 168 samples. The 5- and 6-methyl isomers were separated in most of the samples (Fig. 2; Sup. Table 2, except in older dataset, i.e. soils from Mt. Rungwe (Coffinet et al., 2014, 2017)". L.366

On the whole, data from 7 out of 8 transects investigated in this study were obtained using the "new" chromatography method allowing the separation of 5- and 6-methyl brGDGTs. All these data can be directly compared and exploited (Figs. 2 and 3).

Regarding the data from Mt. Rungwe, they were obtained using the "old" chromatography method relying on a cyano column. As 5- and 6-methyl brGDGTs were not separated, this dataset cannot be directly compared with the 7 others. The brGDGT data were therefore not considered further in the manuscript.

- Ecology of 3-OH FA producers: There are several points in the manuscript where the authors state that a better understanding of the adaptation mechanisms of 3-OH FA producing organisms to changing environmental conditions may improve the proxies (e.g. L484), or that the use of (a) subset(s) of 3-OH FAs could results in better relations with MAAT or pH (as suggested for the RIAN-pH relationship). I feel that there is a missed opportunity here and improvements should be made in a revised version. If it is known that 3-OH FAs are produced by Gram negative bacteria, what can we tell about their occurrence, response to environmental change, etc? Or is this group simply too big to say anything sensible?

3-OH FAs are produced by Gram-negative bacteria which, as suspected by the reviewer, are a highly diverse, non-monophyletic and ubiquitous group of microorganisms represented by numerous genera (Lecointre and Guyader, 2006). This explains the large genetic and biochemical differences between the various Gram-negative bacteria. The numerous genera of Gram-negative bacteria are characterized by highly diverse lipid profiles, with different relative abundances of the C$_{10}$-C$_{18}$ 3-OH FAs, and by the fact that all the homologues are not biosynthesized by all the strains (Wilkinson et al., 1988). Thus, the 3-OH FA lipid distribution in soils is highly dependent on the diversity of Gram-negative bacterial species (Parker et al., 1982; Bhat and Carlson, 1992; Zelles, 1999), which may vary with altitude (Margesin et al., 2009; Siles and Margesin, 2016). Considering these different facts, it is difficult to draw out a general hypothesis on the response of Gram-negative bacteria and their associated lipids to environmental changes. A multiplicity of environmental parameters, including MAAT and pH, may have an influence on 3-OH FA lipid distribution. The latter can reflect changes in the diversity of bacteria communities (e.g. Siles and Margesin, 2016) and/or adaptation of a constant community.

- Proxy model development From the brGDGT and other biomarker proxy studies we have learned that many of the transfer functions based on linear regressions suffer from

regression dilution (e.g. Tierney and Tingley, 2014; Naafs et al., 2017; Dearing- Crampton-Flood et al., 2020). I miss an assessment of this aspect for the 3-OH FA proxies in this manuscript, which are all based on methods that involve linear regressions.

*We agree with the reviewer that a weak point of all methods based on regression methods is that they suffer from the phenomenon of regression dilution. That is why proposing alternative models that are not based on linear regression methods, such as two of those proposed in this study (random forest and k-NN algorithms), represent a major advantage. This aspect is now clearly mentioned in the revised manuscript (beginning of section 4.2., L. 577) and better assessed.*

- I would also like to point out that there is a general move towards using Bayesian statistics to derive proxies. This approach will allow you to treat your biomarker-based index as dependent variable in the calibration model whilst accounting for possible errors in the measurement of both MAAT and your biomarker index (see the explanation in the introduction of Dearing Crampton-Flood et al., 2020). I encourage the authors to look into this method and consider its use for their purpose.

*Several methods were recently used to derive calibrations from organic proxies, such as multivariate regression (Russell et al., 2018), Deming regression (Naafs et al., 2017) , Bayesian linear regressions models (Tierney and Tingley, 2014; Dearing Crampton-Flood et al., 2020) or machine-learning models (Dunkley Jones et al., 2020). None of these methods excludes or prohibits the use of the others. Independent models should be used for the development of environmental calibrations, as each of them has its own advantages and drawbacks. The different models have to be seen as complementary.*

*As mentioned by the reviewer, Bayesian statistics were recently used to develop new calibration models, e.g. with brGDGTs (Dearing Crampton-Flood et al., 2020). The robustness of the Bayesian models relies on the fact that they consider a given index (for example, the MBT'$_{5Me}$ for brGDGTs) as the variable dependent on environmental parameters. This is realistic, as lipids are produced in response to the variations of environmental parameters. This model also avoids regression dilution phenomena, in contrast with most of the models based on linear regressions, as mentioned by the reviewer.*

*Nevertheless, the Bayesian method is always restricted to the use of an index, which allows the reconstruction of a given environmental parameter (for example temperature). This represents a limitation, as the relative distribution of bacterial lipids can be concomitantly influenced by several environmental parameters (e.g. Véquaud et al., 2021). In contrast, using bacterial relative abundances rather than a single index in models appears less restrictive, and more representative of the environmental complexity. The different models (multiple regression, k-NN method, Random Forest) presented in this paper allow overcoming the limitations related to the use of a single index, as they take into account the whole suite of bacterial lipids (here C$_{10}$-C$_{18}$ 3-OH FAs) to estimate MAAT and pH values.*

*In addition, the Bayesian model is parametric and it only takes into account linear relationships. Conversely, the k-NN and Random Forest models used in this study are non-parametric and can take into account non-linear environmental influences, in line with the intrinsic complexity of the environmental settings. This helps in improving the reliability of the models.*

*Moreover, the models based on machine learning algorithms are built on a proportion of the total dataset (randomly defined) and then tested on the rest of the dataset, considered as*

[Figure]

*independent. Such an approach improves the robustness of the models. As the Bayesian model, the k-NN and random forests avoid the phenomenon of regression dilution.*

*In conclusion, we chose the k-NN and random forest models in the present study, as they present major advantages for the development of robust calibrations between bacterial lipid distribution and environmental parameters, as explained above. Such a choice is justified and discussed in detail in the revised manuscript (section 4.2, L. 596). The different models chosen in our study are also more clearly described in the material and method section (paragraph 2.3.) Once again, this does not exclude the use of other independent and complementary models, such as those based on Bayesian statistics, but this is beyond the scope of the present study.*

- Application of the 3-OH FA proxies The authors have chosen a stalagmite as test archive for their proxy. but I really do not think that a speleothem is a suitable archive to test proxies that are based on the occurrence and behavior of 3-OH FAs in soils. Speleothems and soils are completely different environments, and it is no way guaranteed, let alone tested that the sources of 3-OH FAs in both environments are the same, let alone their relationship with temperature and/or pH. I would either leave this example out, or find another (soil!) archive to test their proxies.

*Testing the performance and validity of the proposed global models on an existing archive is essential, as the calibration models are developed from modern samples to then provide robust and reliable paleoreconstructions. Most of the recent studies proposing new global or local MAAT calibrations, based on either brGDGTs or 3-OH FAs, adopted this approach and applied their models to a well-constrained archive to assess its robustness and accuracy (e.g. Naafs et al., 2017; Dearing Crampton-Flood et al., 2020; Wang et al., 2020; Yang et al., 2020).*

*As a result, we developed MAAT calibrations from 3-OH FA distribution in soils using machine learning models (k-NN and random forest algorithms) and then tested and compared these calibrations using a speleothem archive. We agree with the reviewer that a soil archive would have even better fitted to test our new calibrations and that soils and speleothems are different terrestrial settings. Nevertheless, the validation of the terrestrial models we propose requires using a well-known archive, for which paleoclimatic and paleoenvironmental data are available. This allows constraining the climatic/environmental conditions at the time of the reconstruction based on a multi-proxy approach.*

*So far, the only 3-OH FA based paleorecord available in the literature is the Chinese speleothem previously investigated by Wang et al. (2018), which was the object of previous paleostudies, thus providing a context for the interpretation of the MAAT data. Moreover, Wang et al. (2018) suggested that the 3-OH FAs in the investigated speleothem are largely derived from the overlying soils based on the "broad similarity of 3-OH FA distributions in the overlying soils and stalagmites" as well as "site-specific analyses of bacterial diversity and transport pathways". Thus, using 3-OH FA-based soil calibrations to reconstruct MAAT/pH from this speleothem is appropriate. By the way, a local Chinese MAAT calibration obtained after the analysis of 3-OH FAs in 26 soil samples (Wang et al., 2016) was previously applied to this record. In the present study, we compare the MAAT estimates provided by our global soil calibrations with those previously obtained using the local calibration by Wang et al. (2016).*

*In conclusion, it seems reasonable to us to test our new models on a terrestrial archive (even though this is not a soil) to show their potential and limits, as the ultimate goal of such calibrations is to be used in the framework of paleoenvironmental studies. Such a choice is justified in detail the revised version of the manuscript (beginning of section 4.3., L. 721) before the comparison of the different models to avoid any confusion in the mind of the reader.*

[Figure]

*The three MAAT models, which were carefully checked, were then applied to the speleothem and their results compared in Supp. Fig. 6 and section 4.3.1. As the global random forest method appeared to be the most reliable of the three models proposed, it was then compared with the local RAN₁₅ calibration proposed by Wang et al., 2016 (Fig. 10 and section 4.3.2.).*

- Data compilation I notice that the authors have included one dataset that is still under review in Organic Geochemistry (Véquaud et al.,). In case this manuscript is accepted first (actually, the data is already published online as part of this discussion paper), is there any novelty left for the manuscript in OG? Aren't you shooting yourself in the foot here?

*We thank the reviewer for this concern. However, there is no problem since the OG manuscript is in press (Véquaud et al., 2021) and available online (https://doi.org/10.1016/j.orggeochem.2021.104194).*
*The OG paper aimed to investigate in detail the influence of environmental parameters on 3-OH FA distribution in soils from the French Alps. The present paper notably includes the data from the OG paper to create a large dataset based on soils from all over the world.*

**Minor comments:**

L70: the use of Eglinton and Eglinton as a reference to proxies based on membrane lipids is technically not incorrect, but it seems more appropriate to use e.g. Schouten et al., 2002, Weijers et al., 2007 ('they were first'), or the re-view paper of Schouten et al., 2013.

*This was corrected in the revised version of the manuscript : "Some of the existing proxies are based on membrane lipids synthesized by certain microorganisms (Eglinton and Eglinton, 2008; Schouten et al., 2013)." L.70*

L79: For aquatic environments, references to Peterse et al., 2009 (marine brGDGTs) and Tierney and Russell, 2009 and Sinninghe Damsté et al., 2009 (lacustrine brGDGTs) are better suited ('they were first').

*These references were added in the revised version of this paper: "…and aquatic environments ( Peterse et al., 2009; Tierney and Russell, 2009; Sinninghe Damsté et al., 2009; Loomis et al., 2012; Peterse et al., 2015; Weber et al., 2015)"L.80*

L98: explain what RAN15 and RAN17 stand for prior to using the abbreviation.

*This was corrected: "The analysis of 3-OH FAs in soils showed that the ratio of C₁₅ or C₁₇ anteiso 3-OH FA to normal C₁₅ or C₁₇ 3-OH FA (RAN₁₅ and RAN₁₇ indices respectively)…" L.99*

L109: re-place calibration by relation

*This was amended. L.109*

L112: explain RIAN prior to using the abbreviation.

*This was corrected: "Another index, defined as the cologarithm of the sum of anteiso and iso 3-OH FAs divided by the sum of normal homologues (RIAN index),…"L.112*

[Figure]

L130: replace 'collected along globally distributed' by 'determined in several elevation transects'

*This sentence was corrected following this comment: "3-OH FA distribution from 54 soils was determined in several globally distributed altitudinal transects…" L.130*

L134: This sentence is not really grammatically correct. Rewrite to somethinglike 'even though brGDGT-based MAAT and pH reconstructions still have a relatively large uncertainty, they can serve as a reference to test the temperature dependency of 3OH FAs analyzed in the same dataset.'

*This part of the text was totally rephrased (L. 139):*
*"As brGDGTs are the only microbial organic proxies which can be used for temperature and pH reconstructions in terrestrial settings, they can serve as a reference proxy to understand the temperature and pH dependency of 3-OH FAs analyzed in the same dataset, taking into account the large uncertainties persisting in the global temperature/pH brGDGT calibrations (De Jonge et al., 2014; Dearing Crampton-Flood et al., 2020). 3-OH FAs and brGDGTs have thus been concomitantly analyzed to assess their reliability and complementarity as independent temperature and pH proxies."*

L137: use either 'the 3OH FA distribution'or '3OH FA distributions'.

*This was corrected. L.135*

L177: what is a 'fir'?

*Abies forrestii is a species of conifer in the family of Pinaceae, commonly referred as Forrest's fir. This sentence was clarified in the revised version. L.183*

L230: can you add the compound that was used as an internal standard?

*The internal standard was a deuterated $C_{14}$ 3-OH FA, the 3-hydroxytetradecanoic acid, 2,2,3,4,4-d5 (Sigma-Aldrich, France). This was specified in the revised version of the paper. L.234*

L274: Is there a reason why the latest definition of the CBT' index by De Jonge et al., 2014 was not used (their Eq. 10)? This index takes the separation of 5- and 6-methyl isomers into account and also has a lower RMSEcompared to the one of Peterse et al., 2012 that is based on data generated with the'old' chromatography method. If not, I would recommend using the CBT' index.

*As suggested by the reviewer, the CBT' was used in this manuscript instead of the CBT index. Nevertheless, the interpretations and conclusions from the CBT' remain the same as with the CBT, as the relations between the two indices and pH are similar.*

L397: replace inertia by variance (inertia means the tendency to remain unchanged, which isnot what you mean here, I presume)

*This was corrected. L.417*

[Figure]

L400: how reliable is 'location' as a factor influencing 3OH FAs? I don't really think that microorganisms would take their coordinates into account when they synthesize 3OH FAs. Instead, I suspect that location represents an indirect parameter that integrates the local environmental conditions at each site. Thus, 3OH FAs are more likely linked to temperature, precipitation, vegetation, pH, whateveron the first PC, so I suggest re-running the PCA without taking location into account.

*We think there is a misunderstanding here. In fact, the PCAs were performed using the fractional abundances of the bacterial lipids (3-OH FAs and brGDGTs) in the total dataset, without taking the location into account. Nevertheless, it is possible to statistically qualify which variables best explain the distribution of the samples in the PCA. It clearly appears that the samples are located in the PCA based on to their location, which reflects the contrasting environmental conditions influencing the 3-OH FA and brGDGT source organisms at each site. This point was clarified in the revised version of this manuscript:*
"Principal component analyses (PCA) were performed on the different soil samples to statistically compare the 3-OH FA/brGDGT distributions along the different altitudinal transects. The fractional abundances of the bacterial lipids (3-OH FAs and brGDGTs) were used for these PCAs, with MAAT, pH and location of the sampling site representing supplementary variables (i.e. not influencing the principal components of the analysis"). *L.292, part 2.3*

L457-459: this sentence seems to be missing words?

*This sentence was corrected as follows: "The distribution of 3-OH FAs varies greatly among Gram-negative bacterial species (Bhat and Carlson, 1992) which may account for the significant variability in RIAN values observed in soils from a given transect". L. 476*

L459: is the lack of a relationbetween the RIAN and pH caused by the use of a linear model, or by the relativelysmall range of soil pH of this dataset? How can you tell? And would another modelbe better suitable to capture this relation? If so, which kind of model?

*In our global dataset, pH values range between 3 and 8. This cannot be considered as a small range of variation. Our hypothesis is that more complex models (non-parametric models, e.g. Random forest, k-NN method) take into account the non-linear influences of environmental parameters on lipid distribution and can be more suitable to specifically explain the relationship between 3-OH FA distribution and pH values. This was discussed in the revised version of the manuscript (section 4.2).*

L467: Please refer to and compare with the newer calibration studies. Several have been publishedsince 2012, but at least use De Jonge et al., 2014 that includes the 5- and 6-methylisomers.

*As previously suggested, the CBT' index will be used instead of the CBT and will allow a direct comparison with the newer calibration studies (e.g. De Jonge et al., 2014). This reference and comparison was added to the revised version of the manuscript.*

L490: replace similarly by also

*This was corrected.*

L492 (here and elsewhere in the ms): replace'prediction interval' by 'confidence interval'.

*This was corrected.*

L542: How does the MBT'5me relationwith MAAT from this study compare to the one in the global surface soil dataset? Doesit have the same slope and intercept?

*The relationship between the MBT'$_{5Me}$ and the MAAT obtained in this study (MAAT = 24.5 × MBT'$_{5Me}$ -4.78) follows a similar trend as the global calibration proposed by De Jonge et al. 2014 (MAAT = 31.45 × MBT'$_{5Me}$ - 8.57). This comparison is proposed in the revised manuscript.*

**General**: Replace Damsté by Sinninghe Damsté
*This was made.*

Figures 1 and 2: I'm not sure how useful it is to show the relative distributions of 3OH-FAs and brGDGTs, especially since every site represents an elevation gradient along which a large variation in temperature (and thus biomarker composition) is expected.Instead, display the proxy values with elevation (or completely delete them, as the proxies are already plotted against pH and MAAT in Figs 4-6).

*These figures are useful to understand the natural variability of the relative abundances of the bacterial lipids along the different transects. They are also necessary to understand the PCAs calculated with the fractional abundances of 3-OH FAs and brGDGTs.*

Figures 4 and 5 and 6. Can you add the regressions, or at least the regression coefficient ($R^2$) of eachsignificant regression to the panels?
*This was made.*

Figure 9: The y-axis of panel a does not have atitle

*This title was added as suggested.*

[Figure]

**References**

Bhat, U.R., Carlson, R.W., 1992. A new method for the analysis of amide-linked hydroxy fatty acids in lipid-As from gram-negative bacteria. Glycobiology 2, 535–539.

Coffinet, S., Huguet, A., Pedentchouk, N., Bergonzini, L., Omuombo, C., Williamson, D., Anquetil, C., Jones, M., Majule, A., Wagner, T. and Derenne, S., 2017: Evaluation of branched GDGTs and leaf wax n-alkane δ2H as (paleo) environmental proxies in East Africa, Geochimica et Cosmochimica Acta, 198, 182–193, doi:10.1016/j.gca.2016.11.020.

De Jonge, C., Hopmans, E.C., Zell, C.I., Kim, J.-H., Schouten, S., Sinninghe Damsté, J.S., 2014. Occurrence and abundance of 6-methyl branched glycerol dialkyl glycerol tetraethers in soils: Implications for palaeoclimate reconstruction. Geochimica et Cosmochimica Acta 141, 97–112.

Dearing Crampton-Flood, E., Tierney, J.E., Peterse, F., Kirkels, F.M.S.A., Sinninghe Damsté, J.S., 2020. BayMBT: A Bayesian calibration model for branched glycerol dialkyl glycerol tetraethers in soils and peats. Geochimica et Cosmochimica Acta 268, 142–159.

Dunkley Jones, T., Eley, Y.L., Thomson, W., Greene, S.E., Mandel, I., Edgar, K., Bendle, J.A., 2020. OPTiMAL: a new machine learning approach for GDGT-based palaeothermometry. Climate of the Past 16, 2599–2617.

Huguet, A., Coffinet, S., Roussel, A., Gayraud, F., Anquetil, C., Bergonzini, L., Bonanomi, G., Williamson, D., Majule, A., Derenne, S., 2019. Evaluation of 3-hydroxy fatty acids as a pH and temperature proxy in soils from temperate and tropical altitudinal gradients. Organic Geochemistry 129, 1–13.

Lecointre G. and Guyader H. L. 2006. The Tree of Life: A Phylogenetic Classification., Harvard 925 University Press

Margesin, R., Jud, M., Tscherko, D., Schinner, F., 2009. Microbial communities and activities in alpine and subalpine soils: Communities and activities in alpine and subalpine soils. FEMS Microbiology Ecology 67, 208–218.

Naafs, B.D.A., Gallego-Sala, A.V., Inglis, G.N., Pancost, R.D., 2017a. Refining the global branched glycerol dialkyl glycerol tetraether (brGDGT) soil temperature calibration. Organic Geochemistry 106, 48–56.

Naafs, B.D.A., Inglis, G.N., Zheng, Y., Amesbury, M.J., Biester, H., Bindler, R., Blewett, J., Burrows, M.A., del Castillo Torres, D., Chambers, F.M., Cohen, A.D., Evershed, R.P., Feakins, S.J., Gałka, M., Gallego-Sala, A., Gandois, L., Gray, D.M., Hatcher, P.G., Honorio Coronado, E.N., Hughes, P.D.M., Huguet, A., Könönen, M., Laggoun-Défarge, F., Lähteenoja, O., Lamentowicz, M., Marchant, R., McClymont, E., Pontevedra-Pombal, X., Ponton, C., Pourmand, A., Rizzuti, A.M., Rochefort, L., Schellekens, J., De Vleeschouwer, F., Pancost, R.D., 2017b. Introducing global peat-specific temperature and pH calibrations based on brGDGT bacterial lipids. Geochimica et Cosmochimica Acta 208, 285–301.

Parker, J.H., Smith, G.A., Fredrickson, H.L., Vestal, J.R., White, D.C., 1982. Sensitive assay, based on 959 hydroxy fatty acids from lipopolysaccharide lipid A, from Gram negative bacteria in sediments. 960 Applied and Environmental Microbiology 44.

Russell, J.M., Hopmans, E.C., Loomis, S.E., Liang, J., Sinninghe Damsté, J.S., 2018. Distributions of 5- and 6-methyl branched glycerol dialkyl glycerol tetraethers (brGDGTs) in East African lake sediment: Effects of temperature, pH, and new lacustrine paleotemperature calibrations. Organic Geochemistry 117, 56–69.

Siles, J.A., Margesin, R., 2016. Abundance and Diversity of Bacterial, Archaeal, and Fungal Communities Along an Altitudinal Gradient in Alpine Forest Soils: What Are the Driving Factors? Microbial Ecology 72, 207–220.

Tierney, J.E., Tingley, M.P., 2014. A Bayesian, spatially-varying calibration model for the TEX86 proxy. Geochimica et Cosmochimica Acta 127, 83–106.

Véquaud, P., Derenne, S., Anquetil, C., Collin, S., Poulenard, J., Sabatier, P., Huguet, A., 2021. Influence of environmental parameters on the distribution of bacterial lipids in soils from the French Alps: Implications for paleo-reconstructions. Organic Geochemistry 104194.

Wang, C., Bendle, J., Yang, Y., Yang, H., Sun, H., Huang, J. and Xie, S., 2016: Impacts of pH and temperature on soil bacterial 3-hydroxy fatty acids: Development of novel terrestrial proxies, Organic Geochemistry, 94, 21–31, doi:10.1016/j.orggeochem.2016.01.010.

Wang, C., Bendle, J.A., Zhang, H., Yang, Y., Liu, D., Huang, J., Cui, J., Xie, S., 2018. Holocene temperature and hydrological changes reconstructed by bacterial 3-hydroxy fatty acids in a stalagmite from central China. Quaternary Science Reviews 192, 97–105.

Wang, H., An, Z., Lu, H., Zhao, Z., Liu, W., 2020. Calibrating bacterial tetraether distributions towards in situ soil temperature and application to a loess-paleosol sequence. Quaternary Science Reviews 231, 106172.

Wilkinson, S.G. 1988 Gram-negative bacteria. In: Ratledge C., Wilkinson S.G. (Eds), Microbial Lipids, 1063 vol. 1. Academic Press, New York, pp. 199-488.

[Figure]

Yang, H., Lü, X., Ding, W., Lei, Y., Dang, X. and Xie, S., 2015: The 6-methyl branched tetraethers significantly affect the performance of the methylation index (MBT') in soils from an altitudinal transect at Mount Shennongjia, Organic Geochemistry, 82, 42–53, doi:10.1016/j.orggeochem.2015.02.003.

Yang, Y., Wang, C., Bendle, J.A., Yu, X., Gao, C., Lü, X., Ruan, X., Wang, R., Xie, S., 2020. A new sea surface temperature proxy based on bacterial 3-hydroxy fatty acids. Organic Geochemistry 141, 103975.

Zelles, L., 1999. Fatty acid patterns of phospholipids and lipopolysaccharides in the characterisation of microbial communities in soil: a review. Biology and Fertility of Soils 29, 111–129.

[Figure]

**Reviewer #2: Review**

I have finished the review of the manuscript "development of global temperature and p calibrations based on bacterial 3-hydroxy fatty acids in soils", submitted to biogeosciences by Pierre Vequaud and co-authors. In general, the authors present a good contribution to the developing field of 3-OH fatty acids, by compiling several published and novel altitudinal gradients. The performance of the local and compiled linear regressions are compared with that of brGDGT lipids, for those sites where 5- and 6-methyl compounds were separated (and where the MBT'5ME could thus be calculated). This is an interesting dataset, and a necessary next step in the development of 3-OH fatty acids as temperature proxies.

*We would like to thank the reviewer for his positive comments and acknowledging the work made to evaluate 3-OH FAs as temperature and pH proxies on a global scale.*

However, the performance of the three additional models (multiple linear regression, random forest and k-nearest neighbor), that move away from the previously established RIAN, RAN15 and RAN17 ratios, are not evaluated thoroughly. There is no explanation of what the models represent, and why they allow a better correlation between the 3-OH fatty acids and MAAT or pH, compared to the simple linear regression.

There is for instance no analysis of residuals, and no indication of maximum reconstructed temperatures. I would recommend the authors to look at a recent calibration paper (fi Dearing-Crampton Flood, 2020), to see what the state-of-the-art in lipid proxy calibration is. As it stands, the authors will have a difficult time convincing reader that the more complex models are the preferred choice when doing 3-OH FA-based climate reconstructions.

*As suggested by the reviewer, the detailed characteristics of the models proposed in this work were added (Figs. 7 and 8, Table 2). Thus, the residuals of these models are presented in Figs. 7 and 8. A table (Table 2 in the revised manuscript) with the $R^2$ and RMSE values, the variance of the residuals, the mean absolute error and upper/lower limits of pH and MAAT estimates was added to the manuscript (taking as an example Table 2 in the paper by Dearing-Crampton Flood, 2020).*

*The advantage of the three models of the present manuscript over the simple linear regression models is to take into account the whole suite of bacterial lipids (in our case $C_{10}$-$C_{18}$ 3-OH FAs) to estimate MAAT and pH values, which allows overcoming the limitations related to the use of a single index. In addition, the random forest and k-NN models are non-parametric and non-linear models, allowing to take into account the potential non-linear influences of MAAT/pH on the fractional abundances of 3-OH FAs.*

*The functioning of the different models was clarified in the revised manuscript (section 2.3.) and the choice of the models discussed in detail (section 4.2.). Quantitative evaluation of the influence of each 3-OH FA on the MAAT/pH random forest calibrations is also proposed, thus providing mechanistic details on this model (Fig. 9 in the revised manuscript).. Such a quantitative evaluation was not possible for the multiple linear regression and k-NN models*

*These detailed explanations highlight the advantages of the proposed models, which can easily be used through the web application provided in this study: https://athibault.shinyapps.io/paleotools*
.

[Figure]

When developing a MAAT-calibration, the authors should also mention the following weakness: there is currently only 3 samples with MAAT > 15 ǎˇD ˇC in the dataset (maximum temperature: +- 20 âˇDCˇ ). Do these samples influence the MAAT calibration disproportionally?

*There are > 30 samples collected in locations with MAAT > 15 °C, especially from Mt. Rungwe (Tanzania) and Peruvian Andes. Nevertheless, as explained in L. 324 "the training phase required for the random forests, k-NN and multiple linear regression was performed on 75% of the sample set with an iteration of ten cross-validations per model. Data selection was performed randomly on the dataset but with a stratification modality according to the MAAT or the pH to limit the impact of extreme values".*
*Thus, the developed models include 25 % of the dataset, considered as independent from the training dataset. The test dataset, randomly chosen, indeed contains only 3 samples with MAAT > 15 °C. Nevertheless, the cross validations and stratification modalities allow to reduce the influence of extreme values. To go further, the models proposed in this study were tested again without the extreme temperature values. This did not change the proposed MAAT estimates.*

**Minor comments:**

L 91. Include 'bacterial' before 'lipids'.

*This sentence was modified as suggested .L.91*

L 129. 'more developed statistical approach', perhaps rephrase as 'further development of the statistical approaches'?

*This sentence was modified as "… and refined statistical tools" L. 129*

L 148. For easy comparison: include the altitudinal, pH and MAAT rangesnof the previously published transects. L 200-201. Soil sensors have been used, but the calibration is done with mean annual air temperature (MAAT) instead. As the soil sensor data is not used or discussed further, I'd remove their mention here, and use the MAAT for this site as well. L 322 (and further in the manuscript). As far as I can see, the supp. tables referred to are not present. Please include these tables in a revised version.

*pH and MAAT ranges were added for all the previously published transects. In the supplementary tables the new data obtained from Mt Pollino, Shegyla, Peruvian and Chilean Andes are included. For the other previously published transects, the reader is directed towards the references of the corresponding papers. We removed the mention of the soil sensors in the text. Supp. Tables are included in the revised version.*

L 541. Perhaps the authors can refer here to De Jonge et al., 2019, who argue that temperature can modulate the soil bacterial composition, and the dependency between MBT'5ME and soil temperature?

*This reference was added to the manuscript. L.559*

L551. Here the authors should explain better how 'more complex models' allow to take the complexity of each site into account. At this point, in the manuscript, we have observed that the linear dependency between the 3-OH fatty acids is better on a local scale, than on a global scale. What mechanism do the authors propose for this, and how can a more complex model correct for it?

*It should be highlighted that the relative abundances of the 3-OH FAs were linearly correlated with MAAT/pH at a local scale, but only along some of the transects. The absence of linear relationships at a local scale for part of the sites, and more generally at a global scale, may at least "be partly due to the heterogeneity of soils encountered along a given altitudinal transect, representing specific microenvironments and to the large diversity of bacterial communities in soils from different elevations (Siles and Margesin, 2016)", as specified in the manuscript (L. 476).*

*The models proposed in this study for MAAT/pH reconstruction are based on the whole suite of 3-OH FAs ($C_{10}$-$C_{18}$) instead of indices. They allow better capturing the complexity of the microoenvironments found along each transect and the variability of the lipid distribution, in contrast with linear models. In addition, two of the three models presented in this paper (k-NN and random forest) are both non-parametric and non-linear and present the advantage of taking into account the potential non-linear relationships between the relative abundances of 3-OH FAs and environmental parameters (here MAAT / pH).*

L567. Have the authors done a selection of the fatty acids that are necessary for the model? For instance, forward selection or reverse selection? Does the model not suffer from overfitting? Is there any fatty acids that are generally present in low abundance (and can thus be absent in geological archives) that are important for the regression? If yes, is it prudent to include these low abundance compounds in the model as well?

*In order to prevent any overfitting of the models, several steps were followed and are clearly stated in the revised manuscript (section 2.3.):*
*(1) Cross-validation of the training dataset to find the optimal combination of 3-OH FA relative abundances and maximize the model performance.*
*(2) No pre-processing (i.e.no pre-selection) of the individual 3-OH FAs. All the 3-OH FA homologues, whatever the abundance, were used in the different models to keep the maximum variability and take into account the specificity and complexity of each altitudinal transect.*
*(3) Testing of the model on a randomly selected dataset (i.e. test dataset) that was not used to build the model.*
*In order to clarify the importance of the different 3-OH FAs in the models, figures presenting the variance of the different homologues in each model are proposed in the revised manuscript (Supp. Figs. 4 and 5)*

L 587. The random forest model and k-NN model need to be explained much better before the results are presented. What are they based on and how do they compare variability in the lipid distribution with the MAAT? L 593. Same comments as at L 567.

*As explained above, each model is better explained both in the material and method section and in section 4.2.), and the results of these different new models are directly and*

[Figure]

*statistically compared to those based on previously defined indices ($RAN_{15}$, $RAN_{17}$ and RIAN; cf. Table 2 in the revised manuscript).*

L 651. Can the authors comment on the probable source of the 3-OH fatty acids in the speleothem? Can we assume that all lipids are derived from the soil, or is there a (variable) proportion produced in the cave environment as well?

*Wang et al. (2018), who analyzed 3-OH FAs in this speleothem, suggested that these compounds are largely derived from the overlying soils based on geochemical and microbiological analyses. Indeed, as specified by these authors, "the broad similarity of 3-OH-FA distributions in the overlying soils and stalagmites, supported by the site-specific analyses of bacterial diversity and transport pathways, supports a major contribution of 3-OH-FAs from Gram-negative bacteria dwelling in the overlying soils to the stalagmite samples". As also stated by Wang et al. (2018), this does not totally exclude that a proportion of the lipids may be derived from the cave ecosystem. This point is clearly mentioned in the revised manuscript (beginning of section 4.3.).*

Refs: Dearing Crampton-Flood E., Tierney J. E., Peterse F., Kirkels F. M. S. A. and Sinninghe Damsté J. S. (2020) BayMBT: A Bayesian calibration model for branched glycerol dialkyl glycerol tetraethers in soils and peats. Geochimica et Cosmochimica Acta 268, 142–159. De Jonge C., Radujkovi´c D., Sigurdsson B. D., Weedon J. T., Janssens I. and Peterse F. (2019) Lipid biomarker temperature proxy responds to abrupt shift in the bacterial community composition in geothermally heated soils. Organic Geochemistry, S0146638019301275.

**References**

*Dearing Crampton-Flood, E., Tierney, J.E., Peterse, F., Kirkels, F.M.S.A., Sinninghe Damsté, J.S., 2020. BayMBT: A Bayesian calibration model for branched glycerol dialkyl glycerol tetraethers in soils and peats. Geochimica et Cosmochimica Acta 268, 142–159.*

*Siles, J.A., Margesin, R., 2016. Abundance and Diversity of Bacterial, Archaeal, and Fungal Communities Along an Altitudinal Gradient in Alpine Forest Soils: What Are the Driving Factors? Microbial Ecology 72, 207–220.*

*Wang, C., Bendle, J.A., Zhang, H., Yang, Y., Liu, D., Huang, J., Cui, J., Xie, S., 2018. Holocene temperature and hydrological changes reconstructed by bacterial 3-hydroxy fatty acids in a stalagmite from central China. Quaternary Science Reviews 192, 97–105.*

---

## Author Response (AR2)

**Associate editor comments:**

**Summary**

First of all, let me thank both reviewers again for their review and you for your reply. I don't want to be too difficult but a slightly more cleaned up version of the annotated manuscript would have been nice. This multi-coloured mash up, I assume with different colours coming from different co-authors, with French comments in the side-line is not great to read. I also have some other comments. I think new proxies always come with their issues, nothing to be ashamed off. I also think old or more established proxies also have their issues, that is one of the reasons new proxies are always welcome. Multiple proxies for the same parameter and based on different compounds, organisms and analytical techniques are always great to have and use together. Therefore, I do not think you need to sell this proxy based on hydroxy fatty acids by making it very clear the more established proxy has a relatively large RMSE. It is not like your proposed proxy has no issues itself. Personally, I would have framed this, line 88 to 95, a bit different. My guess is that would have also helped with the response. The multi proxy approach is great and wonderful, but for that you need multiple proxies. Something along these lines would have been great. The "despite improvements it is still not great" statement is not necessary and completely ignore the fact that all proxies have their good and bad sides. As long as your proxy is not approaching perfection, universally applicable with extremely small errors just be careful.

I like the multiple models used. I also like the explanation of why one works better under certain conditions etc. It also clearly indicates that this is in the trail and error phase. As the authors mention, more soils need to be analysed both for the calibrations as well as applications. The application here is nice and it helps explain some of the pros and cons of the different models, but it is also fairly limited and definitely not the dataset that ends all discussion on the applicability of these models. Be careful on how much emphasis you want to put on the example

*We would like to thank once again the reviewers and especially the associate editor for his positive and constructive comments. We would like to apologize for the previous unclean version of the annotated manuscript. This was corrected in the new revised version.*

*A detailed list of changes and arguments answering to the different comments is provided below. The line numbers are those of the annotated version of the manuscript.*

*We totally agree with the editor that the different molecular proxies are complementary and should be presented as such. The introduction was modified to take this comment into account:*
*"Even though brGDGT proxies were largely investigated over the last 10 years (e.g. De Jonge et al., 2014; Dearing Crampton-Flood et al., 2020) and were applied to various paleorecords (e.g, Coffinet et al., 2018; Wang et al., 2020), new molecular proxies, independent of and complementary to brGDGTs, are needed to improve the reliability of temperature reconstructions in terrestrial settings." Line 91*
*We also agree with the editor that the paleoapplication presented in this paper is only a first test of the applicability of the global calibrations proposed in the manuscript. Additional studies are needed to improve the models and further test their potential. A sentence was added at the end of the discussion to highlight this point:*

[Figure]

*"Additional paleoapplications are also required to further test and validate the applicability of the global MAAT and pH calibrations based on 3-OH FAs presented in this study."* Line 884

**Minor comments:**

Line 114-115: suggesting that regional relations may be more adapted to apply RAN15 as a temperature proxy in soils. I assume you mean something like "suggesting that regional or local RAN15 calibrations maybe more appropriate to apply".

*This sentence was modified as follows: "...suggesting that regional or local $RAN_{15}$ relations may be more appropriate to apply for temperature reconstruction in terrestrial settings." Line 114*

Line 146-150: The sentence that starts with "In addition" and ends with "settings". This has become a very long and difficult to follow sentence.

*This sentence was amended as follows: "In addition to linear regressions, non-parametric, machine learning models were used to improve the global relationships between 3-OH FA distribution and MAAT/pH. These models present the advantage of taking into account non-linear environmental influences, in line with the intrinsic complexity of the environmental settings." Line 140*

Line 155-157: again without any indication on how great the 3-OH FA proxy might be the authors are very negative about a more established proxy. This is not necessary and only puts readers off.

*The negative part of the sentence was removed, as it was, indeed, not necessary.*
*"As brGDGTs are the only microbial organic proxies which can be used for temperature and pH reconstructions in terrestrial settings so far, they can serve as a reference proxy to understand the temperature and pH dependency of 3-OH FAs analyzed in the same dataset. 3-OH FAs and brGDGTs have thus been concomitantly analyzed to assess their reliability and complementarity as independent temperature and pH proxies." Line 146*

Line 386: never start a sentence with But, ok, only very rarely start a sentence with But. Why not however? Think about rewriting up until line 431. Quite difficult to follow.

*Following this comment, theses sentences were rephrased:*
*"To overcome this limitation of the k-NN method, data selection was performed randomly on the dataset with a stratification modality according to the MAAT or the pH. This approach allows to limit the impact of extreme values as detailed below." Line 327*

Line 530: order of magnitude, from 0.1 to 1? Sounds very dramatic order of magnitude, varied between 0 and 1 is much less dramatic.

[Figure]

*This sentence was amended as follows: "The RIAN index varied between 0.1 and 0.8 among the eight elevation transects (Table 1)." Line 401*

Line 662: low Ph ranges, I think you mean narrow or small.

*This sentence was corrected as follows: "...supporting the hypothesis that narrow pH ranges limit the potential of obtaining linear relationships" Line 485*

Line 794: are pooled very narrowly?

*In order to be clearer, this sentence was modified as follows: "...where the samples from this region are pooled separately from the rest of the dataset." L.575*

Line 844: "and sources of these compounds" I can guess what you try to say, but I think it is best if it is written more clearly.

*This sentence was corrected as follows: "This represents a limitation, as the relative distribution of bacterial lipids can be concomitantly influenced by several environmental parameters (e.g. Véquaud et al., 2021) and can also depend on the diversity of the bacteria producing these compounds (Parker et al., 1982; Bhat and Carlson, 1992; Zelles, 1999)." Line 605*

Line 845-847: Who is using bacterial relative abundances. No one measured bacterial relative abundances for this manuscript, right?

*Indeed, there was a missing word in this sentence. This was amended: "In contrast, using bacterial lipid relative abundances rather than a single index in the relationships with environmental variables appears less restrictive, and more representative of the environmental complexity." Line 610*

Line 937: All or the C10 to C18? If I understood it correctly there were more.

*In order to be clearer, this sentence was corrected as follows: "All the 3-OH FA homologues of Gram-negative bacteria origin (i.e. with chain lengths between $C_{10}$ and $C_{18}$; Wilkinson et al., 1988) were included in the models whatever their abundance to keep the maximum variability and take into account the specificity and complexity of each altitudinal transect." Line 639*

*It should be noted that 3-OH FAs are widely distributed in microorganisms with chain lengths up to 26 C and can be produced by e.g. yeasts, fungi, and Gram-positive bacteria in addition to Gram-negative ones. Nevertheless, in the present paper we only consider 3-OH FAs with 10 to 18 C, typical for Gram-negative bacteria (Wilkinson et al., 1988).*

Line 953: whole suite limited to C10 to C18?

*This was rephrased: "This model, which takes into account the Gram-negative bacterial 3-OH FAs ($C_{10}$-$C_{18}$; Wilkinson et al., 1988), presents..." L. 657*

[Figure]

Line 1181-1182: along some, but not all, of the altitudinal transects … At least I guess that's what you want to say.

*This sentence was modified accordingly to this comment: "This may explain why linear relationships between the $RAN_{15}/RAN_{17}$ and MAAT could be established along some, but not all, of the altitudinal transects investigated until now…" Line 674*

Line 1345-1346: by working on the microbial level, please explain?

*In this section, it is hypothesised that pH can influence the biosynthesis of new 3-OH FAs, and that this mechanism influences all 3-OH FAs between $C_{10}$ and $C_{18}$. But this hypothesis requires the study of the membrane adaptation mechanisms of the 3-OH FAs source bacteria.*
*Thus, in order to be clearer, this sentence was corrected as follows:*
*"These results suggest that soil Gram-negative bacteria may respond to pH variations by modifying the whole distribution of associated 3-OH FAs ($C_{10}$-$C_{18}$). This would need to be further confirmed by e.g. investigating the influence of pH variations on pure strains of Gram-negative bacteria isolated from soils. Line 724*

Line 1813: is it really much smaller than the 6.5 to 19.7 of the k-NN method?

*In order to be more rigorous, the sentence was corrected as follows "Finally, the random forest model yielded MAAT estimates between 10.6 and 19.3°C, i.e. a smaller estimation range than the k-NN algorithm and multiple regression model (Supp. Fig. 4)." Line 790*

Line 217-2019; I don't this is needed in this way. The proxies can strengthen each other, use that. I mean the information is correct, I would just frame it a less "confrontational".

*The confrontational part of this sentence was removed, i.e ". This RMSE is also much lower than the one related to the latest global MAAT-brGDGT calibrations (> 4 °C; De Jonge et al., 2014; Naafs et al., 2017; Dearing Crampton-Flood et al., 2020), even though the latter are based on a larger number of soil samples than the global 3-OH FA model proposed in the present study."*

In your reference database, could you please change J.S.S. Damsté into J.S. Sinninghe Damsté.

*This was corrected.*